# Towards Understanding Catastrophic Forgetting in Two-layer Convolutional Neural Networks

**Boqi Li** [1]  **Youjun Wang** [1]  **Weiwei Liu** [1]

## Abstract

Continual learning (CL) focuses on the ability of models to learn sequentially from a stream of tasks. A major challenge in CL is catastrophic forgetting (CF). CF is a phenomenon where the model experiences significant performance degradation on previously learned tasks after training on new tasks. Although CF is commonly observed in convolutional neural networks (CNNs), the theoretical understanding about CF within CNNs remains limited. To fill the gap, we present a theoretical analysis of CF in a two-layer CNN. By employing a multi-view data model, we analyze the learning dynamics of different features throughout CL and derive theoretical insights. The findings are supported by empirical results from both simulated and real-world datasets.

## 1. Introduction

In recent years, large volumes of data are generated. Designing a general machine learning model to adapt quickly to changing environments becomes a significant concern (Ebrahimi et al., 2021; Goldfarb et al., 2024). One of the popular approaches is to train the model on a sequence of multiple tasks, which is called continual learning (CL) (Parisi et al., 2019; Chen & Liu, 2018; Wang et al., 2024; Gao & Liu, 2025a). A major challenge in CL is that after training on a few new tasks, the model "forgets" knowledge learned from previous tasks with a significant performance degradation, which is also known as catastrophic forgetting (CF) (McCloskey & Cohen, 1989; Goodfellow et al., 2015; Shi et al., 2021; Korbak et al., 2022). CF is widely observed in various models, such as linear models (Evron et al., 2022),

[1]School of Computer Science, Wuhan University National Engineering Research Center for Multimedia Software, Wuhan University Institute of Artificial Intelligence, Wuhan University Hubei Key Laboratory of Multimedia and Network Communication Engineering, Wuhan University. Correspondence to: Weiwei Liu <liuweiwei863@gmail.com>.

*Proceedings of the 42$^{nd}$ International Conference on Machine Learning*, Vancouver, Canada. PMLR 267, 2025. Copyright 2025 by the author(s).

convolutional neural networks (CNNs) (Goodfellow et al., 2015), transformers (Kotha et al., 2024), and others, across different CL scenarios, such as task-incremental CL (Sun et al., 2023), class-incremental CL (Babakniya et al., 2023), domain-incremental CL (Jeeveswaran et al., 2024) and so on.

Alleviating CF is a core challenge in CL, and many methods (Kirkpatrick et al., 2017; Serrà et al., 2018; Gao & Liu, 2023; Wu et al., 2024; Gao & Liu, 2025b) have been proposed to address CF in recent years. Despite tangible improvements in the field of CL, the theoretical understanding of CF remains underexplored. Recent theoretical works investigate CF from the perspectives of optimization (Doan et al., 2021; Lin et al., 2023; Evron et al., 2022) and statistics (Goldfarb & Hand, 2023; Zhao et al., 2024). However, few of these studies analyze the learning dynamics in non-linear models, particularly CNNs.

To bridge the gap, we investigate CF in a two-layer CNN model. Following Allen-Zhu & Li (2023); Shen et al. (2022), we employ a multi-view data model and focus on task-incremental binary-classification CL. In a multi-view data model, the data consists of features and noises. By analyzing the learning dynamics of the features and noise, we identify which components the model ultimately learns after several time steps. Our data model includes four components: task-specific features, general features, random features, and background noise. Using gradient descent (GD) to update the model, we analyze the learning process of the model. Under mild assumptions, our theoretical results offer insights into the underlying causes of CF. Our findings suggest that when task-specific features are learned more rapidly than others, the learning of general features with low signals is hindered. Additionally, when training on new tasks, we observe that if the random feature has a large signal, CF occurs, causing the model to forget knowledge from previous tasks. We also offer a theoretical insight into the effectiveness of replay-based methods (Rolnick et al., 2019; Jeeveswaran et al., 2023), which are commonly used to mitigate CF.

We perform numerical analysis and conduct experiments on real-world datasets to further validate our theoretical findings. In numerical analysis, we validate the key components

derived from our theoretical results. The experiments in real-world datasets further support our theoretical analysis.

The contributions of our work can be summarized as follows:

- We provide a novel theoretical framework to analyze CF in a two-layer CNN, focusing on the learning dynamics of different features and noises.

- Theoretical findings reveal the existence of CF for two reasons: the task-specific feature has a larger signal than the general feature, and the task-specific feature appears as a random feature with a strong signal in other tasks.

- We validate our theoretical results using experiments on both simulated and real-world datasets, demonstrating the practical relevance of our findings.

## 2. Related Works

**Theoretical Analysis on Catastrophic Forgetting.** A few works study CF in linear regression problems. Evron et al. (2022) show connections between CL in the linear setting and alternating projections and the Kaczmarz methods. The authors also study the effect of task orderings and similarity on CF. Li et al. (2023) investigate regularized CL with two linear regression tasks. Goldfarb & Hand (2023) exhibit the effect of over-parameterization on performance degradation due to CF. Lin et al. (2023) provide an explicit characterization of CF and generalization error to analyze how over-parameterization, task similarity, and task ordering are relevant to CF and generalization error. Zhao et al. (2024) provide a statistical analysis of regularization-based CL on a sequence of tasks, and study the effect of regularization terms on model performance. Li et al. (2025) provide a theoretical study of mixture-of-experts models in CL. (Ding et al., 2024) analyze factors contributing to forgetting under linear models in CL. Zheng et al. (2025) examine memory-based CL under overparameterized linear models.

There are also some works that investigate CF in neural networks. Doan et al. (2021) analyze CF under the Neural Tangent Kernel (NTK) regime, and they propose measuring task similarity using an NTK overlap matrix. Andle & Sekeh (2022) show a theoretical analysis of information flow through layers in linear networks for task sequences, and the effect of information flow on learning performance. Cao et al. (2022a) propose a provable CL algorithm that maintains and refines the internal feature representation. Benjamin et al. (2024) introduce the concept of NTE in CL, reformulating a single neural network as an ensemble of fixed classifiers. Their analysis focuses on the linear networks with non-linear activation fucntion. Different from

the above works, we study CF in a two-layer CNN using a multi-view data model.

**Theoretical Analysis in Two-Layer CNN Model.** Recently, a lot of works study the learning dynamics in a two-layer CNN using a multi-view data model under an over-parameterization regime. The multi-view data model is firstly proposed by Allen-Zhu & Li (2023), with a framework to study the learning dynamics in ensemble learning, knowledge distillation, and self-distillation. Within a similar framework, Jelassi & Li (2022) investigate the effect of momentum in (S)GD for improving generalization of CNNs, Shen et al. (2022) study the effect of data augmentation, Huang et al. (2024) investigate the federated learning, Li & Liu (2024) investigate the backdoor poisoning attacks and Kou et al. (2023); Meng et al. (2024) study benign overfitting in CNN models.

However, previous works mainly focus on learning a single task, while we analyze the learning dynamics across a sequence of tasks. A major difficulty in our setting is that, at the beginning of learning a new task, the weights of the model are not randomly initialized, which is not discussed in the above works. Therefore, it is imperative to study the dynamics of CL in a two-layer CNN model.

## 3. Preliminaries

Consider a standard task-incremental CL problem, where a sequence of tasks indexed by $\tau = 1, \ldots, \mathcal{T}$. Each task $\tau$ is a binary classification problem over a dataset $\mathcal{S}^{(\tau)} = \left\{ \left( \mathbf{x}_i^{(\tau)}, y_i^{(\tau)} \right) \right\}_{i=1}^{n_\tau}$. Let $\mathcal{D}_{\mathbf{z}}^{(\tau)}$ be the distribution of $Z^{(\tau)} = (X^{(\tau)}, Y^{(\tau)})$ over $\mathcal{Z} = \mathcal{X} \times \mathcal{Y}$. In this work, we consider the task-incremental scenario with $\mathcal{T} = 2$, meaning the model sequentially trains on a pair of tasks.

**Data.** In each task $\tau$, we assume the data points are drawn from a multi-view data model (Allen-Zhu & Li, 2023). In our multi-view data model, each data point $\mathbf{x}$ consists of $P + 3$ non-overlapping patches $\mathbf{x} = (\mathbf{x}_1, \ldots, \mathbf{x}_{P+3}) \in \mathbb{R}^{d \times (P+3)}$, and each patch is a vector of dimension $d$. We suppose that there exists a general feature space $\mathbb{O} \in \mathbb{R}^{d \times 3}$ spanned by three basis vectors $Features = \{\mathbf{e}_1, \mathbf{e}_2, \mathbf{e_{rob}}\}$. In each task, the task-specific, general, and random features are defined as $\mathbf{u}^{(\tau)} = \alpha_u \mathbf{e}_\tau$, $\mathbf{v}^{(\tau)} = \alpha_v \mathbf{e_{rob}}$, $\boldsymbol{\zeta}^{(\tau)} = \alpha_\zeta \mathbf{e}_{3-\tau}$. Additionally, $P$ background noise vectors $\{\boldsymbol{\xi}(\tau)_p\}_{p=1}^{P}$ exist in $\mathbf{x}$, and the distribution of $\boldsymbol{\xi}(\tau)$ is denoted by $\mathcal{D}_\xi$. We define the feature-noise multi-view data model as follows:

**Definition 3.1.** In task $\tau$, given vectors $\mathbf{u}^{(\tau)}, \mathbf{v}^{(\tau)}, \boldsymbol{\zeta}^{(\tau)}$, and the noise distribution $\mathcal{D}_\xi$, let $U$ be the uniform distribution and $[n]$ is used to denote the set $\{1, \ldots, n\}$. A data point $\mathbf{z} = (\mathbf{x}, y)$ is drawn from the distribution $\mathcal{D}_{\mathbf{z}}^{(\tau)}$ which is defined as follows:

1. Draw two Rademacher variables $(y, \epsilon) \sim U\{+1, -1\}^2$.

2. Given $y, \epsilon$, arbitrarily choose three distinct patches $p^u$, $p^v$ and $p^\zeta$. The feature patches are set as $\{\mathbf{x}_{p^u}, \mathbf{x}_{p^v}, \mathbf{x}_{p^\zeta}\} = \{y\mathbf{u}, y\mathbf{v}, \epsilon\boldsymbol{\zeta}\}$.

3. Each remaining background patch $p^\xi \in [P+3] \setminus \{p^u, p^v, p^\zeta\}$ of $\mathbf{x}$ is set as $\mathbf{x}_{p^\xi} = \boldsymbol{\xi}$, where $\boldsymbol{\xi} \sim \mathcal{D}_\xi$.

*Remark* 3.2. In our data model, each task has both task-specific and general features. In a pair of tasks, $\mathbf{v}^{(1)} = \mathbf{v}^{(2)} = \mathbf{v}$ is an aligned and robust feature that allows the model to achieve high accuracy on both tasks. Using task-specific features $\mathbf{u}^{(\tau)}$, the model can only correctly classify the data points in task $\tau$. Since $\boldsymbol{\zeta}^{(3-\tau)} = \alpha_u^{-1}\alpha_\zeta \mathbf{u}^{(\tau)}$, if the model has not learned $\mathbf{u}^{(3-\tau)}$, and $\alpha_\zeta$ is large, the sign of the model's output in the second task matches the sign of $\epsilon$, which is independent of $y$. The presence of a random feature ensures that the task-specific feature can only be used for classification within its corresponding task.

To ilustrate the data model, we consider an example in two-task incremental learning based on Definition 3.1:

- Task 1: Data $\mathbf{x}$ with label $y$ contains features $\{y\alpha_u\mathbf{e}_1, y\alpha_v\mathbf{e}_{rob}, \epsilon\alpha_\zeta\mathbf{e}_2\}$ and noises.

- Task 2, Data $\mathbf{x}$ with label $y$ contains features $\{y\alpha_u\mathbf{e}_2, y\alpha_v\mathbf{e}_{rob}, \epsilon\alpha_\zeta\mathbf{e}_1\}$ and noises.

*Remark* 3.3. In such an example, all of $\{\mathbf{e}_1, \mathbf{e}_{rob}, \mathbf{e}_2\}$ appear in each task, but only $y\alpha_u\mathbf{e}_\tau$ and $y\alpha_v\mathbf{e}_{rob}$ are correlated with the true label in task $\tau$, which means the two features can be used for classification in task $\tau$. $\epsilon\alpha_\zeta\mathbf{e}_{3-\tau}$ is called a random feature, which is uncorrelated with the true label. In such a model, the task-specific feature is only used in its respective task. $\alpha_u, \alpha_v, \alpha_\zeta$ controls the strength of the features.

In this paper, we assume $\mathcal{D}_\xi$ is a zero-mean Gaussian distribution $\mathcal{N}(0, \sigma_\xi^2 d^{-1}(\mathbf{I}_d - \mathbf{e}_1\mathbf{e}_1^T - \mathbf{e}_2\mathbf{e}_2^T - \mathbf{e_{rob}}\mathbf{e_{rob}}^T))$. We use $\mathcal{I}$ to denote the index set of $\mathcal{S}$. The training set $\mathcal{S}_{tr}^{(\tau)}$ consists of $n^{(\tau)}$ independent and identically distributed (i.i.d.) data points drawn from $\mathcal{D}_\mathbf{z}^n$. In this work, we suppose that $\forall \tau, n^{(\tau)} = n$.

**Model.** We use a patch-wise CNN architecture $F(\mathbf{x})$ with $C$ channels, which is defined as

$$F(\mathbf{x}) = \sum_{c=1}^C \lambda_c \sum_{p=1}^{P+3} \phi\left(\langle \mathbf{w}_c, \mathbf{x}_p\rangle\right), \qquad (1)$$

where $\{\mathbf{w}_1, \ldots, \mathbf{w}_C\}$ are the parameters in the first layer, $\{\lambda_1, \ldots, \lambda_C\}$ are the parameters in the last layer, and $\phi(z) = z^q$ is the activation function. In this work, we

use $q = 3$ while our results can be extended to any $q \geq 3$. The CNN predicts label as $\hat{y} = \text{sign}(F(\mathbf{x}))$, where $\text{sign}(\cdot)$ denotes the sign function.

**Training Process.** The entire training process can be split into two stages:

In the first stage, the model begins with initialization. We use Gaussian initialization to initialize the weights of the first layer, i.e. $\forall c \in [C], \mathbf{w}_c^{(1)}(0) \sim \mathcal{N}(0, \sigma_0^2\mathbf{I}_d)$, while the last layer is fixed and set as the all-one vector, i.e., $\forall c \in [C], \lambda_c = 1$. We then use the training data from the first task $\mathcal{S}_{tr}^{(1)}$ to train the model. After $T_1$ epochs, we stop training the model, and the model at the end of the first stage is denoted by $F^{(1)} = F_{T_1}^{(1)}$ with parameters $\mathbf{w}_c^{(1)}(T_1)$.

In the second stage, the model is initialized as $\mathbf{w}_c^{(2)}(0) = \mathbf{w}_c^{(1)}(T_1)$, and we use the training data of the second task $\mathcal{S}_{tr}^{(2)}$ to train the model for an additional $T_2$ epoch. The model at the end of the second stage is denoted by $F^{(2)} = F_{T_2}^{(2)}$ with parameters $\mathbf{w}_c^{(2)}(T_2)$.

We utilize the logistic loss $\ell(F(\mathbf{x}), y) = \log\left(1 + e^{-yF(\mathbf{x})}\right)$ as the loss function and apply gradient descent (GD) to optimize the parameters. Throughout the learning process, the last layer remains fixed. Given a learning rate $\eta$, at round $t$ of stage $\tau \in \{1, 2\}$, the parameters of the network are updated as follows:

$$\mathbf{w}_c^{(\tau)}(t+1) - \mathbf{w}_c^{(\tau)}(t)$$

$$= -\frac{\eta}{n}\sum_{i=1}^n \ell'(F(\mathbf{x}_i^{(\tau)}), y_i^{(\tau)})\nabla_{\mathbf{w}_c^{(\tau)}}\ell(F(\mathbf{x}_i^{(\tau)}), y_i^{(\tau)})$$

$$= -\frac{\eta}{n}\sum_{i=1}^n\sum_{p=1}^{P+3} y_i^{(\tau)}\lambda_c\ell'(F(\mathbf{x}_i^{(\tau)}), y_i^{(\tau)})\phi'\left(\left\langle\mathbf{w}_c^{(\tau)}(t), \mathbf{x}_{i,p}^{(\tau)}\right\rangle\right)\mathbf{x}_{i,p}^{(\tau)}.$$

The CF is formally defined as follows:

**Definition 3.4** (Catastrophic Forgetting). We say that CF occurs in the training process when the following conditions are satisfied:

1. At the end of the first stage, with a high probability $1 - \delta_1$, the model correctly classifies the test data from the first task:

$$\mathbb{P}_{(\mathbf{x}, y)\sim\mathcal{D}_\mathbf{z}^{(1)}}[yF^{(1)}(\mathbf{x}) > 0] > 1 - \delta_1.$$

2. At the end of the second stage, with a high probability $1 - \delta_2$, the accuracy on the first task degrades significantly:

$$\mathbb{P}_{(\mathbf{x}, y)\sim\mathcal{D}_\mathbf{z}^{(1)}}[yF^{(2)}(\mathbf{x}) > 0] < 1/2 + \delta_2.$$

*Remark* 3.5. For a binary classification task, an algorithm that outputs randomly would achieve approximately 50%

accuracy. The above definition implies that when CF occurs, the model's performance on the first task is close to random guessing. By fixing the last layer, we can focus on the learning dynamics of features to find the hidden reason behind CF. It is crucial for identifying whether and how certain features contribute to CF.

We use the standard asymptotic notations $O, \Theta, \Omega$ in this paper, and $\widetilde{O}, \widetilde{\Theta}, \widetilde{\Omega}$ are used to hide the log factors in $O, \Theta, \Omega$, respectively. We use $f \leq o(g)$ to denote that for every $\alpha > 0$, there exists $x_0$ such that for all $x > x_0$ we have $f(x) \leq \alpha g(x)$. We use $f \geq \omega(g)$ to denote that for every $\alpha > 0$, there exists $x_0$ such that for all $x > x_0$ we have $f(x) \geq \alpha g(x)$. We suppose the following conditions hold in our analysis.

**Condition 3.6.** In the training process, for any $\tau \in \{1, 2\}$, we have

1. The feature vectors $\mathbf{u}^{(\tau)}, \mathbf{v}^{(\tau)}, \boldsymbol{\zeta}^{(\tau)}$ are mutually orthogonal, and the norms of $\mathbf{u}^{(\tau)}, \mathbf{v}^{(\tau)}, \boldsymbol{\zeta}^{(\tau)}$ satisfy $0 \leq \alpha_v, \alpha_\zeta < \alpha_u \leq O(1)$.

2. The network is over-parameterized, and $nP \leq o\left(\sqrt{d}\right)$. The number of channels $C$ is of the order of logarithm of $d$, i.e., $C = \Theta(\log d)$.

3. The network is initialized with a small variance, i.e., $P\sigma_0 \leq o(1)$ and the variance of background noise is large, i.e., $\omega(1) \leq \sigma_\xi \leq o\left(1/(P\sigma_0)\right)$.

4. The learning rate is not large, i.e., $\eta \leq O\left(1/(\sigma_0 \alpha_u^3)\right)$.

5. Signal-to-Noise Ratio (SNR) is sufficiently large to ensure that the model does not fit the noise during the training process, i.e., $n\alpha_u^3/\sigma_\xi^3 \geq \omega(1)$.

*Remark* 3.7. In Condition 3.6, the condition on $\sigma_\xi$ is to ensure that at the beginning of the training process, the network cannot easily classify the data with the task-specific feature. We use a small $\sigma_0$ to ensure that the output of the network after initialization is the order of $o(1)$. The last condition is a lower bound for $\text{SNR}^3 := \alpha_u^3/\sigma_\xi^3$, and a similar condition of SNR is also shown in Cao et al. (2022b)

## 4. Theoretical Insights in Learning Process

We first simplify our CNN model as $C = 1$ and consider four ideal models. Let $\kappa > 0$ be a large constant. First, we consider task-specific models. Using $\mathbf{w}_1 = \kappa \mathbf{u}^{(1)}$, the model achieves low training and test error in the first task but has low accuracy for data drawn from the second task. Similarly, using $\mathbf{w}_1 = \kappa \mathbf{u}^{(2)}$, the model only achieves high training and test accuracy in the second task and misclassifies half of the data points drawn from the first task. Both of these models fail to generalize robustly to unseen tasks.

In contrast, we consider two task-robust models. Firstly, employing the robust vector $\mathbf{v}^{(1)} = \mathbf{v}^{(2)}$ allows for accurate data classification, expressed as $\mathbf{w}_1 = \mathbf{w}^{\mathbf{rob}} = \kappa \mathbf{v}^{(1)}$, successfully categorizing both training and test datasets in both tasks. Secondly, the model can be built using $\mathbf{u}^{(1)}$ and $\mathbf{u}^{(2)}$, represented as $\mathbf{w}_1 = \mathbf{w}^{\mathbf{gen}} = \kappa \mathbf{u}^{(1)} + \kappa \mathbf{u}^{(2)}$. When $\alpha_\zeta < \alpha_u$, both vectors $\mathbf{u}^{(1)}$ and $\mathbf{u}^{(2)}$ can effectively classify the data across both tasks.

Among the four models outlined previously, the goal of the user is to develop a model with the ability of generalization, akin to the last two models $\mathbf{w}^{\mathbf{rob}}$ and $\mathbf{w}^{\mathbf{gen}}$. We then analyze the gradient descent dynamics involved in sequential task learning and investigate the conditions under which generalized models can be acquired. For the remainder of this section, we assume $C > 1$ and set $\sigma_\xi = 0$. The conditions involving $\sigma_\xi$ in Condition 3.6 can be ignored in this section.

### 4.1. Features Learned by the Model in the First Stage

At the beginning of the first stage, the weights of $F$ are closer to the initialization. The following lemma shows that the inner products of weight vectors and patch vectors are all small, meaning that none of the channels capture the features at initialization.

**Lemma 4.1.** *Given the weights $\mathbf{w}_c^{(1)}$ initialized as $\mathbf{w}_c^{(1)}(0) \sim \mathcal{N}(0, \sigma_0 \mathbf{I}_d)$, at the beginning of the first stage, with a probability of $1 - O\left(\frac{n^2 P^2 C}{poly(d)}\right)$, we have*

$$\forall \mathbf{e} \in \left\{\mathbf{u}^{(1)}, \mathbf{v}^{(1)}, \boldsymbol{\zeta}^{(1)}\right\}, \max_{c \in [C]} \left|\left\langle \mathbf{w}_c^{(1)}(0), \mathbf{e} \right\rangle\right| \leq \widetilde{O}\left(\|\mathbf{e}\|_2 \sigma_0\right),$$

$$\forall \mathbf{e} \in \left\{\mathbf{u}^{(1)}, \mathbf{v}^{(1)}, \boldsymbol{\zeta}^{(1)}\right\}, \max_{c \in [C]} \left\langle \mathbf{w}_c^{(1)}(0), \mathbf{e} \right\rangle \geq \Omega\left(\|\mathbf{e}\|_2 \sigma_0\right),$$

$$\forall i \in [n], p \in \mathcal{P}_i^\xi, \max_{c \in [C]} \left|\left\langle \mathbf{w}_c^{(1)}(0), \boldsymbol{\xi}_{i,p}^{(1)} \right\rangle\right| \leq \widetilde{O}\left(\sigma_0 \sigma_\xi\right),$$

$$\forall i \in [n], p \in \mathcal{P}_i^\xi, \max_{c \in [C]} \left\langle \mathbf{w}_c^{(1)}(0), \boldsymbol{\xi}_{i,p}^{(1)} \right\rangle \geq \Omega\left(\sigma_0 \sigma_\xi\right).$$

The proof of Lemma 4.1 can be found in Appendix C. We then analyze the dynamics of $\left\langle \mathbf{w}_c^{(1)}, \mathbf{u}^{(1)} \right\rangle$:

$$
\frac{d \left\langle \mathbf{w}_c^{(1)}, \mathbf{u}^{(1)} \right\rangle}{dt}
$$

$$
= -\frac{1}{n} \sum_{i=1}^n \sum_{p=1}^P y_i^{(1)} \ell_i' \phi'\left(\left\langle \mathbf{w}_c^{(1)}, \mathbf{x}_{i,p}^{(1)} \right\rangle\right) \left\langle \mathbf{x}_{i,p}^{(1)}, \mathbf{u}^{(1)} \right\rangle
$$

$$
= -\frac{1}{n} \sum_{i=1}^n \ell_i' \phi'\left(\left\langle \mathbf{w}_c^{(1)}, \mathbf{u}^{(1)} \right\rangle\right) \left\|\mathbf{u}^{(1)}\right\|_2^2.
$$

At initialization, $\forall i \in [n], F(\mathbf{x}_i) = o(1)$, and $\ell_i' = \ell'(y_i^{(1)} F(\mathbf{x}^{(1)})) \approx -1/2$. The dynamic reduces to

an ODE. Let $g(t) = \left\langle \mathbf{w}_c^{(1)}, \mathbf{u}^{(1)} \right\rangle$, we have $g'(t) \approx \left\| \mathbf{u}^{(1)} \right\|_2^2 \phi'(g(t))$. When $g(t) \leq 1$, we have $(g(t)^{-1})' = \alpha_u^2$ due to the definition of $\phi$. Then at $T_u^{(1)} = \Theta\left(\frac{1}{\alpha_u^2 g(0)}\right) = \Theta\left(\frac{1}{\sigma_0 \alpha_u^3}\right)$, we yield $\max_c \left\langle \mathbf{w}_c^{(1)}, \mathbf{u}^{(1)} \right\rangle \geq \Omega(1)$, which implies that $\mathbf{u}^{(1)}$ has been captured by the NN.

By replacing $\mathbf{v}^{(1)}$ to $\mathbf{u}^{(1)}$, we yield that

$$
\begin{aligned}
&\frac{d\left\langle \mathbf{w}_c^{(1)}, \mathbf{v}^{(1)} \right\rangle}{dt} \\
&= -\frac{1}{n} \sum_{i=1}^n \sum_{p=1}^P y_i^{(1)} \ell_i' \phi'\left(\left\langle \mathbf{w}_c^{(1)}, \mathbf{x}_{i,p}^{(1)} \right\rangle\right) \left\langle \mathbf{x}_{i,p}^{(1)}, \mathbf{v}^{(1)} \right\rangle \\
&= -\frac{1}{n} \sum_{i=1}^n \ell_i' \phi'\left(\left\langle \mathbf{w}_c^{(1)}, \mathbf{v}^{(1)} \right\rangle\right) \left\| \mathbf{v}^{(1)} \right\|_2^2.
\end{aligned}
$$

At $T_v^{(1)} = \Theta\left(\frac{1}{\sigma_0 \alpha_v^3}\right)$, we yield $\max_c \left\langle \mathbf{w}_c^{(1)}, \mathbf{v}^{(1)} \right\rangle \geq \Omega(1)$. When $T_v^{(1)} \ll T_u^{(1)}$, which means $\alpha_v \ll \alpha_u$, the feature $\mathbf{v}^{(1)}$ cannot be captured by the CNN model. The above analysis shows that $\mathbf{u}^{(1)}$ is easier to learn than $\mathbf{v}^{(1)}$ when $\mathbf{u}^{(1)}$ has a stronger signal than $\mathbf{v}^{(1)}$.

As for $\boldsymbol{\zeta}^{(1)}$, let $\mathcal{I}_{=}^{(\tau)}$ and $\mathcal{I}_{\neq}^{(\tau)}$ denote the sets $\left\{i : i \in \mathcal{I}^{(\tau)}, y_i = \epsilon_i\right\}$ and $\left\{i : i \in \mathcal{I}^{(\tau)}, y_i \neq \epsilon_i\right\}$, respectively. $n_{=}^{(\tau)}$ and $n_{\neq}^{\tau}$ are used to denote the sizes of $\mathcal{I}_{=}^{(\tau)}$ and $\mathcal{I}_{\neq}^{(\tau)}$, respectively. We have

$$
\begin{aligned}
&\frac{d\left\langle \mathbf{w}_c^{(1)}, \boldsymbol{\zeta}^{(1)} \right\rangle}{dt} \\
&= -\frac{1}{n} \sum_{i=1}^n \sum_{p=1}^P y_i^{(1)} \ell_i' \phi'\left(\left\langle \mathbf{w}_c^{(1)}, \mathbf{x}_{i,p}^{(1)} \right\rangle\right) \left\langle \mathbf{x}_{i,p}^{(1)}, \boldsymbol{\zeta}^{(1)} \right\rangle \\
&= -\frac{1}{n} \left( \sum_{i \in \mathcal{I}_{=}} \ell_i' - \sum_{i \in \mathcal{I}_{\neq}} \ell_i' \right) \phi'\left(\left\langle \mathbf{w}_c^{(1)}, \boldsymbol{\zeta}^{(1)} \right\rangle\right) \left\| \boldsymbol{\zeta}^{(1)} \right\|_2^2 \\
&= \Theta\left(n_{=}^{(1)} - n_{\neq}^{(1)}\right) \phi'\left(\left\langle \mathbf{w}_c^{(1)}, \boldsymbol{\zeta}^{(1)} \right\rangle\right) \left\| \boldsymbol{\zeta}^{(1)} \right\|_2^2.
\end{aligned}
$$

Since $\epsilon$ is a sub-Gaussian random variable, with a high probability, we yield $\left| n_{=}^{(1)} - n_{\neq}^{(1)} \right| \leq \widetilde{O}(\sqrt{n})$. Even if $\alpha_\zeta = \alpha_u$, the update speed of $\max_c \left\langle \mathbf{w}_c^{(1)}, \boldsymbol{\zeta}^{(1)} \right\rangle$ is still slower than $\max_c \left\langle \mathbf{w}_c^{(1)}, \mathbf{u}^{(1)} \right\rangle$. The time step that $\max_c \left\langle \mathbf{w}_c^{(1)}, \boldsymbol{\zeta}^{(1)} \right\rangle$ reaches $\Omega(1)$ is at least $T_\zeta^{(1)} = \Theta\left(\frac{n}{\left(n_{=}^{(1)} - n_{\neq}^{(1)}\right)\sigma_0 \|\boldsymbol{\zeta}^{(1)}\|_2^3}\right) \geq \Omega\left(\frac{\sqrt{n}}{\sigma_0 \alpha_\zeta^3}\right)$.

Note that $\mathbf{u}^{(2)} = \alpha_\zeta^{-1} \alpha_u \boldsymbol{\zeta}^{(1)}$, we have

$$
\begin{aligned}
&\frac{d\left\langle \mathbf{w}_c^{(1)}, \mathbf{u}^{(2)} \right\rangle}{dt} \\
&= \Theta\left(\left(n_{=}^{(1)} - n_{\neq}^{(1)}\right) \alpha_u^{-3} \alpha_\zeta^3\right) \phi'\left(\left\langle \mathbf{w}_c^{(1)}, \mathbf{u}^{(2)} \right\rangle\right) \left\| \mathbf{u}^{(2)} \right\|_2^2.
\end{aligned}
$$

The time step that $\max_c \left\langle \mathbf{w}_c^{(1)}, \mathbf{u}^{(2)} \right\rangle \geq \Omega(1)$ is also at least $T_{u'}^{(1)} = \Theta\left(\frac{n}{\left(n_{=}^{(1)} - n_{\neq}^{(1)}\right)\sigma_0 \alpha_\zeta^3}\right) \geq \Omega\left(\frac{\sqrt{n}}{\sigma_0 \alpha_\zeta^3}\right)$. Finally, after the task-specific feature $\mathbf{u}^{(1)}$ or general feature $\mathbf{v}^{(1)}$ is captured by the model, the condition that $\forall i \in [n], F(\mathbf{x}_i) = o(1)$ and $\ell_i' = \ell'(y_i F(\mathbf{x})) \approx -1/2$ no longer hold, and the update speed of each component decreases.

To summarize, in the first stage, if the general feature $\mathbf{v}^{(1)}$ has a minimal signal, the model only learns the task-specific feature $\mathbf{u}^{(1)}$, and the updates of $\left\langle \mathbf{w}_c^{(1)}, \mathbf{v}^{(1)} \right\rangle$ and $\left\langle \mathbf{w}_c^{(1)}, \boldsymbol{\zeta}^{(1)} \right\rangle$ are both slight. From a practical perspective, the existence of the general feature is difficult to meet in two randomly sampled tasks. Our experiments demonstrate that the general feature does not exist in the majority of cases. When $\left\langle \mathbf{w}_c^{(1)}, \mathbf{u}^{(1)} \right\rangle \geq \Omega(1)$, we stop training the model in the first stage.

### 4.2. Model Forgets Task-Specific Feature in the Second Stage

In the second stage, we show that the model tends to forget $\mathbf{u}^{(1)}$ when the norm of random feature $\boldsymbol{\zeta}^{(2)}$ becomes large. Let $\Delta_c\left(\boldsymbol{\zeta}^{(\tau)}\right) = \left\langle \mathbf{w}_c^{(\tau)}(t+1), \boldsymbol{\zeta}^{(\tau)} \right\rangle - \left\langle \mathbf{w}_c^{(\tau)}(t), \boldsymbol{\zeta}^{(\tau)} \right\rangle$, since $\phi'(\cdot)$ is an even function, we have

$$
\Delta_c\left(\boldsymbol{\zeta}^{(\tau)}\right) = \frac{\eta \lambda_c \|\boldsymbol{\zeta}\|_2^2 \phi'\left(\left\langle \mathbf{w}_c^{(1)}(t), \boldsymbol{\zeta} \right\rangle\right)}{n} (\mathcal{G}_{=}^{(\tau)} - \mathcal{G}_{\neq}^{(\tau)}),
$$

where $\mathcal{G}_{=}^{(\tau)} = \sum_{i \in \mathcal{I}_{=}^{(\tau)}} -\ell'(F(\mathbf{x}_i), y_i)$ and $\mathcal{G}_{\neq}^{(\tau)} = \sum_{i \in \mathcal{I}_{\neq}^{(\tau)}} -\ell'(F(\mathbf{x}_i), y_i)$. The update direction of $\langle \mathbf{w}_c, \boldsymbol{\zeta} \rangle$ depends on the sign of $\mathcal{G}^{(2)} = (\mathcal{G}_{=}^{(2)} - \mathcal{G}_{\neq}^{(2)})$, which is different from the task-specific and general features. Let $h(\mathbf{e}) = \sum_{c \in [C]} \phi\left(\left\langle \mathbf{w}_c^{(2)}, \mathbf{e} \right\rangle\right)$, we rewrite $\ell'$ as

$$
\begin{aligned}
&-\ell'(F(\mathbf{x}_i), y_i) = (1 + \exp(y_i F(\mathbf{x}_i)))^{-1} \\
&= \left(1 + \exp\left(y_i h\left(y_i \mathbf{u}^{(2)}\right) + y_i h\left(y_i \mathbf{v}^{(2)}\right) + y_i h\left(\epsilon_i \boldsymbol{\zeta}^{(2)}\right)\right)\right)^{-1} \\
&= \left(1 + \exp\left(h\left(\mathbf{u}^{(2)}\right) + h\left(\mathbf{v}^{(2)}\right) + y_i \epsilon_i h\left(\boldsymbol{\zeta}^{(2)}\right)\right)\right)^{-1} \\
&= \begin{cases} \left(1 + \exp\left(h\left(\mathbf{u}^{(2)}\right) + h\left(\mathbf{v}^{(2)}\right) + h\left(\boldsymbol{\zeta}^{(2)}\right)\right)\right)^{-1}, & i \in \mathcal{I}_{=}^{(2)}; \\ \left(1 + \exp\left(h\left(\mathbf{u}^{(2)}\right) + h\left(\mathbf{v}^{(2)}\right) - h\left(\boldsymbol{\zeta}^{(2)}\right)\right)\right)^{-1}, & i \in \mathcal{I}_{\neq}^{(2)}. \end{cases}
\end{aligned}
$$

Since $\ell'(\mathbf{z}_1)/\ell'(\mathbf{z}_2) = \Theta(\exp(\mathbf{z}_2 - \mathbf{z}_1))$, we have

$$\begin{aligned}\mathcal{G}_{=}^{(2)}/\mathcal{G}_{\neq}^{(2)} &= \Theta\left(n_{=}^{(2)}n_{\neq}^{(2)^{-1}}\exp\left(-2h\left(\boldsymbol{\zeta}^{(2)}\right)\right)\right)\\&=\Theta\left(n_{=}^{(2)}n_{\neq}^{(2)^{-1}}\exp\left(-2\alpha_{\zeta}^3\alpha_u^{-3}h\left(\mathbf{u}^{(1)}\right)\right)\right).\end{aligned}$$

On the one hand, when $\alpha_\zeta \leq o(\alpha_u)$, we yield that $\max_c \left\langle \mathbf{w}_c^{(2)}, \boldsymbol{\zeta}^{(2)} \right\rangle \leq o(1)$, and $\ell_i' = \ell'(y_i F(\mathbf{x})) = -1/2 \pm o(1)$. In such a case,

$$\left|\mathcal{G}^{(2)}\right| = \Theta\left(|n_= - n_{\neq}|\right) \leq O\left(\sqrt{n}\right),$$

and the update of $\left\langle \mathbf{w}_c^{(2)}, \boldsymbol{\zeta}^{(2)} \right\rangle$ has a relatively slow speed.

On the other hand, for some $\alpha_\zeta \geq \Omega(\alpha_u)$, we have $\max_c \left\langle \mathbf{w}_c^{(2)}, \boldsymbol{\zeta}^{(2)} \right\rangle \geq \Omega(1)$ and $\mathcal{G}_{=}^{(2)}/\mathcal{G}_{\neq}^{(2)}$ can achieve the order of $\Omega(1)$. At the beginning of the second stage, $h\left(\mathbf{u}^{(2)}\right) = \alpha_u^3\alpha_\zeta^{-3}h\left(\boldsymbol{\zeta}^{(1)}\right) \leq o(1)$, $h\left(\mathbf{v}^{(2)}\right) \leq o(1)$, we yield that $-\ell'(F(\mathbf{x}_i), y_i) \leq O(1)$ for $i \in \mathcal{I}_{=}^{(2)}$ while $-\ell'(F(\mathbf{x}_i), y_i) \geq \Omega(1)$ for $i \in \mathcal{I}_{\neq}^{(2)}$. We then have that

$$\mathcal{G}^{(2)} = -\Theta\left(n_{\neq}\right) = -\Theta\left(n\right),$$

and the model forgets $\mathbf{u}^{(1)}$.

The speed of forgetting depends on $\alpha_\zeta$. As $\alpha_\zeta$ increases, $\max_c \left\langle \mathbf{w}_c, \mathbf{u}^{(1)} \right\rangle$ decreases to a low value. Additionally, as the task-specific feature $\mathbf{u}^{(2)}$ is gradually learned by the model, the speed of forgetting also decreases. As a result, after sequentially training on two tasks, the CNN model gains the ability of generalization in two ways:

1. When $\alpha_v \geq \Omega(\alpha_u)$, the model learns the general feature $\mathbf{v}^{(1)} = \mathbf{v}^{(2)}$, and behaves like $F$ with $\mathbf{w}^{\mathbf{rob}}$.

2. When $\alpha_\zeta \leq o(\alpha_u)$, the model learns both $\mathbf{u}^{(1)}$ and $\mathbf{u}^{(2)}$, and behaves like $F$ with $\mathbf{w}^{\mathbf{gen}}$.

CF does not occur in the above two cases. Instead, CF occurs when $\alpha_v < o(\alpha_u)$ and $\alpha_\zeta \geq \Omega(\alpha_u)$. In such a case, the model ends up classifying the data using only feature $\mathbf{u}^{(2)}$, resulting in a high error on the first task. We then show the formal results in Section 5.

### 4.3. Replay-Based Methods Can Suppress CF

When $\alpha_v \leq o(\alpha_u)$ and $\alpha_\zeta \geq \Omega(\alpha_u)$, our analysis implies that CF occurs with a high probability. The remaining question is how to suppress CF. In the second stage, replay-based methods add data points from previous tasks into the training set to suppress CF. We then analyze the effect of replay-based methods.

Given $\widetilde{n}$ data points from the first task, the update rule for $\Delta_c(\mathbf{u}^{(1)})$ and $\Delta_c(\mathbf{u}^{(2)})$ can be rewritten as

$$\begin{aligned}\Delta_c(\mathbf{u}^{(1)}) &= m(\mathbf{u}^{(1)})(\widetilde{\mathcal{G}}_{=}^{(1)} + \widetilde{\mathcal{G}}_{\neq}^{(1)} + \alpha_\zeta\mathcal{G}_{=}^{(2)} - \alpha_\zeta\mathcal{G}_{\neq}^{(2)})\\\Delta_c(\mathbf{u}^{(2)}) &= m(\mathbf{u}^{(2)})(\alpha_\zeta\widetilde{\mathcal{G}}_{=}^{(1)} - \alpha_\zeta\widetilde{\mathcal{G}}_{\neq}^{(1)} + \mathcal{G}_{=}^{(2)} + \mathcal{G}_{\neq}^{(2)}),\end{aligned}$$

where $m(\mathbf{e}) = -\eta(n+\widetilde{n})^{-1}\lambda_c\|\mathbf{e}\|_2^2\phi'(\langle\mathbf{w}_c(t), \mathbf{e}\rangle)$.

Note that $\widetilde{\mathcal{G}}_{=}^{(1)} = \Theta\left(\widetilde{\mathcal{G}}_{\neq}^{(1)}\right) = \Theta(\widetilde{n})$ while $\mathcal{G}_{=}^{(2)} = \Theta\left(\mathcal{G}_{\neq}^{(2)}\right) = \Theta(n)$, and $\alpha_\zeta = \Theta(\alpha_u)$. We can replay at least $\widetilde{n} \geq \Omega(n)$ data points from past tasks to ensure that

$$\widetilde{\mathcal{G}}_{=}^{(1)} + \widetilde{\mathcal{G}}_{\neq}^{(1)} + \alpha_\zeta\mathcal{G}_{=}^{(2)} - \alpha_\zeta\mathcal{G}_{\neq}^{(2)} > 0,$$

which suppresses CF in the second stage. Moreover, we should add $\widetilde{n} \leq O(n)$ to ensure that $\mathbf{u}^{(2)}$ can also be successfully captured by the model. Adding data points from the first task enables the model to correctly classify data points from both tasks.

## 5. Main Results

In this section, we show the theoretical results in this work. The proofs of Lemma 5.1 and Theorem 5.2 are deferred in Appendix D and the proofs of Corollaries 5.3 and 5.5 and Theorem 5.4 can be found in Appendix E.

The following lemma shows that the background noises are not fitted by the network during the learning process.

**Lemma 5.1.** *In the first stage, under Condition 3.6, and given $T \geq \widetilde{\Omega}\left(\frac{1}{\eta\sigma_0\alpha_u^3}\right)$, for any $0 < t \leq T, i \in \mathcal{I}_{tr}^{(1)}, p \in \mathcal{P}_i^\xi$, we have following:*

$$\max_{c\in C}\left\langle\mathbf{w}_c^{(1)}(t+1), \boldsymbol{\xi}_{i,p}^{(1)}\right\rangle - \max_{c\in C}\left\langle\mathbf{w}_c^{(1)}(t), \boldsymbol{\xi}_{i,p}^{(1)}\right\rangle \leq o(\sigma_0\sigma_\xi).$$

*Moreover, we have*

$$\max_{c\in C}\left\langle\mathbf{w}_c^{(1)}(t), \boldsymbol{\xi}_{i,p}^{(1)}\right\rangle \leq \widetilde{O}(\sigma_0\sigma_\xi).$$

In the first stage, we show that when the task-specific feature has a larger norm than the general feature, the model only learns $\mathbf{u}^{(1)}$ at the end of the first stage.

**Theorem 5.2.** *In the first stage, given a training set $\mathcal{S}_{tr}^{(1)}$ with size $n$, if $\alpha_v \leq o(\alpha_u)$, there exists $\widetilde{T}^{(1)} \leq \widetilde{O}\left(\frac{1}{\eta\sigma_0\alpha_u^3}\right)$ such that for any $T^{(1)} \geq \widetilde{T}^{(1)}$, the network $F_{T^{(1)}}$ fits all training data points with a high probability:*

$$\mathbb{P}\left[\forall i \in \mathcal{S}_{tr}^{(1)}, y_i F_{T^{(1)}}^{(1)}(\mathbf{x}_i) \geq \widetilde{\Omega}(1)\right] \geq 1 - O\left(\frac{n^2 P^2 C}{poly(d)}\right).$$

*Moreover, $F_{T^{(1)}}^{(1)}$ achieves a high accuracy on test data points at $T^{(1)}$:*

$$\mathbb{P}_{(\mathbf{x},y)\sim\mathcal{D}_{\mathbf{z}}^{(1)}}\left[yF_{T^{(1)}}^{(1)}(\mathbf{x}) > 0\right] \geq 1 - O\left(\frac{nP^2 C}{poly(d)}\right).$$

We stop training the model at $\widetilde{T}^{(1)}$ in the first stage. At the beginning of the second stage, we initialize $F_0^{(2)} = F_{T_u^{(1)}}^{(1)}$. The parameters are set as $\forall c \in [C], \mathbf{w}_c^{(2)}(0) = \mathbf{w}_c^{(1)}(T_u^{(1)})$. We have the following corollary.

**Corollary 5.3.** *At the beginning of the second task, if $\alpha_u \geq \omega(\alpha_v)$, we have*

$$\max_{c \in [C]} \left\langle \mathbf{w}_c^{(2)}(0), \mathbf{u}^{(2)} \right\rangle = \widetilde{\Theta}(\sigma_0 \alpha_u),$$

$$\max_{c \in [C]} \left\langle \mathbf{w}_c^{(2)}(0), \mathbf{v}^{(2)} \right\rangle = \widetilde{\Theta}(\sigma_0 \alpha_v),$$

$$\max_{c \in [C]} \left\langle \mathbf{w}_c^{(2)}(0), \mathbf{u}^{(1)} \right\rangle \geq \widetilde{\Omega}(1).$$

In the second stage, the following theorem shows that if the $\zeta$ has a significant norm, CF occurs, and the model cannot achieve a high accuracy in the first task again.

**Theorem 5.4.** *In the second stage, given a training set $\mathcal{S}_{tr}^{(2)}$ with size $n$, there exists $\widetilde{T}^{(2)} = \widetilde{\Theta}\left(\frac{1}{\eta \sigma_0 \alpha_u^3}\right)$ such that for $T^{(2)} \geq \widetilde{T}^{(2)}$, the network $F_{T^{(2)}}$ fits all training data points with a high probability:*

$$\mathbb{P}\left[\forall i \in \mathcal{S}_{tr}^{(2)}, y_i F_{T^{(2)}}^{(2)}(\mathbf{x}_i) \geq \widetilde{\Omega}(1)\right] \geq 1 - O\left(\frac{n^2 P^2 C}{poly(d)} + \frac{1}{poly(n)}\right).$$

*Moreover, $F_{T^{(2)}}$ achieves a high accuracy on test data sampled from the second task:*

$$\mathbb{P}_{(\mathbf{x},y) \sim \mathcal{D}_{\mathbf{z}}^{(2)}}\left[y F_{T^{(2)}}^{(2)}(\mathbf{x}) > 0\right] \geq 1 - O\left(\frac{n P^2 C}{poly(d)} + \frac{1}{poly(n)}\right).$$

*If $\alpha_v \leq o(\alpha_u)$, and $\alpha_\zeta \geq \Omega(\alpha_u)$, $F_{T^{(2)}}$ achieves a low accuracy on test data sampled from the first task*

$$\mathbb{P}_{(\mathbf{x},y) \sim \mathcal{D}_{\mathbf{z}}^{(1)}}\left[y F_{T^{(2)}}^{(2)}(\mathbf{x}) > 0\right] \leq \frac{1}{2} + O\left(\frac{n P^2 C}{poly(d)} + \frac{1}{poly(n)}\right).$$

We stop to train the model at $T^{(2)} = \widetilde{T}^{(2)}$ in the second stage. The following corollary demonstrates that the model forgets the task-specific feature of the previous task.

**Corollary 5.5.** *At the end of the second task, for $\widetilde{T}^{(2)} = \widetilde{\Theta}\left(\frac{1}{\eta \sigma_0 \alpha_u^3}\right)$, if $\alpha_u \geq \omega(\alpha_v)$ and $\alpha_\zeta \geq \Omega(\alpha_u)$, we have*

$$\max_{c \in [C]} \left\langle \mathbf{w}_c^{(2)}(\widetilde{T}^{(2)}), \mathbf{u}^{(1)} \right\rangle \leq o(1),$$

$$\max_{c \in [C]} \left\langle \mathbf{w}_c^{(2)}(\widetilde{T}^{(2)}), \mathbf{v}^{(2)} \right\rangle = \widetilde{\Theta}(\sigma_0 \alpha_v),$$

$$\max_{c \in [C]} \left\langle \mathbf{w}_c^{(2)}(\widetilde{T}^{(2)}), \mathbf{u}^{(2)} \right\rangle \geq \widetilde{\Omega}(1).$$

Through our theoretical analysis on the entire learning process in CL, we show that CF occurs during the learning process due to the following reasons:

1. The general and robust feature $\mathbf{v}^{(1)}$ has a low signal, while the task-specific feature $\mathbf{u}^{(1)}$ has a relatively large signal. As shown in Corollary 5.3, if $\alpha_v \leq o(\alpha_u)$, after training in the first stage, only $\mathbf{u}^{(1)}$ is learned by the model.

2. The task-specific feature from the first task manifests as a random feature with a strong signal in the second task. If $\alpha_\zeta \geq \Omega(\alpha_u)$, Corollary 5.5 shows that the model forgets $\mathbf{u}^{(1)}$ while learns $\mathbf{u}^{(2)}$ in the second stage.

*Remark* 5.6. Based on our analysis, CF can be mitigated by violating the two conditions identified above, which is empirically supported by simulated data in Figure 2 of Section 6. Furthermore, our findings indicate that CNNs trained with (S)GD tend to learn the features with the strongest signal rather than the most robust ones. When the robust feature has a weak signal, the CNNs fail to capture the robust feature during training, which highlights the necessity of developing robust training algorithms that encourage the learning of robust features to prevent CF. In addition, by fixing $\alpha_u$ and $\alpha_\zeta$ and increasing $\alpha_v$, the similarity between tasks increases, and our results suggest that CF is less likely to occur when learning on such similar tasks. We hope that our findings will inspire future research to investigate CF from the perspective of feature learning.

## 6. Experiments

In this section, we conduct experiments on both simulated and real-world datasets to validate our findings.

### 6.1. Numerical Analysis on Simulated Dataset

We conduct a numerical analysis on simulated data, drawn from our data model, and use the CNN model defined in Equation (1). To analyze the feature extraction capabilities of the model, we fix the last layer of the CNN.

To investigate which features the model captures during the two stages of training, we generate images containing only $\mathbf{u}$, $\mathbf{v}$, and $\zeta$. We evaluate the model at the end of both the first and second stages, and the results are shown in Figure 1. We set $\alpha_u = 1$, $\alpha_v = 0.1$, and $\alpha_\zeta = 0.9$. In the first stage, as shown in the first column, we find that only $\mathbf{u}^{(1)}$ is captured by the model using two channels, while the learning of $\mathbf{v}^{(1)}$ and $\zeta^{(1)}$ progresses slowly, which aligns with Corollary 5.3. In the second stage, as shown in the last two columns of the first row, we observe that $\mathbf{u}^{(1)}$ is forgotten by the model, and $\mathbf{u}^{(2)}$ dominates the model's output, which aligns with Corollary 5.5. Moreover, the images in the last row of Figure 1 illustrate the model's output with different inputs. We find that the output of the model is dominated by $\mathbf{u}^{(1)}$ in the first stage, while dominated by $\mathbf{u}^{(2)}$ in the second stage.

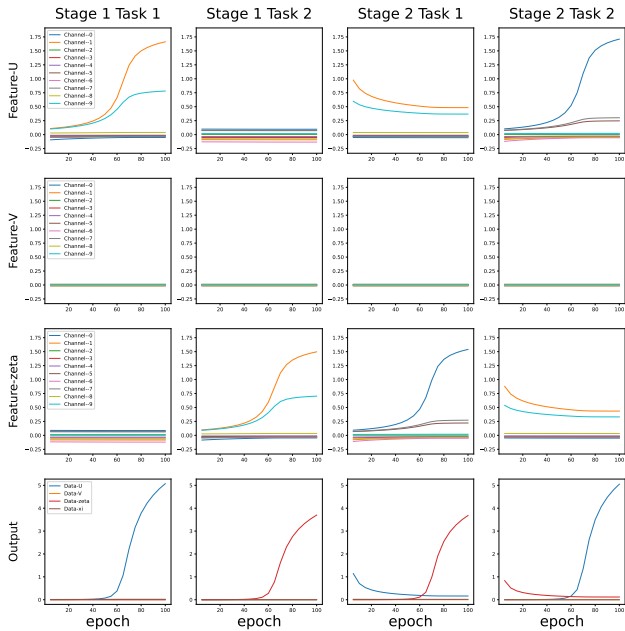

Figure 1. The output of each layer in the CNN model using different inputs. **Top three rows**: The output of each channel in the first layer. **Bottom row**: The output of the last layer.

We then set the norm of **u** to $\alpha_u = 1$, and vary $\alpha_v$ and $\alpha_\zeta$ within the range of $[0, 1]$ to study the condition under which CF occurs. In the first row of Figure 2, the results in the first column demonstrate that at the end of the first stage, the model correctly classifies the data points drawn from the first task. At the end of the second stage, as shown in the last column, the model also correctly classifies the data points drawn from the second task. However, the results in the third column show that the value of $\alpha_v$ and $\alpha_\zeta$ affects the performance of the model on the first task in the second stage. For a small $\alpha_v$, as $\alpha_\zeta$ increases, the model struggles to achieve high accuracy on both old and new tasks. We also observe that as the norm of the general feature $\mathbf{v}^{(2)}$ increases, CF is suppressed even when $\alpha_\zeta$ is large. The findings support Corollary 5.3. Additionally, in the second column, we observe that when $\alpha_v = \alpha_u = 1$, the model successfully extracts the general feature, achieving high accuracy on the second task without needing to train on the second task's data points.

We further investigate the effect of the replay-based method. We fix $\alpha_u = 1, \alpha_v = 0.5, \alpha_\zeta = 0.8$ and show the results in the second row of Figure 2. The third column in the last row shows the performance of the model on the first task at the end of the second stage. The result demonstrates that when no additional data is used, CF occurs in the second stage. However, by incorporating data points from previous tasks into the training set, CF is suppressed. These results further validate our theoretical insights, as shown in Section 4.3.

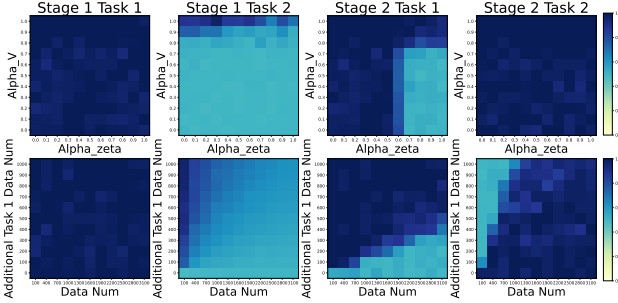

Figure 2. The effect of $\alpha_v$ and $\alpha_\zeta$ on the model's performance. **First Row**: Performance of the model in a standard CL setup. **Second Row**: Performance of the model when additional samples from the previous task are added in the second stage to suppress CF. The X-axis represents the number of data points $n$, and the Y-axis represents the number of additional data points from Task-1.

## 6.2. Experiments on Real-World Datasets

In real-world datasets, it is challenging to decompose images into distinct features. Therefore, we focus on studying the features extracted by the model in the representation space. We conduct experiments on CIFAR-10, CIFAR-100 (Krizhevsky et al., 2009), and Tiny-ImageNet (Deng et al., 2009) to evaluate the model's performance on real-world datasets. For each dataset, we split the data into $K/2$ binary tasks, where $K$ is the number of classes. We then sequentially select a pair of tasks, and train the model on the pair of tasks.

Our first empirical observation is that, in practice, the model is easier to extract non-robust features that do not generalize to unseen tasks in most cases, as shown in Figure 3. This implies that for two independently sampled tasks, achieving a shared general feature is difficult. In CIFAR-10, for example, when the model is trained on Task-0, the model successfully extracts the general feature which can be used in Task-4. However, in most cases, the model tends to extract task-specific features for each task. Therefore, this empirical result aligns with the condition $\alpha_v \leq o(\alpha_u)$.

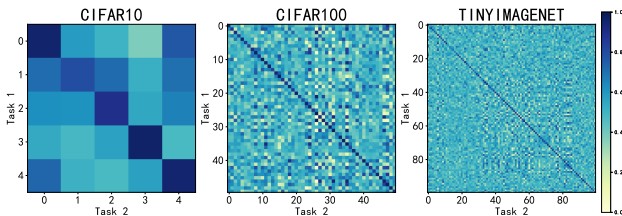

Figure 3. Overview of CF in different datasets: CIFAR-10 (Left), CIFAR-100 (Middle), and Tiny-ImageNet (Right). We record the model's performance on the second task at the end of the first stage. A deeper color means a better performance of the model.

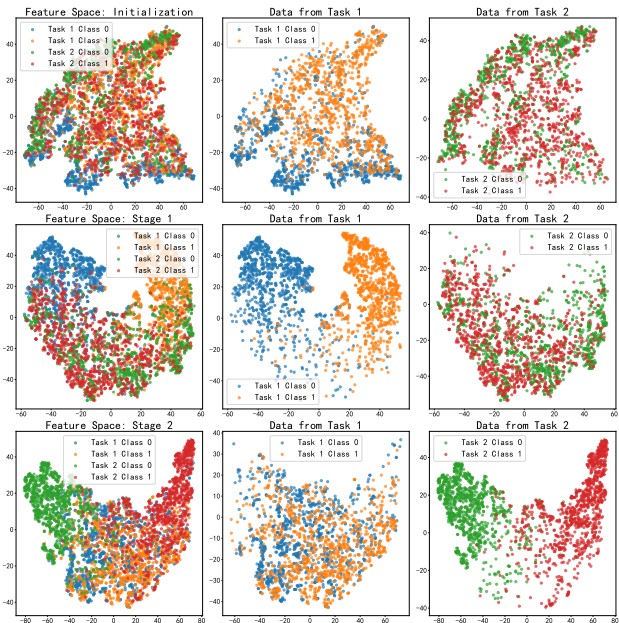

*Figure 4.* Using T-SNE to visualize the feature space at different stages of CL when sequentially train the model on the Task-0 and Task-3 in CIFAR-10. **First row**. The model is randomly initialized. **Second row**. At the end of the first stage. **Third row**. At the end of the second stage.

We then use T-SNE (van der Maaten & Hinton, 2008) to visualize the feature in both the first and second stages of CL. As shown in the second row of Figure 4, when using the same feature extractor, the features from the first task are clustered, while the features from the second task are not. This suggests that the features extracted in the first stage are task-specific. As features from task 1 and task 2 exhibit significant overlap, which aligns the condition $\alpha_\zeta \geq \Omega(\alpha_u)$. Additionally, we calculate the maximal singular vector in the feature space. The result, shown in the last column of Figure 5, demonstrates that in the direction of the maximal singular vector, the features from the first task have a weak signal in the second stage, implying that the model forgets the learned features.

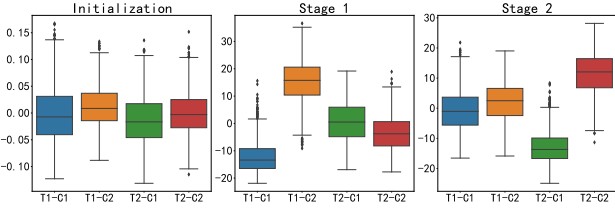

*Figure 5.* Inner product of the features with the maximal singular vector at different stages of CL. The label T$a$-C$b$ indicates that the data drawn from class-$b$ in $a^{\mathbf{th}}$ task. We choose Task-0 and Task-3 in CIFAR-10 as the first and second task.

## 7. Conclusion

In this work, we theoretically analyze the condition that CF occurs in a two-layer CNN model using a multi-view data model. We consider a task-incremental CL scenario. Our theoretical results demonstrate that, in a pair of tasks, if the general feature is either absent or has a low signal, the model will learn the task-specific feature during the first stage. Moreover, if the task-specific feature has a large norm in the second task, CF manifests in the second stage. We also provide theoretical insights into the effectiveness of replay-based methods. Finally, experiments on both simulated and real-world datasets validate our findings.

## Impact Statement

This paper presents work whose goal is to advance the field of Machine Learning. There are many potential societal consequences of our work, none which we feel must be specifically highlighted here.

## Acknowledgments

We would like to thank Xin Zou for helpful suggestions. This work is supported by the Key R&D Program of Hubei Province under Grant 2024BAB038, the National Key R&D Program of China under Grant 2023YFC3604702, the Fundamental Research Funds for the Central Universities under Grant 2042025kf0045.

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

## A. Useful Lemmas

We first show the important lemmas, which are useful in our proof.

**Lemma A.1** (Lemma 1 in Laurent & Massart (2000)). *Suppose $X_i \ldots, X_n$ are $n$ i.i.d. Gaussian random variables with mean 0 and variance 1. Let $a_1, \ldots, a_n$ be non-negative. We set*

$$|a|_\infty = \sup_{i=1,\ldots,n} |a_i| , \ |a|_2^2 = \sum_{i=1}^n a_i^2.$$

*Let*

$$Z = \sum_{i=1}^n a_i(X^2 - 1).$$

*Then, the following inequalities hold for any positive $t$:*

$$\mathbb{P}\left[ Z \geq 2|a|_2 \sqrt{t} + 2|a|_\infty t \right] \leq \exp(-t).$$

$$\mathbb{P}\left[ Z \leq -2|a|_2 \sqrt{t} \right] \leq \exp(-t).$$

**Lemma A.2** (Lemma 4 in Shen et al. (2022)). *Consider independently sampled Gaussian vectors $\mathbf{z}_1 \sim \mathcal{N}(0, \sigma_1^2 \mathbf{I}_d)$ and $\mathbf{z}_2 \sim \mathcal{N}(0, \sigma_2^2 \mathbf{I}_d)$. For any $\delta \in (0, 1)$ and a large enough $d$, there exists constants $c_1, c_2$ such that*

$$\mathbb{P}\left[ |\langle \mathbf{z}_1, \mathbf{z}_2 \rangle| \leq c_1 \sigma_1 \sigma_2 \sqrt{d \log(2/\delta)} \right] \geq 1 - \delta,$$

$$\mathbb{P}\left[ \langle \mathbf{z}_1, \mathbf{z}_2 \rangle \geq c_2 \sigma_1 \sigma_2 \sqrt{d} \right] \geq 1/4.$$

**Lemma A.3** (Proposition 2.5 in Wainwright (2019)). *Suppose that the variables $X_i, i = 1, \ldots, n$, are independent, and $X_i$ has mean $\mu_i$ and sub-Gaussian parameter $\sigma_i$. Then for all $r \geq 0$, we have*

$$\mathbb{P}\left[ \sum_{i=1}^n (X_i - \mu_i) \geq t \right] \leq \exp\left\{ -\frac{t^2}{2\sum_{i=1}^n \sigma_i^2} \right\}$$

**Lemma A.4** (Lemma A.1 in Zou & Liu (2023a)). *If $\mathbf{X} \sim \mathcal{N}(\boldsymbol{\mu}, \Sigma)$ where $X \in \mathbb{R}^n$, then for any $A \in \mathbb{R}^{m \times n}$, we have:*

$$A\mathbf{X} \sim \mathcal{N}(A\boldsymbol{\mu}, A\Sigma A^T). \tag{2}$$

**Lemma A.5** (Fact A.2. in Frei et al. (2022)). *Let $g(z) = \ell'(z) = -1/(1 + \exp(z))$, for any $z_1, z_2 \in R$, we have*

$$\frac{g(z_1)}{g(z_2)} \leq \max\left( 2, 2\frac{\exp(-z_1)}{\exp(-z_2)} \right).$$

The following lemma shows a lower bound for $\frac{g(z_1)}{g(z_2)}$, which is inspired by Lemma C.6 in Kou et al. (2023).

**Lemma A.6.** *Let $g(z) = \ell'(z) = -1/(1 + \exp(z))$. Let $c_1, c_2 \in \mathbb{R}$ be two constants and suppose $c_1 \leq z_1 \leq c_2$, $z_2 \in \mathbb{R}$. Then, there exists a constant $c' > 0$ such that*

$$\frac{g(z_1)}{g(z_2)} \geq c' \exp(z_2 - z_1).$$

*Proof.* We begin by rewriting the ratio as

$$\frac{g(z_1)}{g(z_2)} = \frac{1 + \exp(z_2)}{1 + \exp(z_1)} = 1 + \frac{\exp(z_2) - \exp(z_1)}{1 + \exp(z_1)} = 1 + \frac{\exp(z_2 - z_1) - 1}{1 + \exp(-z_1)}.$$

We first consider the case where $z_2 \geq z_1$. Since $\exp(z_2 - z_1) \geq 1$, we obtain

$$\frac{g(z_1)}{g(z_2)} \geq 1 + \frac{\exp(z_2 - z_1) - 1}{1 + \exp(-c_1)} = \frac{\exp(z_2 - z_1) + \exp(-c_1)}{1 + \exp(-c_1)} \geq c' \exp(z_2 - z_1),$$

where $c' = \frac{1}{1+\exp(-c_1)}$.

Next, consider the case where $z_2 < z_1$, so that $\exp(z_2 - z_1) < 1$. We have

$$\frac{g(z_1)}{g(z_2)} \geq 1 + \frac{\exp(z_2 - z_1) - 1}{1 + \exp(-c_2)} = \frac{\exp(z_2 - z_1) + \exp(-c_2)}{1 + \exp(-c_2)} \geq c' \exp(z_2 - z_1),$$

where $c' = \frac{1}{1+\exp(-c_2)}$.

Combining both cases, the desired inequality holds for all $z_1 \in [c_1, c_2]$ and $z_2 \in \mathbb{R}$, which completes the proof. $\square$

**Proposition A.7.** *Given a standard Gaussian variable $Z \sim \mathcal{N}(0, 1)$, then we have $\mathbb{P}[Z \geq 1/2] \geq 1/4$.*

**Lemma A.8.** *Given a training dataset $\mathcal{S}^{(\tau)}$ containing data drawn from the data model defined in Definition 3.1. Define the index sets*

$$\mathcal{I}_{=}^{(\tau)} := \left\{ i : y_i^{(\tau)} = \epsilon_i \right\}, \quad \mathcal{I}_{\neq}^{(\tau)} := \left\{ i : y_i^{(\tau)} \neq \epsilon_i \right\}.$$

*Then, with probability at least $1 - O\left(\frac{1}{\text{poly}(n)}\right)$, we have*

$$\forall \mathcal{I} \in \left\{ \mathcal{I}_{=}^{(\tau)}, \mathcal{I}_{\neq}^{(\tau)} \right\}, \quad \left| |\mathcal{I}| - \frac{n}{2} \right| \leq \widetilde{O}\left( n^{1/2} \right). \tag{3}$$

*and*

$$\max \left\{ \frac{\left| \mathcal{I}_{\neq}^{(\tau)} \right|}{\left| \mathcal{I}_{=}^{(\tau)} \right|}, \frac{\left| \mathcal{I}_{=}^{(\tau)} \right|}{\left| \mathcal{I}_{\neq}^{(\tau)} \right|} \right\} \leq 1 + \widetilde{O}\left( n^{-1/2} \right). \tag{4}$$

*Proof.* By Lemma A.3, with probability at least $1 - \delta$, we have the following concentration bound for both subsets:

$$\forall \mathcal{I} \in \left\{ \mathcal{I}_{=}^{(\tau)}, \mathcal{I}_{\neq}^{(\tau)} \right\}, \quad \left| |\mathcal{I}| - \frac{n}{2} \right| \leq \sqrt{2n \log(2/\delta)}.$$

Setting $\delta \leq O\left(\frac{1}{\text{poly}(n)}\right)$, we have Equation (3).

Let $\Delta := \sqrt{2n \log(2/\delta)}$. We analyze the size ratio through the following steps:

$$\begin{aligned}
\max \left\{ \frac{|\mathcal{I}_{\neq}|}{|\mathcal{I}_{=}|}, \frac{|\mathcal{I}_{=}|}{|\mathcal{I}_{\neq}|} \right\} &\leq \max_{\mathcal{I} \in \{\mathcal{I}_{=}, \mathcal{I}_{\neq}\}} \frac{\frac{n}{2} + \Delta}{\frac{n}{2}} \cdot \frac{\frac{n}{2}}{\frac{n}{2} - \Delta} \\
&= \max_{\mathcal{I} \in \{\mathcal{I}_{=}, \mathcal{I}_{\neq}\}} \left( 1 + \frac{2\Delta}{n} \right) \cdot \left( 1 - \frac{2\Delta}{n} \right)^{-1} \\
&\leq 1 + \frac{4\Delta/n}{1 - 2\Delta/n} \\
&= 1 + \frac{4\sqrt{2 \log(2/\delta)}}{\sqrt{n} \left( 1 - 2\sqrt{2 \log(2/\delta)}/\sqrt{n} \right)}
\end{aligned}$$

Setting $\delta = n^{-a}$ for constant $a > 0$, we have $\sqrt{\log(2/\delta)} = \sqrt{a \log n + \log 2} = \widetilde{O}(1)$. Thus:

$$\frac{4\sqrt{2 \log(2/\delta)}}{\sqrt{n}} = \widetilde{O}\left( n^{-1/2} \right), \quad 1 - 2\sqrt{2 \log(2/\delta)}/\sqrt{n} = 1 - o(1).$$

Therefore, the ratio bound simplifies to $1 + \widetilde{O}(n^{-1/2})$. By a union bound over both subsets, the total failure probability is $2\delta = O(n^{-a}) = O(1/\text{poly}(n))$ $\square$

## B. Update Rule for Each Components

Given a learning rate $\eta$, the parameters are optimized as

$$\mathbf{w}_c(t+1) = \mathbf{w}_c(t) - \frac{\eta\lambda_c}{n}\sum_{i=1}^{n}\nabla\ell(F(\mathbf{x}_i), y_i)$$

$$= \mathbf{w}_c(t) - \frac{\eta\lambda_c}{n}\sum_{i=1}^{n}\sum_{p=1}^{P}y_i\ell'(F(\mathbf{x}_i), y_i)\phi'\left(\langle\mathbf{w}_c(t), \mathbf{x}_{i,p}\rangle\right)\mathbf{x}_{i,p}$$

$$= \mathbf{w}_c(t) - \frac{\eta\lambda_c}{n}\sum_{i=1}^{n}\ell'(F(\mathbf{x}_i), y_i)\phi'\left(\langle\mathbf{w}_c(t), y_i\mathbf{u}\rangle\right)\mathbf{u} - \frac{\eta\lambda_c}{n}\sum_{i=1}^{n}\ell'(F(\mathbf{x}_i), y_i)\phi'\left(\langle\mathbf{w}_c(t), y_i\mathbf{v}\rangle\right)\mathbf{v}$$

$$- \frac{\eta\lambda_c}{n}\sum_{i=1}^{n}y_i\epsilon_i\ell'(F(\mathbf{x}_i), y_i)\phi'\left(\langle\mathbf{w}_c(t), \epsilon_i\boldsymbol{\zeta}\rangle\right)\boldsymbol{\zeta} - \frac{\eta\lambda_c}{n}\sum_{i=1}^{n}\sum_{p\in\mathcal{P}_i^\xi}y_i\ell'(F(\mathbf{x}_i), y_i)\phi'\left(\langle\mathbf{w}_c(t), \boldsymbol{\xi}_{i,p}\rangle\right)\boldsymbol{\xi}_{i,p}$$

$$(5)$$

Equation (5) due to the decomposition of $\mathbf{x}$. We then analyze the update of each component.

**Lemma B.1.** *[Task-Specific and General Features]. In stage $\tau$, Let $\Delta_c\left(\mathbf{u}^{(\tau)}\right) = \left\langle\mathbf{w}_c^{(\tau)}(t+1), \mathbf{u}^{(\tau)}\right\rangle - \left\langle\mathbf{w}_c^{(\tau)}(t), \mathbf{u}^{(\tau)}\right\rangle$, and $\Delta_c\left(\mathbf{v}^{(\tau)}\right) = \left\langle\mathbf{w}_c^{(\tau)}(t+1), \mathbf{v}^{(\tau)}\right\rangle - \left\langle\mathbf{w}_c^{(\tau)}(t), \mathbf{v}^{(\tau)}\right\rangle$ we have*

$$\Delta_c\left(\mathbf{u}^{(\tau)}\right) = -\frac{\eta\lambda_c}{n}\sum_{i=1}^{n}\ell'(F(\mathbf{x}_i^{(\tau)}), y_i^{(\tau)})\phi'\left(\left\langle\mathbf{w}_c^{(\tau)}(t), \mathbf{u}^{(\tau)}\right\rangle\right)\left\|\mathbf{u}^{(\tau)}\right\|_2^2$$

$$\Delta_c\left(\mathbf{v}^{(\tau)}\right) = -\frac{\eta\lambda_c}{n}\sum_{i=1}^{n}\ell'(F(\mathbf{x}_i^{(\tau)}), y_i^{(\tau)})\phi'\left(\left\langle\mathbf{w}_c^{(\tau)}(t), \mathbf{v}^{(\tau)}\right\rangle\right)\left\|\mathbf{v}^{(\tau)}\right\|_2^2$$

*Proof.* We can rewrite $\Delta_c\left(\mathbf{u}^{(\tau)}\right)$ as

$$\Delta_c\left(\mathbf{u}^{(\tau)}\right) = \left\langle\mathbf{w}_c^{(\tau)}(t+1), \mathbf{u}^{(\tau)}\right\rangle - \left\langle\mathbf{w}_c^{(\tau)}(t), \mathbf{u}^{(\tau)}\right\rangle$$

$$= -\frac{\eta\lambda_c}{n}\sum_{i=1}^{n}y_iy_i\ell'(F(\mathbf{x}_i^{(\tau)}), y_i^{(\tau)})\phi'\left(\left\langle\mathbf{w}_c^{(\tau)}(t), y_i^{(\tau)}\mathbf{u}^{(\tau)}\right\rangle\right)\left\langle\mathbf{u}^{(\tau)}, \mathbf{u}^{(\tau)}\right\rangle$$

$$- \frac{\eta}{n}\sum_{i=1}^{n}\ell'(F(\mathbf{x}_i^{(\tau)}), y_i^{(\tau)})\phi'\left(\left\langle\mathbf{w}_c^{(\tau)}(t), y_i^{(\tau)}\mathbf{v}^{(\tau)}\right\rangle\right)\left\langle\mathbf{v}^{(\tau)}, \mathbf{u}^{(\tau)}\right\rangle$$

$$- \frac{\eta\lambda_c}{n}\sum_{i=1}^{n}y_i^{(\tau)}\epsilon_i\ell'(F(\mathbf{x}_i^{(\tau)}), y_i^{(\tau)})\phi'\left(\left\langle\mathbf{w}_c^{(\tau)}(t), \epsilon_i\boldsymbol{\zeta}^{(\tau)}\right\rangle\right)\left\langle\boldsymbol{\zeta}^{(\tau)}, \mathbf{u}^{(\tau)}\right\rangle$$

$$- \frac{\eta\lambda_c}{n}\sum_{i=1}^{n}\sum_{p\in\mathcal{P}_i^\xi}y_i^{(\tau)}\ell'(F(\mathbf{x}_i^{(\tau)}), y_i^{(\tau)})\phi'\left(\left\langle\mathbf{w}_c^{(\tau)}(t), \boldsymbol{\xi}_{i,p}^{(\tau)}\right\rangle\right)\left\langle\boldsymbol{\xi}_{i,p}^{(\tau)}, \mathbf{u}^{(\tau)}\right\rangle$$

$$= -\frac{\eta\lambda_c}{n}\sum_{i=1}^{n}\ell'\left(F(\mathbf{x}_i^{(\tau)}), y_i^{(\tau)}\right)\phi'\left(\left\langle\mathbf{w}_c^{(\tau)}(t), y_i^{(\tau)}\mathbf{u}^{(\tau)}\right\rangle\right)\left\|\mathbf{u}^{(\tau)}\right\|_2^2$$

Similarly, we have

$$
\begin{aligned}
\Delta_c\left(\mathbf{v}^{(\tau)}\right) &= \left\langle \mathbf{w}_c^{(\tau)}(t+1), \mathbf{v}^{(\tau)} \right\rangle - \left\langle \mathbf{w}_c^{(\tau)}(t), \mathbf{v}^{(\tau)} \right\rangle \\
&= -\frac{\eta\lambda_c}{n}\sum_{i=1}^n y_i y_i \ell'(F(\mathbf{x}_i^{(\tau)}), y_i^{(\tau)})\phi'\left(\left\langle \mathbf{w}_c^{(\tau)}(t), y_i^{(\tau)}\mathbf{u}^{(\tau)}\right\rangle\right)\left\langle \mathbf{u}^{(\tau)}, \mathbf{v}^{(\tau)}\right\rangle \\
&\quad -\frac{\eta}{n}\sum_{i=1}^n \ell'(F(\mathbf{x}_i^{(\tau)}), y_i^{(\tau)})\phi'\left(\left\langle \mathbf{w}_c^{(\tau)}(t), y_i^{(\tau)}\mathbf{v}^{(\tau)}\right\rangle\right)\left\langle \mathbf{v}^{(\tau)}, \mathbf{v}^{(\tau)}\right\rangle \\
&\quad -\frac{\eta\lambda_c}{n}\sum_{i=1}^n y_i^{(\tau)}\epsilon_i \ell'(F(\mathbf{x}_i^{(\tau)}), y_i^{(\tau)})\phi'\left(\left\langle \mathbf{w}_c^{(\tau)}(t), \epsilon_i\boldsymbol{\zeta}^{(\tau)}\right\rangle\right)\left\langle \boldsymbol{\zeta}^{(\tau)}, \mathbf{v}^{(\tau)}\right\rangle \\
&\quad -\frac{\eta\lambda_c}{n}\sum_{i=1}^n \sum_{p\in\mathcal{P}_i^\xi} y_i^{(\tau)}\ell'(F(\mathbf{x}_i^{(\tau)}), y_i^{(\tau)})\phi'\left(\left\langle \mathbf{w}_c^{(\tau)}(t), \boldsymbol{\xi}_{i,p}^{(\tau)}\right\rangle\right)\left\langle \boldsymbol{\xi}_{i,p}^{(\tau)}, \mathbf{v}^{(\tau)}\right\rangle \\
&= -\frac{\eta\lambda_c}{n}\sum_{i=1}^n \ell'(F(\mathbf{x}_i^{(\tau)}), y_i^{(\tau)})\phi'\left(\left\langle \mathbf{w}_c^{(\tau)}(t), y_i^{(\tau)}\mathbf{v}^{(\tau)}\right\rangle\right)\left\| \mathbf{v}^{(\tau)}\right\|_2^2
\end{aligned}
$$

We conclude our proof. $\qquad\square$

Note that $\phi' \geq 0, \ell' < 0$, $\Delta_c\left(\mathbf{u}\right)$ is increasing because $\lambda_c = 1$.

**Lemma B.2.** *[Random Feature]. In any stage $\tau$, let $\Delta_c\left(\boldsymbol{\zeta}^{(\tau)}\right) = \left\langle \mathbf{w}_c^{(\tau)}(t+1), \boldsymbol{\zeta}^{(\tau)}\right\rangle - \left\langle \mathbf{w}_c^{(\tau)}(t), \boldsymbol{\zeta}^{(\tau)}\right\rangle$, if $\phi'(\cdot)$ is an even function, we have*

$$
\Delta_c\left(\boldsymbol{\zeta}^{(\tau)}\right) = \frac{\eta\lambda_c\left\|\boldsymbol{\zeta}^{(\tau)}\right\|_2^2 \phi'\left(\left\langle \mathbf{w}_c^{(\tau)}(t), \boldsymbol{\zeta}^{(\tau)}\right\rangle\right)}{n}\mathcal{G}^{(\tau)},
$$

*where $\mathcal{G}^{(\tau)} = \sum_{i:y_i=\epsilon_i}\left(-\ell'(F\left(\mathbf{x}_i^{(\tau)}\right), y_i^{(\tau)})\right) - \sum_{i:y_i=-\epsilon_i}\left(-\ell'(F\left(\mathbf{x}_i^{(\tau)}\right), y_i^{(\tau)})\right).$*

*Proof.* We ignore $\tau$ in expression, and rewrite $\Delta_c\left(\boldsymbol{\zeta}\right)$ as

$$
\begin{aligned}
\Delta_c\left(\boldsymbol{\zeta}^{(\tau)}\right) &= \left\langle \mathbf{w}_c^{(\tau)}(t+1), \boldsymbol{\zeta}^{(\tau)}\right\rangle - \left\langle \mathbf{w}_c^{(\tau)}(t), \boldsymbol{\zeta}^{(\tau)}\right\rangle \\
&= -\frac{\eta\lambda_c}{n}\sum_{i=1}^n y_i^{(\tau)}y_i^{(\tau)}\ell'(F(\mathbf{x}_i^{(\tau)}), y_i^{(\tau)})\phi'\left(\left\langle \mathbf{w}_c^{(\tau)}(t), y_i^{(\tau)}\mathbf{u}^{(\tau)}\right\rangle\right)\left\langle \mathbf{u}^{(\tau)}, \boldsymbol{\zeta}^{(\tau)}\right\rangle \\
&\quad -\frac{\eta\lambda_c}{n}\sum_{i=1}^n y_i^{(\tau)}\epsilon_i \ell'(F(\mathbf{x}_i^{(\tau)}), y_i^{(\tau)})\phi'\left(\left\langle \mathbf{w}_c^{(\tau)}(t), \boldsymbol{\zeta}^{(\tau)}\right\rangle\right)\left\langle \boldsymbol{\zeta}^{(\tau)}, \boldsymbol{\zeta}^{(\tau)}\right\rangle \\
&\quad -\frac{\eta}{n}\sum_{i=1}^n y_i^{(\tau)}y_i^{(\tau)}\ell'(F(\mathbf{x}_i^{(\tau)}), y_i^{(\tau)})\phi'\left(\left\langle \mathbf{w}_c^{(\tau)}(t), y_i^{(\tau)}\mathbf{v}^{(\tau)}\right\rangle\right)\left\langle \mathbf{v}^{(\tau)}, \boldsymbol{\zeta}^{(\tau)}\right\rangle \\
&\quad -\frac{\eta\lambda_c}{n}\sum_{i=1}^n \sum_{p\in\mathcal{P}_i^\xi} y_i^{(\tau)}\ell'(F(\mathbf{x}_i^{(\tau)}), y_i^{(\tau)})\phi'\left(\left\langle \mathbf{w}_c^{(\tau)}(t), \boldsymbol{\xi}_{i,p}^{(\tau)}\right\rangle\right)\left\langle \boldsymbol{\xi}_{i,p}^{(\tau)}, \boldsymbol{\zeta}^{(\tau)}\right\rangle \\
&= -\frac{\eta\lambda_c}{n}\sum_{i=1}^n y_i^{(\tau)}\epsilon_i^{(\tau)}\ell'(F(\mathbf{x}_i^{(\tau)}), y_i^{(\tau)})\phi'\left(\left\langle \mathbf{w}_c^{(\tau)}(t), \epsilon_i\boldsymbol{\zeta}^{(\tau)}\right\rangle\right)\left\langle \boldsymbol{\zeta}^{(\tau)}, \boldsymbol{\zeta}^{(\tau)}\right\rangle \\
&= \frac{\eta\lambda_c\left\|\boldsymbol{\zeta}^{(\tau)}\right\|_2^2 \phi'\left(\left\langle \mathbf{w}_c^{(\tau)}(t), \boldsymbol{\zeta}^{(\tau)}\right\rangle\right)}{n}\mathcal{G}^{(\tau)},
\end{aligned}
$$

where $\mathcal{G}^{(\tau)} = \sum_{i:y_i=\epsilon_i}\left(-\ell'(F\left(\mathbf{x}_i^{(\tau)}\right), y_i^{(\tau)})\right) - \sum_{i:y_i=-\epsilon_i}\left(-\ell'(F\left(\mathbf{x}_i^{(\tau)}\right), y_i^{(\tau)})\right).$ We conclude our proof. $\qquad\square$

**Lemma B.3** (Background Noises). *We use $\Delta(\boldsymbol{\xi}_{i',p'}^{(\tau)})$ to denote $\left\langle \mathbf{w}_c^{(\tau)}(t+1), \boldsymbol{\xi}_{i',p'}^{(\tau)} \right\rangle - \left\langle \mathbf{w}_c^{(\tau)}(t), \boldsymbol{\xi}_{i',p'}^{(\tau)} \right\rangle$, and have*

$$\Delta(\boldsymbol{\xi}_{i',p'}^{(\tau)}) = -\frac{\eta\lambda_c}{n} \sum_{i=1}^{n} \sum_{p \in \mathcal{P}_i^{\xi}} y_i^{(\tau)} \ell'(F^{(\tau)}(\mathbf{x}_i^{(\tau)}), y_i^{(\tau)}) \phi'\left(\left\langle \mathbf{w}_c^{(\tau)}(t), \boldsymbol{\xi}_{i,p}^{(\tau)} \right\rangle\right) \left\langle \boldsymbol{\xi}_{i,p}^{(\tau)}, \boldsymbol{\xi}_{i',p'}^{(\tau)} \right\rangle$$

*Proof.* We can rewrite $\Delta(\boldsymbol{\xi}_{i',p'}^{(\tau)})$ as

$$\begin{aligned}
\Delta(\boldsymbol{\xi}_{i',p'}^{(\tau)}) = &-\frac{\eta\lambda_c}{n} \sum_{i=1}^{n} \sum_{p \in \mathcal{P}_i^{\xi}} y_i^{(\tau)} y_i^{(\tau)} \ell'(F(\mathbf{x}_i^{(\tau)}), y_i^{(\tau)}) \phi'\left(\left\langle \mathbf{w}_c^{(\tau)}(t), y_i\mathbf{u} \right\rangle\right) \left\langle \mathbf{u}^{(\tau)}, \boldsymbol{\xi}_{i',p'}^{(\tau)} \right\rangle \\
&-\frac{\eta\lambda_c}{n} \sum_{i=1}^{n} \sum_{p \in \mathcal{P}_i^{\xi}} y_i^{(\tau)} y_i^{(\tau)} \ell'(F(\mathbf{x}_i^{(\tau)}), y_i^{(\tau)}) \phi'\left(\left\langle \mathbf{w}_c^{(\tau)}(t), y_i^{(\tau)}\mathbf{v}^{(\tau)} \right\rangle\right) \left\langle \mathbf{v}^{(\tau)}, \boldsymbol{\xi}_{i',p'}^{(\tau)} \right\rangle \\
&-\frac{\eta\lambda_c}{n} \sum_{i=1}^{n} \sum_{p \in \mathcal{P}_i^{\xi}} y_i^{(\tau)} \epsilon_i \ell'(F(\mathbf{x}_i^{(\tau)}), y_i^{(\tau)}) \phi'\left(\left\langle \mathbf{w}_c^{(\tau)}(t), \epsilon_i^{(\tau)}\boldsymbol{\zeta} \right\rangle\right) \left\langle \boldsymbol{\zeta}^{(\tau)}, \boldsymbol{\xi}_{i',p'}^{(\tau)} \right\rangle \\
&-\frac{\eta\lambda_c}{n} \sum_{i=1}^{n} \sum_{p \in \mathcal{P}_i^{\xi}} y_i^{(\tau)} \ell'(F(\mathbf{x}_i^{(\tau)}), y_i^{(\tau)}) \phi'\left(\left\langle \mathbf{w}_c^{(\tau)}(t), \boldsymbol{\xi}_{i,p}^{(\tau)} \right\rangle\right) \left\langle \boldsymbol{\xi}_{i,p}^{(\tau)}, \boldsymbol{\xi}_{i',p'}^{(\tau)} \right\rangle \\
= &-\frac{\eta\lambda_c}{n} \sum_{i=1}^{n} \sum_{p \in \mathcal{P}_i^{\xi}} y_i^{(\tau)} \ell'(F(\mathbf{x}_i^{(\tau)}), y_i^{(\tau)}) \phi'\left(\left\langle \mathbf{w}_c^{(\tau)}(t), \boldsymbol{\xi}_{i,p}^{(\tau)} \right\rangle\right) \left\langle \boldsymbol{\xi}_{i,p}^{(\tau)}, \boldsymbol{\xi}_{i',p'}^{(\tau)} \right\rangle
\end{aligned}$$

We conclude our proof. $\qquad \square$

## C. Theoretical Results at the Initialization

With the lemmas shown in Appendix A, we can individually analyze the inner product of different components. In task $\tau$, as the vectors $\left\{ \mathbf{u}^{(\tau)}, \mathbf{v}^{(\tau)}, \boldsymbol{\zeta}^{(\tau)} \right\}$ are mutually orthogonal in our assumption, we have $\left\langle \mathbf{u}^{(\tau)}, \mathbf{v}^{(\tau)} \right\rangle = \left\langle \mathbf{u}^{(\tau)}, \boldsymbol{\zeta}^{(\tau)} \right\rangle = \left\langle \mathbf{v}^{(\tau)}, \boldsymbol{\zeta}^{(\tau)} \right\rangle = 0$. We then analyze the inner product of two noise vectors.

**Lemma C.1.** *Let $\mathcal{I}_{\text{tr}}^{(\tau)} = \{(\mathbf{x}_i, y_i)\}_{i=1}^{n}$ be i.i.d. samples drawn from the distribution $\mathcal{D}_{\mathbf{z}}^{(\tau)}$ defined in Definition 3.1. Then, with probability at least $1 - O\left(\frac{n^2P^2 + nP}{d}\right)$, the following holds simultaneously:*

$$\forall i, i' \in [n],\ p, p' \in \mathcal{P}_i^{\xi},\ (i,p) \neq (i',p'),\quad \left|\left\langle \boldsymbol{\xi}_{i,p}^{(\tau)}, \boldsymbol{\xi}_{i',p'}^{(\tau)} \right\rangle\right| \leq a_1 \sigma_{\xi}^2 d^{-1}\sqrt{(d-3)\log(2d)}, \tag{6}$$

$$\forall i \in [n],\ p \in \mathcal{P}_i^{\xi},\quad \sigma_{\xi}^2 d^{-1}\left(d - 3 - 2\sqrt{(d-3)\log d}\right) \leq \left\|\boldsymbol{\xi}_{i,p}^{(\tau)}\right\|_2^2 \leq \sigma_{\xi}^2 d^{-1}\left(d - 3 + 2\sqrt{(d-3)\log d} + 2\log d\right). \tag{7}$$

*Proof.* We construct an orthonormal basis $\{\mathbf{e}_1, \mathbf{e}_2, \mathbf{e}_{\mathbf{rob}}, \tilde{\mathbf{e}}_1, \ldots, \tilde{\mathbf{e}}_{d-3}\}$ in $\mathbb{R}^d$, and define two row-orthonormal matrices:

$$M = [\tilde{\mathbf{e}}_1, \ldots, \tilde{\mathbf{e}}_{d-3}]^{\top} \in \mathbb{R}^{(d-3) \times d}, \quad M_{\perp} = [\mathbf{e}_1, \mathbf{e}_2, \mathbf{e}_{\mathbf{rob}}]^{\top} \in \mathbb{R}^{3 \times d}.$$

Then for any $i \in [n]$, $p \in \mathcal{P}_i^{\xi}$, we decompose the Gaussian vector $\boldsymbol{\xi}_{i,p}^{(\tau)}$ as:

$$\boldsymbol{\xi}_{i,p}^{(\tau)} = M^{\top}M\boldsymbol{\xi}_{i,p}^{(\tau)} + M_{\perp}^{\top}M_{\perp}\boldsymbol{\xi}_{i,p}^{(\tau)}.$$

Based on our definition, each $\boldsymbol{\xi}_{i,p}^{(\tau)} \sim \mathcal{N}(0, \sigma_{\xi}^2 d^{-1}(\mathbf{I}_d - \mathbf{e}_1\mathbf{e}_1^{\top} - \mathbf{e}_2\mathbf{e}_2^{\top} - \mathbf{e}_{\mathbf{rob}}\mathbf{e}_{\mathbf{rob}}^{\top}))$. Therefore, by using Lemma A.4, the projected components satisfy:

$$M\boldsymbol{\xi}_{i,p}^{(\tau)} \sim \mathcal{N}(0, \sigma_{\xi}^2 d^{-1}\mathbf{I}_{d-3}), \quad M_{\perp}\boldsymbol{\xi}_{i,p}^{(\tau)} = \mathbf{0}_{3 \times 1}.$$

Consequently, the inner product between any pair of distinct vectors $\boldsymbol{\xi}_{i,p}^{(\tau)}$ and $\boldsymbol{\xi}_{i',p'}^{(\tau)}$ satisfies:

$$\left\langle \boldsymbol{\xi}_{i,p}^{(\tau)}, \boldsymbol{\xi}_{i',p'}^{(\tau)} \right\rangle = \left\langle M\boldsymbol{\xi}_{i,p}^{(\tau)}, M\boldsymbol{\xi}_{i',p'}^{(\tau)} \right\rangle + \left\langle M_\perp \boldsymbol{\xi}_{i,p}^{(\tau)}, M_\perp \boldsymbol{\xi}_{i',p'}^{(\tau)} \right\rangle = \left\langle M\boldsymbol{\xi}_{i,p}^{(\tau)}, M\boldsymbol{\xi}_{i',p'}^{(\tau)} \right\rangle,$$

since both projections onto $M_\perp$ are zero.

According to Lemma A.2, for any distinct pair $(i,p) \neq (i',p')$, there exists a constant $a'$ such that:

$$\mathbb{P}\left( \left| \left\langle M\boldsymbol{\xi}_{i,p}^{(\tau)}, M\boldsymbol{\xi}_{i',p'}^{(\tau)} \right\rangle \right| \leq a'\sigma_\xi^2 d^{-1}\sqrt{(d-3)\log(2/\delta)} \right) \geq 1 - \delta.$$

Applying the union bound over all $nP(nP-1)$ distinct index pairs, we conclude that with probability at least $1 - nP(nP-1)\delta$, the following inequality holds uniformly:

$$\left| \left\langle \boldsymbol{\xi}_{i,p}^{(\tau)}, \boldsymbol{\xi}_{i',p'}^{(\tau)} \right\rangle \right| \leq a''\sigma_\xi^2 d^{-1}\sqrt{(d-3)\log(2/\delta)}, \quad \forall(i,p) \neq (i',p'),$$

for some constant $a'' > 0$. Substituting $\delta = 1/d$ yields the desired bound in (6) with constant $a_1$, with overall failure probability bounded by $O\left(n^2P^2/d\right)$.

To prove (7), since the distribution of $M\boldsymbol{\xi}_{i,p}^{(\tau)}$ is $\mathcal{N}(0, \sigma_\xi^2 d^{-1}\mathbf{I}_{d-3})$, we have

$$\frac{\left\| \boldsymbol{\xi}_{i,p}^{(\tau)} \right\|_2^2}{\sigma_\xi^2 d^{-1}} = \frac{\left\| M\boldsymbol{\xi}_{i,p}^{(\tau)} \right\|_2^2}{\sigma_\xi^2 d^{-1}} \sim \chi^2(d-3),$$

where $\chi^2(d-3)$ denotes the chi-squared distribution with $d-3$ degrees of freedom. Applying Lemma A.1 with $a_i = 1$, we yield:

$$\mathbb{P}\left( \left\| \boldsymbol{\xi}_{i,p}^{(\tau)} \right\|_2^2 \geq \sigma_\xi^2 d^{-1}\left( d-3+2\sqrt{(d-3)\log d}+2\log d \right) \right) \leq \frac{1}{d},$$

$$\mathbb{P}\left( \left\| \boldsymbol{\xi}_{i,p}^{(\tau)} \right\|_2^2 \leq \sigma_\xi^2 d^{-1}\left( d-3-2\sqrt{(d-3)\log d} \right) \right) \leq \frac{1}{d}.$$

Finally, applying a union bound over all $nP$ such vectors shows that the inequalities in (7) hold simultaneously with probability at least $1 - O\left(nP/d\right)$. $\qquad\square$

Since $\mathbf{w}$ is initialized as $\mathbf{w}_c^{(1)}(0) \sim \mathcal{N}(0, \sigma_0^2\mathbf{I}_d)$, the analysis of the inner product of the weight vector and the patch vector is similar to the proof of Lemma C.1.

**Lemma C.2.** *Let the network weights be initialized as $\mathbf{w}_c^{(1)}(0) \sim \mathcal{N}(0, \sigma_0^2\mathbf{I}_d)$ for each class $c \in [C]$, and let $\mathcal{S}_{tr}^{(\tau)} = \{(\mathbf{x}_i^{(\tau)}, y_i^{(\tau)})\}_{i=1}^n$ be i.i.d. samples from the distribution $\mathcal{D}_\mathbf{z}$ defined in Definition 3.1. If $C = \Theta\left(\log d\right)$, then with probability at least $1 - O\left(\frac{nPC}{d}\right)$, there exists constants $a_1, a_2$ such that following inequalities hold:*

$$\forall \mathbf{e} \in \{\mathbf{u}^{(\tau)}, \mathbf{v}^{(\tau)}, \boldsymbol{\zeta}^{(\tau)}\}, \quad \max_{c\in[C]}\left| \left\langle \mathbf{w}_c^{(1)}(0), \mathbf{e} \right\rangle \right| \leq \sqrt{\log d}\,\|\mathbf{e}\|_2\,\sigma_0, \quad \max_{c\in[C]}\left\langle \mathbf{w}_c^{(1)}(0), \mathbf{e} \right\rangle \geq \tfrac{1}{2}\|\mathbf{e}\|_2\,\sigma_0,$$

$$\forall i \in [n],\ p \in \mathcal{P}_i^\xi, \quad \max_{c\in[C]}\left| \left\langle \mathbf{w}_c^{(1)}(0), \boldsymbol{\xi}_{i,p}^{(\tau)} \right\rangle \right| \leq a_1\sigma_0\sigma_\xi\sqrt{d^{-1}(d-3)\log(2d)}, \quad \max_{c\in[C]}\left\langle \mathbf{w}_c^{(1)}(0), \boldsymbol{\xi}_{i,p}^{(\tau)} \right\rangle \geq a_2\sigma_0\sigma_\xi.$$

*Proof.* Since each $\mathbf{w}_c^{(1)}(0)$ is sampled from a spherical Gaussian, the inner products $\left\langle \mathbf{w}_c^{(1)}(0), \mathbf{e} \right\rangle$ for any fixed vector $\mathbf{e} \in \{\mathbf{u}, \mathbf{v}, \boldsymbol{\zeta}\}$ follow the distribution $\mathcal{N}(0, \|\mathbf{e}\|_2^2\,\sigma_0^2)$. By using standard sub-Gaussian tail bounds in Lemma A.3, we have:

$$\mathbb{P}\left[ \exists c \in [C] : \left| \left\langle \mathbf{w}_c^{(1)}(0), \mathbf{e} \right\rangle \right| \geq \sqrt{2\log d}\,\|\mathbf{e}\|_2\,\sigma_0 \right] \leq \sum_{c=1}^C \mathbb{P}\left[ \left| \left\langle \mathbf{w}_c^{(1)}(0), \mathbf{e} \right\rangle \right| \geq \sqrt{2\log d}\,\|\mathbf{e}\|_2\,\sigma_0 \right] \leq O\left(\frac{C}{d}\right). \quad (8)$$

Moreover, applying Proposition A.7, we yield:

$$\mathbb{P}\left[ \max_{c\in[C]}\left\langle \mathbf{w}_c^{(1)}(0), \mathbf{e} \right\rangle \leq \tfrac{1}{2}\|\mathbf{e}\|_2\,\sigma_0 \right] = \prod_{c=1}^C \mathbb{P}\left[ \left\langle \mathbf{w}_c^{(1)}(0), \mathbf{e} \right\rangle \leq \tfrac{1}{2}\|\mathbf{e}\|_2\,\sigma_0 \right] \leq \left(\tfrac{3}{4}\right)^C \leq \mathbf{e}^{-C/4}. \quad (9)$$

Inequalities (8) and (9) jointly establish the first line of the lemma.

For the second line, consider the noise vectors $\boldsymbol{\xi}_{i,p}^{(\tau)} \sim \mathcal{N}\left(0, \sigma_\xi^2 d^{-1}(\mathbf{I}_d - \mathbf{e}_1\mathbf{e}_1^\top - \mathbf{e}_2\mathbf{e}_2^\top - \mathbf{e}_{\text{rob}}\mathbf{e}_{\text{rob}}^\top)\right)$. Let $\{\mathbf{e}_1, \mathbf{e}_2, \mathbf{e}_{\text{rob}}, \tilde{\mathbf{e}}_1, \ldots, \tilde{\mathbf{e}}_{d-3}\}$ be an orthonormal basis of $\mathbb{R}^d$, and define two projection matrices:

$$M = [\tilde{\mathbf{e}}_1, \ldots, \tilde{\mathbf{e}}_{d-3}]^\top \in \mathbb{R}^{(d-3)\times d}, \quad M_\perp = [\mathbf{e}_1, \mathbf{e}_2, \mathbf{e}_{\text{rob}}]^\top \in \mathbb{R}^{3\times d}.$$

By Lemma A.4, we have

$$M\mathbf{w}_c^{(1)}(0) \sim \mathcal{N}(0, \sigma_0^2\mathbf{I}_{d-3}), \quad M\boldsymbol{\xi}_{i,p}^{(\tau)} \sim \mathcal{N}(0, \sigma_\xi^2 d^{-1}\mathbf{I}_{d-3}), \quad M_\perp\boldsymbol{\xi}_{i,p}^{(\tau)} = \mathbf{0}_{3\times 1}.$$

Thus, we can write

$$\langle \mathbf{w}_c^{(1)}(0), \boldsymbol{\xi}_{i,p}^{(\tau)} \rangle = \langle M\mathbf{w}_c^{(1)}(0), M\boldsymbol{\xi}_{i,p}^{(\tau)} \rangle + \langle M_\perp\mathbf{w}_c^{(1)}(0), M_\perp\boldsymbol{\xi}_{i,p}^{(\tau)} \rangle = \langle M\mathbf{w}_c^{(1)}(0), M\boldsymbol{\xi}_{i,p}^{(\tau)} \rangle.$$

Applying Lemma A.2 and taking a union bound over $nPC$ such vectors yields:

$$\mathbb{P}\left[\max_{i,p,c}\left|\langle \mathbf{w}_c^{(1)}(0), \boldsymbol{\xi}_{i,p}^{(\tau)} \rangle\right| \leq a_1\sigma_0\sigma_\xi\sqrt{d^{-1}(d-3)\log(2d)}\right] \geq 1 - O\left(\frac{nPC}{d}\right), \tag{10}$$

where $a_1$ is a constant. Likewise, applying the lower bound in Lemma A.2, we have:

$$\mathbb{P}\left[\min_{i\in[n],p\in\mathcal{P}_i^\xi}\max_{c\in[C]}\langle \mathbf{w}_c^{(1)}(0), \boldsymbol{\xi}_{i,p}^{(\tau)} \rangle \geq a_2\sigma_0\sigma_\xi\sqrt{d^{-1}(d-3)}\right] \geq 1 - O\left(nP\left(\tfrac{3}{4}\right)^C\right) \geq 1 - O\left(nPe^{-C/4}\right). \tag{11}$$

where $a_2$ is a constant. Finally, combining (8) to (11) and applying a union bound completes the proof. $\square$

At the initialization, we have the following result:

**Lemma 4.1.** *Given the weights $\mathbf{w}_c^{(1)}$ initialized as $\mathbf{w}_c^{(1)}(0) \sim \mathcal{N}(0, \sigma_0\mathbf{I}_d)$, at the beginning of the first stage, with a probability of $1 - O\left(\frac{n^2P^2C}{poly(d)}\right)$, we have*

$$\forall\mathbf{e}\in\left\{\mathbf{u}^{(1)}, \mathbf{v}^{(1)}, \boldsymbol{\zeta}^{(1)}\right\}, \max_{c\in[C]}\left|\left\langle\mathbf{w}_c^{(1)}(0), \mathbf{e}\right\rangle\right| \leq \widetilde{O}\left(\|\mathbf{e}\|_2\sigma_0\right),$$

$$\forall\mathbf{e}\in\left\{\mathbf{u}^{(1)}, \mathbf{v}^{(1)}, \boldsymbol{\zeta}^{(1)}\right\}, \max_{c\in[C]}\left\langle\mathbf{w}_c^{(1)}(0), \mathbf{e}\right\rangle \geq \Omega\left(\|\mathbf{e}\|_2\sigma_0\right),$$

$$\forall i\in[n], p\in\mathcal{P}_i^\xi, \max_{c\in[C]}\left|\left\langle\mathbf{w}_c^{(1)}(0), \boldsymbol{\xi}_{i,p}^{(1)}\right\rangle\right| \leq \widetilde{O}\left(\sigma_0\sigma_\xi\right),$$

$$\forall i\in[n], p\in\mathcal{P}_i^\xi, \max_{c\in[C]}\left\langle\mathbf{w}_c^{(1)}(0), \boldsymbol{\xi}_{i,p}^{(1)}\right\rangle \geq \Omega\left(\sigma_0\sigma_\xi\right).$$

*Proof.* The results can be obtained by using Lemma C.2. Recall that $C = \Theta(\log d)$ in Condition 3.6, using a union bound, the probability of the inequalities hold is at least $1 - O\left(\frac{n^2P^2C}{poly(d)}\right)$. $\square$

## D. Learning Dynamics in the First Stage

In this section, we study the learning process of the first task, which can be seen as single-task learning. If the signal of the task-specific feature is larger than other components, the feature can be captured by CNN.

**Lemma 5.1.** *In the first stage, under Condition 3.6, and given $T \geq \widetilde{\Omega}\left(\frac{1}{\eta\sigma_0\alpha_u^3}\right)$, for any $0 < t \leq T, i \in \mathcal{I}_{tr}^{(1)}, p \in \mathcal{P}_i^\xi$, we have following:*

$$\max_{c\in C}\left\langle\mathbf{w}_c^{(1)}(t+1), \boldsymbol{\xi}_{i,p}^{(1)}\right\rangle - \max_{c\in C}\left\langle\mathbf{w}_c^{(1)}(t), \boldsymbol{\xi}_{i,p}^{(1)}\right\rangle \leq o\left(\sigma_0\sigma_\xi\right).$$

*Moreover, we have*

$$\max_{c\in C}\left\langle\mathbf{w}_c^{(1)}(t), \boldsymbol{\xi}_{i,p}^{(1)}\right\rangle \leq \widetilde{O}\left(\sigma_0\sigma_\xi\right).$$

*Proof.* We analyze the change in the projection $\left\langle \mathbf{w}_c^{(1)}(t), \boldsymbol{\xi}_{i',p'}^{(1)} \right\rangle$ induced by the gradient update, denoted by $\Delta(\boldsymbol{\xi}_{i',p'}^{(1)})$ from Lemma B.3:

$$\Delta(\boldsymbol{\xi}_{i',p'}^{(1)}) = -\frac{\eta\lambda_c}{n} \sum_{i=1}^{n} \sum_{p\in\mathcal{P}_i^\xi} y_i^{(1)} \ell' \left( F(\mathbf{x}_i^{(1)}), y_i^{(1)} \right) \phi' \left( \left\langle \mathbf{w}_c^{(1)}(t), \boldsymbol{\xi}_{i,p}^{(1)} \right\rangle \right) \left\langle \boldsymbol{\xi}_{i,p}^{(1)}, \boldsymbol{\xi}_{i',p'}^{(1)} \right\rangle .$$

We separate the summand with $(i, p) = (i', p')$:

$$\Delta(\boldsymbol{\xi}_{i',p'}^{(1)}) = -\frac{\eta\lambda_c}{n} \sum_{\substack{i\in[n],\, p\in\mathcal{P}_i^\xi \\ (i,p)\neq(i',p')}} y_i^{(1)} \ell' \left( F(\mathbf{x}_i^{(1)}), y_i^{(1)} \right) \phi' \left( \left\langle \mathbf{w}_c^{(1)}(t), \boldsymbol{\xi}_{i,p}^{(1)} \right\rangle \right) \left\langle \boldsymbol{\xi}_{i,p}^{(1)}, \boldsymbol{\xi}_{i',p'}^{(1)} \right\rangle$$

$$- \frac{\eta\lambda_c}{n} y_{i'}^{(1)} \ell' \left( F(\mathbf{x}_{i'}^{(1)}), y_{i'}^{(1)} \right) \phi' \left( \left\langle \mathbf{w}_c^{(1)}(t), \boldsymbol{\xi}_{i',p'}^{(1)} \right\rangle \right) \left\| \boldsymbol{\xi}_{i',p'}^{(1)} \right\|^2 .$$

We now upper bound both terms. Note that $|\ell'| \leq 1$. Lemma C.1 implies that

$$\left| \Delta(\boldsymbol{\xi}_{i',p'}^{(1)}) \right| \leq \frac{\eta\lambda_c}{n} \sum_{\substack{i\in[n],\, p\in\mathcal{P}_i^\xi \\ (i,p)\neq(i',p')}} \left| \phi' \left( \left\langle \mathbf{w}_c^{(1)}(t), \boldsymbol{\xi}_{i,p}^{(1)} \right\rangle \right) \left\langle \boldsymbol{\xi}_{i,p}^{(1)}, \boldsymbol{\xi}_{i',p'}^{(1)} \right\rangle \right| + \frac{\eta\lambda_c}{n} \left| \phi' \left( \left\langle \mathbf{w}_c^{(1)}(t), \boldsymbol{\xi}_{i',p'}^{(1)} \right\rangle \right) \right| \left\| \boldsymbol{\xi}_{i',p'}^{(1)} \right\|^2$$

$$\leq \eta\lambda_c \widetilde{O} \left( \max_{c,i,p} \left| \phi' \left( \left\langle \mathbf{w}_c^{(1)}(t), \boldsymbol{\xi}_{i,p}^{(1)} \right\rangle \right) \right| \cdot \sigma_\xi^2 \cdot \left( \frac{P}{\sqrt{d}} + \frac{1}{n} \right) \right) .$$

Given that $d$ is sufficiently large under Condition 3.6, we have $P/\sqrt{d} = o(1/n)$. Thus, the bound simplifies to

$$\left| \Delta(\boldsymbol{\xi}_{i',p'}^{(1)}) \right| \leq \lambda_c \widetilde{O} \left( \eta \max_{c,i,p} \left| \phi' \left( \left\langle \mathbf{w}_c^{(1)}(t), \boldsymbol{\xi}_{i,p}^{(1)} \right\rangle \right) \right| \cdot \frac{\sigma_\xi^2}{n} \right) .$$

We proceed by induction. At $t = 0$, by Lemma 4.1 and $\eta \leq O\left(1/\left(\sigma_0\alpha_u^3\right)\right)$, for all $i \in \mathcal{I}_{tr}^{(1)}, p \in \mathcal{P}_i^\xi, c \in [C]$, we have

$$\left\langle \mathbf{w}_c^{(1)}(0), \boldsymbol{\xi}_{i,p}^{(1)} \right\rangle \leq \widetilde{O}(\sigma_0\sigma_\xi).$$

Suppose this upper bound holds at step $t$. Then, the update satisfies

$$\left| \Delta(\boldsymbol{\xi}_{i,p}^{(1)}) \right| \leq \widetilde{O} \left( \frac{\eta\sigma_0^2\sigma_\xi^4}{n} \right) = o(\sigma_0\sigma_\xi),$$

since $\eta \leq O\left(1/\left(\sigma_0\alpha_u^3\right)\right)$, $n\alpha_u^3/\sigma_\xi^3 \geq \omega(1)$ and $\lambda_c = 1$.

Thus, we obtain

$$\left\langle \mathbf{w}_c^{(1)}(t+1), \boldsymbol{\xi}_{i,p}^{(1)} \right\rangle = \left\langle \mathbf{w}_c^{(1)}(t), \boldsymbol{\xi}_{i,p}^{(1)} \right\rangle + \Delta(\boldsymbol{\xi}_{i,p}^{(1)}) \leq \widetilde{O}(\sigma_0\sigma_\xi) + o(\sigma_0\sigma_\xi) = \widetilde{O}(\sigma_0\sigma_\xi).$$

By induction, we conclude that for all $0 \leq t \leq T$,

$$\max_{c\in[C]} \left\langle \mathbf{w}_c^{(1)}(t), \boldsymbol{\xi}_{i,p}^{(1)} \right\rangle \leq \widetilde{O}(\sigma_0\sigma_\xi), \quad \text{and} \quad \left| \Delta(\boldsymbol{\xi}_{i,p}^{(1)}) \right| \leq o(\sigma_0\sigma_\xi),$$

as claimed. We conclude our proof. $\qquad\square$

**Lemma D.1** (Learning the task-specific feature). *Under Condition 3.6, suppose that for all iterations $t \in [0, T]$ with some $T \geq \widetilde{\Omega}\left(\frac{1}{\eta\sigma_0\alpha_u^3}\right)$, the weights satisfy*

$$\max_{c\in[C]} \max_{\mathbf{e}\in\{\mathbf{u}^{(1)},\mathbf{v}^{(1)},\boldsymbol{\zeta}^{(1)}\}} \left\langle \mathbf{w}_c^{(1)}(t), \mathbf{e} \right\rangle \leq O\left(C^{-1/3}\right).$$

*Then there exists an iteration $T_u^{(1)} \leq \widetilde{O}\left(\frac{1}{\eta\sigma_0\alpha_u^3}\right)$ such that*

$$\max_{c\in[C]}\left\langle \mathbf{w}_c^{(1)}(T_u^{(1)}), \mathbf{u}^{(1)} \right\rangle \geq \Omega\left(C^{-1/3}\right).$$

*Proof.* Suppose that for all $t \in [0, T]$,

$$\max_{c\in[C]}\max_{\mathbf{e}\in\{\mathbf{u}^{(1)},\mathbf{v}^{(1)},\boldsymbol{\zeta}^{(1)}\}}\left\langle \mathbf{w}_c^{(1)}(t), \mathbf{e} \right\rangle \leq O\left(C^{-1/3}\right),$$

where $T \geq \Omega\left(1/(\eta\sigma_0\alpha_u^3)\right)$. Then for each $i \in [n]$, the margin satisfies

$$y_i^{(1)} F_t^{(1)}(\mathbf{x}_i^{(1)}) \leq O(1),$$

which implies that

$$\Omega(1) \leq -\ell_i' \leq 1, \quad \forall i \in [n]. \tag{12}$$

We can both upper and lower bound the growth in alignment with the task-specific feature $\mathbf{u}^{(1)}$ as

$$\max_{c\in[C]}\left\langle \mathbf{w}_c^{(1)}(t+1), \mathbf{u}^{(1)} \right\rangle - \max_{c\in[C]}\left\langle \mathbf{w}_c^{(1)}(t), \mathbf{u}^{(1)} \right\rangle$$
$$= \Theta\left(n^{-1}\eta\alpha_u^2\sum_{i=1}^n(-\ell_i')\cdot\phi'\left(\left|\left\langle \mathbf{w}_c^{(1)}(t), \mathbf{u}^{(1)} \right\rangle\right|\right)\right) = \Theta\left(\eta\alpha_u^2\phi'\left(\left|\left\langle \mathbf{w}_c^{(1)}(t), \mathbf{u}^{(1)} \right\rangle\right|\right)\right), \tag{13}$$

where the last equality follows from Equation (12).

Equation (13) shows that the sequence $\left\{\max_{c\in[C]}\left\langle \mathbf{w}_c^{(1)}(t), \mathbf{u}^{(1)} \right\rangle\right\}_{t=0}^T$ is monotonically increasing. By Lemma 4.1, we have the initialization

$$\max_{c\in[C]}\left\langle \mathbf{w}_c^{(1)}(0), \mathbf{u}^{(1)} \right\rangle = \widetilde{\Theta}\left(\sigma_0\alpha_u\right).$$

Let $T_u^{(1)}$ be the first time $t$ such that $\max_{c\in C}\left\langle \mathbf{w}_c^{(1)}(t), \mathbf{u}^{(1)} \right\rangle \geq \Omega\left(C^{-1/3}\right)$. Starting from some $\max_{c\in C}\left\langle \mathbf{w}_c^{(1)}(t'), \mathbf{u}^{(1)} \right\rangle$, the number of iterations it takes to reach $\max_{c\in C}\left\langle \mathbf{w}_c^{(1)}(t), \mathbf{u}^{(1)} \right\rangle \geq 2\max_{c\in C}\left\langle \mathbf{w}_c^{(1)}(t'), \mathbf{u}^{(1)} \right\rangle$ is at most $O\left(\frac{\max_{c\in C}\left\langle \mathbf{w}_c^{(1)}(t'), \mathbf{u}^{(1)} \right\rangle}{\eta\alpha_u^2\left(\max_{c\in C}\left\langle \mathbf{w}_c^{(1)}(t'), \mathbf{u}^{(1)} \right\rangle\right)^2}\right)$. Then, starting from $\widetilde{\Theta}\left(\sigma_0\alpha_u\right)$, it takes at most

$$T_u^{(1)} \leq \widetilde{O}\left(\sum_{i=0}^\infty \frac{2^i\sigma_0\alpha_u}{\eta\alpha_u^2\left(2^i\sigma_0\alpha_u\right)^2}\right) \leq \widetilde{O}\left(\frac{1}{\eta\sigma_0\alpha_u^3}\right) \tag{14}$$

times steps to reach $\max_{c\in C}\left\langle \mathbf{w}_c^{(1)}(t), \mathbf{u}^{(1)} \right\rangle \geq \Omega\left(C^{-1/3}\right)$. This completes the proof.

$\square$

**Theorem 5.2.** *In the first stage, given a training set $\mathcal{S}_{tr}^{(1)}$ with size $n$, if $\alpha_v \leq o\left(\alpha_u\right)$, there exists $\widetilde{T}^{(1)} \leq \widetilde{O}\left(\frac{1}{\eta\sigma_0\alpha_u^3}\right)$ such that for any $T^{(1)} \geq \widetilde{T}^{(1)}$, the network $F_{T^{(1)}}$ fits all training data points with a high probability:*

$$\mathbb{P}\left[\forall i \in \mathcal{S}_{tr}^{(1)}, y_i F_{T^{(1)}}^{(1)}(\mathbf{x}_i) \geq \widetilde{\Omega}(1)\right] \geq 1 - O\left(\frac{n^2P^2C}{poly(d)}\right).$$

*Moreover, $F_{T^{(1)}}^{(1)}$ achieves a high accuracy on test data points at $T^{(1)}$:*

$$\mathbb{P}_{(\mathbf{x},y)\sim\mathcal{D}_\mathbf{z}^{(1)}}\left[y F_{T^{(1)}}^{(1)}(\mathbf{x}) > 0\right] \geq 1 - O\left(\frac{nP^2C}{poly(d)}\right).$$

At $t = 0$, by Lemma 4.1, we have

$$\max_{c \in [C]} \left\langle \mathbf{w}_c^{(1)}(0), \mathbf{u}^{(1)} \right\rangle = \widetilde{\Theta}(\sigma_0 \alpha_u), \qquad \max_{c \in [C]} \left\langle \mathbf{w}_c^{(1)}(0), \mathbf{v}^{(1)} \right\rangle = \widetilde{\Theta}(\sigma_0 \alpha_v), \qquad \max_{c \in [C]} \left\langle \mathbf{w}_c^{(1)}(0), \boldsymbol{\zeta}^{(1)} \right\rangle = \widetilde{\Theta}(\sigma_0 \alpha_\zeta).$$

Let $T^\star$ be the final time such that $\max_{c \in [C]} \left\langle \mathbf{w}_c^{(1)}(t), \mathbf{u}^{(1)} \right\rangle \leq O(C^{-1/3})$. Suppose that at some time $t \leq T^\star$, we have

$$\max_{c \in [C]} \left\langle \mathbf{w}_c^{(1)}(t), \mathbf{v}^{(1)} \right\rangle = \widetilde{\Theta}(\sigma_0 \alpha_v), \qquad \max_{c \in [C]} \left| \left\langle \mathbf{w}_c^{(1)}(t), \boldsymbol{\zeta}^{(1)} \right\rangle \right| = \widetilde{\Theta}(\sigma_0 \alpha_\zeta),$$

and additionally,

$$\max_{c \in [C]} \left\langle \mathbf{w}_c^{(1)}(t), \mathbf{u}^{(1)} \right\rangle \leq O(C^{-1/3}).$$

Then the margin satisfies

$$y_i^{(1)} F_t^{(1)}(\mathbf{x}_i^{(1)}) \leq O(1), \qquad \forall i \in [n].$$

This implies that

$$\Omega(1) \leq -\ell_i' \leq 1. \tag{15}$$

We now analyze the update of $\max_{c \in [C]} \left\langle \mathbf{w}_c^{(1)}(t), \mathbf{v}^{(1)} \right\rangle$. At time $t$, we have

$$\max_{c \in [C]} \left\langle \mathbf{w}_c^{(1)}(t+1), \mathbf{v}^{(1)} \right\rangle - \max_{c \in [C]} \left\langle \mathbf{w}_c^{(1)}(t), \mathbf{v}^{(1)} \right\rangle$$

$$= \Theta \left( \frac{\eta \alpha_v^2}{n} \sum_{i=1}^n (-\ell_i') \cdot \phi' \left( \left\langle \mathbf{w}_c^{(1)}(t), \mathbf{v}^{(1)} \right\rangle \right) \right) = \Theta \left( \eta \alpha_v^2 \phi' \left( \left\langle \mathbf{w}_c^{(1)}(t), \mathbf{v}^{(1)} \right\rangle \right) \right),$$

and $\left\{ \max_{c \in [C]} \left\langle \mathbf{w}_c^{(1)}(t), \mathbf{v}^{(1)} \right\rangle \right\}$ is an increasing sequence, hence

$$\max_{c \in [C]} \left\langle \mathbf{w}_c^{(1)}(t+1), \mathbf{v}^{(1)} \right\rangle \geq \max_{c \in [C]} \left\langle \mathbf{w}_c^{(1)}(t), \mathbf{v}^{(1)} \right\rangle \geq \widetilde{\Omega}(\sigma_0 \alpha_v). \tag{16}$$

Since $\max_{c \in [C]} \left\langle \mathbf{w}_c^{(1)}(t), \mathbf{v}^{(1)} \right\rangle \leq \widetilde{O}(\sigma_0 \alpha_v)$, $\eta \leq O\left( 1/\left( \sigma_0 \alpha_u^3 \right) \right)$, and $\alpha_v \leq o\left( \alpha_u \right)$, it follows that

$$\max_{c \in [C]} \left\langle \mathbf{w}_c^{(1)}(t+1), \mathbf{v}^{(1)} \right\rangle \leq \max_{c \in [C]} \left\langle \mathbf{w}_c^{(1)}(t), \mathbf{v}^{(1)} \right\rangle + \Theta(\eta \alpha_v^4 \sigma_0^2) \leq \widetilde{O}(\sigma_0 \alpha_v) + o(\sigma_0 \alpha_v) \leq \widetilde{O}(\sigma_0 \alpha_v). \tag{17}$$

Combining Equations (16) and (17), we yield

$$\max_{c \in [C]} \left\langle \mathbf{w}_c^{(1)}(t+1), \mathbf{v}^{(1)} \right\rangle = \widetilde{\Theta}(\sigma_0 \alpha_v).$$

We now analyze the alignment with the random feature $\boldsymbol{\zeta}^{(1)}$. For any $t$, we have

$$\left| \max_{c \in [C]} \left\langle \mathbf{w}_c^{(1)}(t+1), \boldsymbol{\zeta}^{(1)} \right\rangle - \max_{c \in [C]} \left\langle \mathbf{w}_c^{(1)}(t), \boldsymbol{\zeta}^{(1)} \right\rangle \right| \leq \frac{\eta \lambda_c \left\| \boldsymbol{\zeta}^{(1)} \right\|_2^2 \cdot \phi' \left( \max_{c \in [C]} \left| \left\langle \mathbf{w}_c^{(1)}(t), \boldsymbol{\zeta}^{(1)} \right\rangle \right| \right)}{n} \cdot \left| \mathcal{G}^{(1)} \right|.$$

Define a sequence $\{\Phi(t)\}_{t=0}^T$ by

$$\Phi(0) = \max_{c \in [C]} \left| \left\langle \mathbf{w}_c^{(1)}(0), \boldsymbol{\zeta}^{(1)} \right\rangle \right|,$$

$$\Phi(t+1) = \Phi(t) + \frac{\eta \lambda_c \left\| \boldsymbol{\zeta}^{(1)} \right\|_2^2 \cdot \phi' \left( \max_{c \in [C]} \left| \left\langle \mathbf{w}_c^{(1)}(t), \boldsymbol{\zeta}^{(1)} \right\rangle \right| \right)}{n} \cdot \left| \mathcal{G}^{(1)} \right|.$$

By the bounded margin condition in (15) and Lemma A.8, we have

$$\left| \mathcal{G}^{(1)} \right| = \Theta(|n_= - n_{\neq}|) \leq \widetilde{O}(n^{-1/2}),$$

so

$$\Phi(t+1) \leq \Phi(t) + \frac{\eta \lambda_c \left\| \boldsymbol{\zeta}^{(1)} \right\|_2^2 \cdot \phi'\left( \left\langle \mathbf{w}_c^{(1)}(t), \boldsymbol{\zeta}^{(1)} \right\rangle \right)}{\sqrt{n}}.$$

Since $\Phi(t) \leq \widetilde{O}(\sigma_0 \alpha_\zeta)$, $\eta \leq O\left(1/\left(\sigma_0 \alpha_u^3\right)\right)$, it follows that

$$\Phi(t+1) \leq \Phi(t) + O(\eta \alpha_\zeta^4 \sigma_0^2 n^{-1/2}) \leq \Phi(t) + o(\sigma_0 \alpha_\zeta) \leq \widetilde{O}(\sigma_0 \alpha_\zeta).$$

and

$$\Phi(t+1) \geq \Phi(t) - O(\eta \alpha_\zeta^4 \sigma_0^2 n^{-1/2}) \geq \Phi(t) - o(\sigma_0 \alpha_\zeta) \geq \widetilde{\Omega}(\sigma_0 \alpha_\zeta).$$

Then for all $t \in [0, T^\star]$, we have

$$\max_{c \in [C]} \left\langle \mathbf{w}_c^{(1)}(t), \mathbf{v}^{(1)} \right\rangle = \widetilde{\Theta}(\sigma_0 \alpha_v), \qquad \max_{c \in [C]} \left| \left\langle \mathbf{w}_c^{(1)}(t), \boldsymbol{\zeta}^{(1)} \right\rangle \right| = \widetilde{\Theta}(\sigma_0 \alpha_\zeta). \tag{18}$$

By Lemma D.1, there exists $T_u^{(1)} \leq \widetilde{O}(1/(\eta \sigma_0 \alpha_u^3))$ such that

$$\max_{c \in [C]} \left\langle \mathbf{w}_c^{(1)}(T_u^{(1)}), \mathbf{u}^{(1)} \right\rangle \geq \Omega(C^{-1/3}). \tag{19}$$

At $T^{(1)} \geq \widetilde{T}^{(1)} = T_u^{(1)}$, the output for any training sample $(\mathbf{x}_i, y_i) \in \mathcal{S}_{tr}^{(1)}$ can be expressed as

$$y_i F_{T^{(1)}}^{(1)}(\mathbf{x}_i) = \sum_{c=1}^{C} \lambda_c \phi\left( \left\langle \mathbf{w}_c^{(1)}(T^{(1)}), \mathbf{u}^{(1)} \right\rangle \right) + \sum_{c=1}^{C} \lambda_c \phi\left( \left\langle \mathbf{w}_c^{(1)}(T^{(1)}), \mathbf{v}^{(1)} \right\rangle \right)$$

$$+ y_i \epsilon_i \sum_{c=1}^{C} \lambda_c \phi\left( \left\langle \mathbf{w}_c^{(1)}(T^{(1)}), \boldsymbol{\zeta}^{(1)} \right\rangle \right) + y_i \sum_{c=1}^{C} \sum_{p \in \mathcal{P}_i^\xi} \phi\left( \left\langle \mathbf{w}_c^{(1)}(T^{(1)}), \boldsymbol{\xi}_{i,p}^{(1)} \right\rangle \right).$$

By Lemma 5.1, we have

$$\max_{c \in [C]} \max_{p \in \mathcal{P}_i^\xi} \left| \left\langle \mathbf{w}_c^{(1)}(T^{(1)}), \boldsymbol{\xi}_{i,p}^{(1)} \right\rangle \right| \leq o(\sigma_0 \sigma_\xi). \tag{20}$$

Combining Equations (18) to (20), we yield

$$y_i F_{T^{(1)}}^{(1)}(\mathbf{x}_i) \geq \max_{c \in [C]} \lambda_c \phi\left( \left\langle \mathbf{w}_c^{(1)}(T^{(1)}), \mathbf{u}^{(1)} \right\rangle \right) + (C-1) \min_{c \in [C]} \lambda_c \phi\left( \left\langle \mathbf{w}_c^{(1)}(T^{(1)}), \mathbf{u}^{(1)} \right\rangle \right)$$

$$- C \max_{c \in [C]} \lambda_c \phi\left( \left| \left\langle \mathbf{w}_c^{(1)}(T^{(1)}), \mathbf{v}^{(1)} \right\rangle \right| \right) - C \max_{c \in [C]} \lambda_c \phi\left( \left| \left\langle \mathbf{w}_c^{(1)}(T^{(1)}), \boldsymbol{\zeta}^{(1)} \right\rangle \right| \right)$$

$$- CP \max_{c \in [C]} \max_{p \in \mathcal{P}_i^\xi} \phi\left( \left| \left\langle \mathbf{w}_c^{(1)}(T^{(1)}), \boldsymbol{\xi}_{i,p}^{(1)} \right\rangle \right| \right)$$

$$\geq \Omega(1/C) - CO(\sigma_0^3 \alpha_u^3) - CO(\sigma_0^3 \alpha_v^3) - CO(\sigma_0^3 \alpha_\zeta^3) - CPO(\sigma_0^3 \sigma_\xi^3) \geq \widetilde{\Omega}(1). \tag{21}$$

With probability at least $1 - O(n^2 P^2 C / \text{poly}(d))$, the bound in (21) holds, showing that the model correctly classifies all training samples with a significant margin.

Now consider a test sample $(\mathbf{x}, y)$ from the same data model. With probability at least $1 - O(nP^2C/\text{poly}(d))$, we have

$$yF_{T^{(1)}}^{(1)}(\mathbf{x}) = \sum_{c=1}^{C} \lambda_c \phi\left(\left\langle \mathbf{w}_c^{(1)}(T^{(1)}), \mathbf{u}^{(1)} \right\rangle\right) + \sum_{c=1}^{C} \lambda_c \phi\left(\left\langle \mathbf{w}_c^{(1)}(T^{(1)}), \mathbf{v}^{(1)} \right\rangle\right)$$

$$+ y\epsilon \sum_{c=1}^{C} \lambda_c \phi\left(\left\langle \mathbf{w}_c^{(1)}(T^{(1)}), \boldsymbol{\zeta}^{(1)} \right\rangle\right) + y \sum_{c=1}^{C} \sum_{p \in \mathcal{P}_i^{\xi}} \phi\left(\left\langle \mathbf{w}_c^{(1)}(T^{(1)}), \boldsymbol{\xi}_p^{(1)} \right\rangle\right) \geq \widetilde{\Omega}(1). \tag{22}$$

This completes the proof.

## E. Learning Dynamics in the Second Stage

In the second task, recall that $\mathbf{u}^{(2)} = \alpha_u \alpha_\zeta^{-1} \boldsymbol{\zeta}^{(1)}, \mathbf{v}^{(2)} = \mathbf{v}^{(1)}, \boldsymbol{\zeta}^{(2)} = \alpha_\zeta \alpha_u^{-1} \mathbf{u}^{(1)}$. Let $\mathbf{w}_c^{(2)}(0) = \mathbf{w}_c^{(1)}(T_u)$, at $T_u$, the following lemma shows the status of $\mathbf{w}_c^{(2)}$ at $T = 0$.

**Corollary 5.3.** *At the beginning of the second task, if $\alpha_u \geq \omega(\alpha_v)$, we have*

$$\max_{c \in [C]} \left\langle \mathbf{w}_c^{(2)}(0), \mathbf{u}^{(2)} \right\rangle = \widetilde{\Theta}(\sigma_0 \alpha_u),$$

$$\max_{c \in [C]} \left\langle \mathbf{w}_c^{(2)}(0), \mathbf{v}^{(2)} \right\rangle = \widetilde{\Theta}(\sigma_0 \alpha_v),$$

$$\max_{c \in [C]} \left\langle \mathbf{w}_c^{(2)}(0), \mathbf{u}^{(1)} \right\rangle \geq \widetilde{\Omega}(1).$$

*Proof.* Lemma D.1 shows that at the end of first task, we have $\max_{c \in [C]} \left\langle \mathbf{w}_c^{(1)}(\widetilde{T}^{(1)}), \mathbf{u}^{(1)} \right\rangle \geq \Omega\left(C^{-1/3}\right)$. Moreover, Equation (18) shows that at $\widetilde{T}^{(1)}$, we have $\max_{c \in [C]} \left\langle \mathbf{w}_c^{(1)}(\widetilde{T}^{(1)}), \mathbf{v}^{(1)} \right\rangle = \widetilde{\Theta}\left(\max_{c \in [C]} \left\langle \mathbf{w}_c^{(1)}(0), \mathbf{v}^{(1)} \right\rangle\right) = \widetilde{\Theta}(\sigma_0 \alpha_v)$ and $\max_{c \in [C]} \left\langle \mathbf{w}_c^{(1)}(\widetilde{T}^{(1)}), \boldsymbol{\zeta}^{(1)} \right\rangle = \widetilde{\Theta}\left(\max_{c \in [C]} \left\langle \mathbf{w}_c^{(1)}(0), \boldsymbol{\zeta}^{(1)} \right\rangle\right) = \widetilde{\Theta}(\sigma_0 \alpha_\zeta)$. Using the definition of $\boldsymbol{\zeta}^{(1)}, \mathbf{u}^{(2)}$, we conclude our proof. $\square$

**Corollary E.1.** *In the second stage, under Condition 3.6, given $T \geq \widetilde{\Omega}\left(\frac{1}{\eta \sigma_0 \alpha_u^3}\right)$, for any $0 < t \leq T$, we have*

$$\max_{c \in C} \left\langle \mathbf{w}_c^{(2)}(t+1), \boldsymbol{\xi}_{i,p}^{(2)} \right\rangle - \max_{c \in C} \left\langle \mathbf{w}_c^{(2)}(t), \boldsymbol{\xi}_{i,p}^{(2)} \right\rangle \leq o(\sigma_0 \sigma_\xi)$$

*hold for any $i, p$. Moreover, we have*

$$\forall i, p \quad \max_{c \in C} \left\langle \mathbf{w}_c^{(2)}(t), \boldsymbol{\xi}_{i,p}^{(2)} \right\rangle \leq \widetilde{O}(\sigma_0 \sigma_\xi).$$

*Proof.* Similar to the analysis in Lemma 5.1 for the first stage, we study the change in the projection $\left\langle \mathbf{w}_c^{(2)}(t), \boldsymbol{\xi}_{i',p'}^{(2)} \right\rangle$ due to the gradient update. Denoting this change by $\Delta(\boldsymbol{\xi}_{i',p'}^{(2)})$, we have

$$\Delta(\boldsymbol{\xi}_{i',p'}^{(2)}) = -\frac{\eta \lambda_c}{n} \sum_{i=1}^{n} \sum_{p \in \mathcal{P}_i^{\xi}} y_i^{(2)} \ell'\left(F(\mathbf{x}_i^{(2)}), y_i^{(2)}\right) \phi'\left(\left\langle \mathbf{w}_c^{(2)}(t), \boldsymbol{\xi}_{i,p}^{(2)} \right\rangle\right) \left\langle \boldsymbol{\xi}_{i,p}^{(2)}, \boldsymbol{\xi}_{i',p'}^{(2)} \right\rangle$$

$$- \frac{\eta \lambda_c}{n} y_{i'}^{(2)} \ell'\left(F(\mathbf{x}_{i'}^{(2)}), y_{i'}^{(2)}\right) \phi'\left(\left\langle \mathbf{w}_c^{(2)}(t), \boldsymbol{\xi}_{i',p'}^{(2)} \right\rangle\right) \left\|\boldsymbol{\xi}_{i',p'}^{(2)}\right\|^2.$$

We now upper bound the magnitude of the update. Using the fact that $|\ell'| \leq 1$, we obtain

$$\left|\Delta(\boldsymbol{\xi}_{i',p'}^{(2)})\right| \leq \frac{\eta \lambda_c}{n} \sum_{\substack{i \in [n], p \in \mathcal{P}_i^{\xi} \\ (i,p) \neq (i',p')}} \left|\phi'\left(\left\langle \mathbf{w}_c^{(2)}(t), \boldsymbol{\xi}_{i,p}^{(2)} \right\rangle\right) \left\langle \boldsymbol{\xi}_{i,p}^{(2)}, \boldsymbol{\xi}_{i',p'}^{(2)} \right\rangle\right| + \frac{\eta \lambda_c}{n} \left|\phi'\left(\left\langle \mathbf{w}_c^{(2)}(t), \boldsymbol{\xi}_{i',p'}^{(2)} \right\rangle\right)\right| \left\|\boldsymbol{\xi}_{i',p'}^{(2)}\right\|^2$$

$$\leq \lambda_c \widetilde{O}\left(\eta \max_{c,i,p} \left|\phi'\left(\left\langle \mathbf{w}_c^{(2)}(t), \boldsymbol{\xi}_{i,p}^{(2)} \right\rangle\right)\right| \cdot \sigma_\xi^2 \left(\frac{P}{\sqrt{d}} + \frac{1}{n}\right)\right).$$

Given that $d$ is sufficiently large under Condition 3.6, we have $P/\sqrt{d} = o(1/n)$. Thus, we obtain

$$\left|\Delta(\boldsymbol{\xi}_{i',p'}^{(2)})\right| \leq \lambda_c \widetilde{O}\left(\frac{\eta\sigma_0^2\sigma_\xi^4}{n}\right).$$

Under the condition that $n \geq \omega\left(\sigma_\xi^3\alpha_u^{-3}\right), \eta \leq O\left(1/\left(\sigma_0\alpha_u^3\right)\right)$ and $\lambda_c = 1$, this bound simplifies to

$$\left|\Delta(\boldsymbol{\xi}_{i',p'}^{(2)})\right| = o(\sigma_0\sigma_\xi).$$

We then proceed by induction to bound the projection $\left\langle \mathbf{w}_c^{(2)}(t), \boldsymbol{\xi}_{i',p'}^{(2)} \right\rangle$ across all steps. At $t = 0$, $\mathbf{w}_c^{(2)}(0) = \mathbf{w}_c^{(1)}(\widetilde{T}^{(1)})$ by the definition, Lemma 5.1 implies that

$$\left\langle \mathbf{w}_c^{(2)}(0), \boldsymbol{\xi}_{i',p'}^{(2)} \right\rangle \leq \widetilde{O}(\sigma_0\sigma_\xi).$$

Assume the inductive hypothesis holds at step $t$, i.e.,

$$\left\langle \mathbf{w}_c^{(2)}(t), \boldsymbol{\xi}_{i',p'}^{(2)} \right\rangle \leq \widetilde{O}(\sigma_0\sigma_\xi).$$

Then, by the bound derived earlier in the proof, we have

$$\left|\Delta(\boldsymbol{\xi}_{i',p'}^{(2)})\right| = o(\sigma_0\sigma_\xi).$$

Therefore, the updated projection satisfies

$$\left\langle \mathbf{w}_c^{(2)}(t+1), \boldsymbol{\xi}_{i',p'}^{(2)} \right\rangle = \left\langle \mathbf{w}_c^{(2)}(t), \boldsymbol{\xi}_{i',p'}^{(2)} \right\rangle + \Delta(\boldsymbol{\xi}_{i',p'}^{(2)}) \leq \widetilde{O}(\sigma_0\sigma_\xi) + o(\sigma_0\sigma_\xi) \leq \widetilde{O}(\sigma_0\sigma_\xi).$$

By induction, we conclude that for all $0 \leq t \leq T$,

$$\max_{c\in[C]} \left\langle \mathbf{w}_c^{(2)}(t), \boldsymbol{\xi}_{i',p'}^{(2)} \right\rangle \leq \widetilde{O}(\sigma_0\sigma_\xi), \quad \text{and} \quad \left|\Delta(\boldsymbol{\xi}_{i',p'}^{(2)})\right| \leq o(\sigma_0\sigma_\xi),$$

as claimed. This completes the proof. $\qquad\square$

Recall that the update of $\left\langle \mathbf{w}_c^{(\tau)}, \boldsymbol{\zeta}^{(\tau)} \right\rangle$ is $\Delta_c\left(\boldsymbol{\zeta}^{(\tau)}\right) = -\frac{\eta\lambda_c\|\boldsymbol{\zeta}^{(\tau)}\|_2^2\phi'(\langle\mathbf{w}_c^{(\tau)}(t),\boldsymbol{\zeta}^{(\tau)}\rangle)}{n}\mathcal{G}^{(\tau)}$. The direction of $\left\langle \mathbf{w}_c^{(2)}, \boldsymbol{\zeta}^{(2)} \right\rangle$ depends on the sign of $\mathcal{G}^{(2)}$, which is studied in the following lemma.

**Lemma E.2.** *In the second stage, let $h^{(2)}\left(\boldsymbol{\zeta}^{(2)}\right) = \sum_{c\in[C]} \phi\left(\left\langle \mathbf{w}_c^{(2)}, \boldsymbol{\zeta}^{(2)} \right\rangle\right)$, given $T \geq \widetilde{\Omega}\left(\frac{1}{\eta\sigma_0\alpha_u^3}\right)$, we suppose*

$$\max_{c\in[C]} \max_{\mathbf{e}\in\{\mathbf{u}^{(2)},\mathbf{v}^{(2)}\}} \left|\left\langle \mathbf{w}_c^{(2)}(t), \mathbf{e} \right\rangle\right| \leq O\left(C^{-1/3}\right)$$

*holds for all $t \leq T$. Then we have*

$$\begin{cases} \mathcal{G}^{(2)} < 0, & \text{if } h^{(2)}\left(\boldsymbol{\zeta}^{(2)}\right) \geq \widetilde{\Omega}\left(n^{-1/2} + CP\sigma_0^3\sigma_\xi^3\right), \\ \mathcal{G}^{(2)} > 0, & \text{if } h^{(2)}\left(\boldsymbol{\zeta}^{(2)}\right) \leq -\widetilde{\Omega}\left(n^{-1/2} + CP\sigma_0^3\sigma_\xi^3\right). \end{cases}$$

*Proof.* Since $\Delta_c\left(\boldsymbol{\zeta}^{(\tau)}\right) = \frac{\eta\lambda_c\left\|\boldsymbol{\zeta}^{(\tau)}\right\|_2^2 \phi'\left(\langle\mathbf{w}_c^{(\tau)}(t),\boldsymbol{\zeta}^{(\tau)}\rangle\right)}{n}\mathcal{G}^{(\tau)}$, we show the upper bound and lower bound for $\mathcal{G}^{(\tau)}$. Note that $\mathcal{I}^{(2)} = \mathcal{I}_=^{(2)} \cup \mathcal{I}_{\neq}^{(2)}$, we have

$$
\sum_{i\in\mathcal{I}_=^{(2)}} -\ell'(F(\mathbf{x}_i^{(2)}), y_i^{(2)}) - \sum_{i\in\mathcal{I}_{\neq}^{(2)}} \left(-\ell'(F(\mathbf{x}_i^{(2)}), y_i^{(2)})\right)
$$

$$
= \sum_{i\in\mathcal{I}_=^{(2)}} \left(1 + \exp\left(h^{(2)}\left(\mathbf{u}^{(2)}\right) + h^{(2)}\left(\mathbf{v}^{(2)}\right) + h^{(2)}\left(\boldsymbol{\zeta}^{(2)}\right) + y_i^{(2)}\sum_{p\in\mathcal{P}_i^\xi} h^{(2)}\left(\boldsymbol{\xi}_{i,p}^{(2)}\right)\right)\right)^{-1}
$$

$$
- \sum_{i\in\mathcal{I}_{\neq}^{(2)}} \left(1 + \exp\left(h^{(2)}\left(\mathbf{u}^{(2)}\right) + h^{(2)}\left(\mathbf{v}^{(2)}\right) - h^{(2)}\left(\boldsymbol{\zeta}^{(2)}\right) + y_i^{(2)}\sum_{p\in\mathcal{P}_i^\xi} h^{(2)}\left(\boldsymbol{\xi}_{i,p}^{(2)}\right)\right)\right)^{-1}
$$

$$
\leq \sum_{i\in\mathcal{I}_=^{(2)}} \left(1 + \exp\left(h^{(2)}\left(\mathbf{u}^{(2)}\right) + h^{(2)}\left(\mathbf{v}^{(2)}\right) + h^{(2)}\left(\boldsymbol{\zeta}^{(2)}\right) - CP\max_{c\in[C]i\in\mathcal{I}^{(2)},p\in\mathcal{P}_i^\xi}\phi\left(\left|\left\langle\mathbf{w}_c^{(2)},\boldsymbol{\xi}_{i,p}^{(2)}\right\rangle\right|\right)\right)\right)^{-1}
$$

$$
- \sum_{i\in\mathcal{I}_{\neq}^{(2)}} \left(1 + \exp\left(h^{(2)}\left(\mathbf{u}^{(2)}\right) + h^{(2)}\left(\mathbf{v}^{(2)}\right) - h^{(2)}\left(\boldsymbol{\zeta}^{(2)}\right) + CP\max_{c\in[C]i\in\mathcal{I}^{(2)},p\in\mathcal{P}_i^\xi}\phi\left(\left|\left\langle\mathbf{w}_c^{(2)},\boldsymbol{\xi}_{i,p}^{(2)}\right\rangle\right|\right)\right)\right)^{-1}
$$

$$
= n_=^{(2)} \left(1 + \exp\left(h^{(2)}\left(\mathbf{u}^{(2)}\right) + h^{(2)}\left(\mathbf{v}^{(2)}\right) + h^{(2)}\left(\boldsymbol{\zeta}^{(2)}\right) - CP\max_{c\in[C]i\in\mathcal{I}^{(2)},p\in\mathcal{P}_i^\xi}\phi\left(\left|\left\langle\mathbf{w}_c^{(2)},\boldsymbol{\xi}_{i,p}^{(2)}\right\rangle\right|\right)\right)\right)^{-1}
$$

$$
- n_{\neq}^{(2)} \left(1 + \exp\left(h^{(2)}\left(\mathbf{u}^{(2)}\right) + h^{(2)}\left(\mathbf{v}^{(2)}\right) - h^{(2)}\left(\boldsymbol{\zeta}^{(2)}\right) + CP\max_{c\in[C]i\in\mathcal{I}^{(2)},p\in\mathcal{P}_i^\xi}\phi\left(\left|\left\langle\mathbf{w}_c^{(2)},\boldsymbol{\xi}_{i,p}^{(2)}\right\rangle\right|\right)\right)\right)^{-1}
$$

Let $\gamma = \sum_{c\in[C]}\phi\left(\left\langle\mathbf{w}_c^{(2)},\boldsymbol{\zeta}^{(2)}\right\rangle\right) - CP\max_{c,i,p}\phi\left(\left|\left\langle\mathbf{w}_c^{(2)},\boldsymbol{\xi}_{i,p}^{(2)}\right\rangle\right|\right)$, we yield

$$
\sum_{i\in\mathcal{I}_=^{(2)}} -\ell'(F(\mathbf{x}_i^{(2)}), y_i^{(2)}) - \sum_{i\in\mathcal{I}_{\neq}^{(2)}} \left(-\ell'(F(\mathbf{x}_i^{(2)}), y_i^{(2)})\right)
$$

$$
\leq \frac{n_=^{(2)} - n_{\neq}^{(2)} + \exp\left(h^{(2)}\left(\mathbf{u}^{(2)}\right) + h^{(2)}\left(\mathbf{v}^{(2)}\right)\right)\left(n_=^{(2)}\exp\left(-\gamma\right) - n_{\neq}^{(2)}\exp\left(\gamma\right)\right)}{\left(1 + \exp\left(h^{(2)}\left(\mathbf{u}^{(2)}\right) + h^{(2)}\left(\mathbf{v}^{(2)}\right) + \gamma\right)\right)\left(1 + \exp\left(h^{(2)}\left(\mathbf{u}^{(2)}\right) + h^{(2)}\left(\mathbf{v}^{(2)}\right) - \gamma\right)\right)}
$$

$$
= \frac{\exp\left(-\gamma\right)\left(-n_{\neq}^{(2)}\exp\left(h^{(2)}\left(\mathbf{u}^{(2)}\right) + h^{(2)}\left(\mathbf{v}^{(2)}\right)\right)\exp\left(2\gamma\right) + n_=^{(2)}\exp\left(h^{(2)}\left(\mathbf{u}^{(2)}\right) + h^{(2)}\left(\mathbf{v}^{(2)}\right)\right) + \left(n_=^{(2)} - n_{\neq}^{(2)}\right)\exp\left(\gamma\right)\right)}{\left(1 + \exp\left(h^{(2)}\left(\mathbf{u}^{(2)}\right) + h^{(2)}\left(\mathbf{v}^{(2)}\right) + \gamma\right)\right)\left(1 + \exp\left(h^{(2)}\left(\mathbf{u}^{(2)}\right) + h^{(2)}\left(\mathbf{v}^{(2)}\right) - \gamma\right)\right)}.
$$

We fix the values of $h^{(2)}\left(\mathbf{u}^{(2)}\right)$ and $h^{(2)}\left(\mathbf{v}^{(2)}\right)$, and consider the equation

$$
-n_{\neq}^{(2)}\exp\left(h^{(2)}\left(\mathbf{u}^{(2)}\right) + h^{(2)}\left(\mathbf{v}^{(2)}\right)\right)\exp\left(2\gamma\right) + n_=^{(2)}\exp\left(h^{(2)}\left(\mathbf{u}^{(2)}\right) + h^{(2)}\left(\mathbf{v}^{(2)}\right)\right) + \left(n_=^{(2)} - n_{\neq}^{(2)}\right)\exp\left(\gamma\right) = 0,
$$

where the solution is given by

$$
\gamma^\star = \log\left(\frac{n_=^{(2)} - n_{\neq}^{(2)} + \sqrt{(n_=^{(2)} - n_{\neq}^{(2)})^2 + 4n_=^{(2)}n_{\neq}^{(2)}\exp\left(2h^{(2)}\left(\mathbf{u}^{(2)}\right) + 2h^{(2)}\left(\mathbf{v}^{(2)}\right)\right)}}{2n_{\neq}^{(2)}\exp\left(h^{(2)}\left(\mathbf{u}^{(2)}\right) + h^{(2)}\left(\mathbf{v}^{(2)}\right)\right)}\right).
$$

This implies that $\mathcal{G}^{(2)} > 0$ requires

$$
h^{(2)}\left(\boldsymbol{\zeta}^{(2)}\right) \leq \gamma^\star + CP\max_{c,i,p}\phi\left(\left|\left\langle\mathbf{w}_c^{(2)},\boldsymbol{\xi}_{i,p}^{(2)}\right\rangle\right|\right)
$$

as a sufficient condition. Since $\sqrt{a^2 + b^2} \leq |a| + |b|$, we further upper bound $\gamma^\star$ as

$$
\begin{aligned}
\gamma^\star &= \log\left( \frac{n_{=}^{(2)} - n_{\neq}^{(2)} + \sqrt{(n_{=}^{(2)} - n_{\neq}^{(2)})^2 + 4n_{=}^{(2)}n_{\neq}^{(2)} \exp\left(2h^{(2)}\left(\mathbf{u}^{(2)}\right) + 2h^{(2)}\left(\mathbf{v}^{(2)}\right)\right)}}{2n_{\neq}^{(2)} \exp\left(h^{(2)}\left(\mathbf{u}^{(2)}\right) + h^{(2)}\left(\mathbf{v}^{(2)}\right)\right)} \right) \\
&\leq \log\left( \frac{\left|n_{=}^{(2)} - n_{\neq}^{(2)}\right| + \sqrt{n_{=}^{(2)}n_{\neq}^{(2)} \exp\left(2h^{(2)}\left(\mathbf{u}^{(2)}\right) + 2h^{(2)}\left(\mathbf{v}^{(2)}\right)\right)}}{n_{\neq}^{(2)} \exp\left(h^{(2)}\left(\mathbf{u}^{(2)}\right) + h^{(2)}\left(\mathbf{v}^{(2)}\right)\right)} \right) \\
&\leq \log\left( \frac{\left|n_{=}^{(2)} - n_{\neq}^{(2)}\right|}{n_{\neq}^{(2)} \exp\left(h^{(2)}\left(\mathbf{u}^{(2)}\right) + h^{(2)}\left(\mathbf{v}^{(2)}\right)\right)} + \sqrt{\frac{n_{=}^{(2)}}{n_{\neq}^{(2)}}} \right) \\
&\leq \frac{\left|n_{=}^{(2)} - n_{\neq}^{(2)}\right|}{n_{\neq}^{(2)} \exp\left(h^{(2)}\left(\mathbf{u}^{(2)}\right) + h^{(2)}\left(\mathbf{v}^{(2)}\right)\right)} + \sqrt{\frac{n_{=}^{(2)}}{n_{\neq}^{(2)}} - 1}.
\end{aligned}
$$

Note that $\left|h^{(2)}\left(\mathbf{u}^{(2)}\right) + h^{(2)}\left(\mathbf{v}^{(2)}\right)\right| \leq O(1)$, by using Equations (3) and (4) in Lemma A.8, with a probability of $1 - O\left(\frac{1}{\text{poly}(n)}\right)$, we yield

$$
\frac{\left|n_{=}^{(2)} - n_{\neq}^{(2)}\right|}{n_{\neq}^{(2)} \exp\left(h^{(2)}\left(\mathbf{u}^{(2)}\right) + h^{(2)}\left(\mathbf{v}^{(2)}\right)\right)} \leq O\left( \frac{\left|n_{=}^{(2)} - n_{\neq}^{(2)}\right|}{n_{\neq}^{(2)}} \right) \leq \widetilde{O}\left(n^{-1/2}\right)
$$

and

$$
\sqrt{\frac{n_{=}^{(2)}}{n_{\neq}^{(2)}} - 1} \leq \sqrt{1 + \widetilde{O}\left(n^{-1/2}\right)} - 1 \leq \widetilde{O}\left(n^{-1/4}\right).
$$

Hence, we have

$$
\gamma^\star \leq \widetilde{O}\left(n^{-1/2}\right).
$$

Therefore, if

$$
h^{(2)}\left(\zeta^{(2)}\right) \geq \widetilde{\Omega}\left(n^{-1/2} + CP\sigma_0^3\sigma_\xi^3\right),
$$

we yield $\mathcal{G}^{(2)} < 0$.

On the other hand, we upper bound $-\mathcal{G}^{(2)}$ as

$$\sum_{i\in\mathcal{I}_{\neq}^{(2)}} -\ell'(F(\mathbf{x}_i^{(2)}), y_i^{(2)}) - \sum_{i\in\mathcal{I}_{=}^{(2)}} \left(-\ell'(F(\mathbf{x}_i^{(2)}), y_i^{(2)})\right)$$

$$= \sum_{i\in\mathcal{I}_{\neq}^{(2)}} \left(1 + \exp\left(h^{(2)}\left(\mathbf{u}^{(2)}\right) + h^{(2)}\left(\mathbf{v}^{(2)}\right) - h^{(2)}\left(\boldsymbol{\zeta}^{(2)}\right) + y_i \sum_{p\in\mathcal{P}_i^{\xi}} h^{(2)}\left(\boldsymbol{\xi}_{i,p}^{(2)}\right)\right)\right)^{-1}$$

$$- \sum_{i\in\mathcal{I}_{=}^{(2)}} \left(1 + \exp\left(h^{(2)}\left(\mathbf{u}^{(2)}\right) + h^{(2)}\left(\mathbf{v}^{(2)}\right) + h^{(2)}\left(\boldsymbol{\zeta}^{(2)}\right) + y_i \sum_{p\in\mathcal{P}_i^{\xi}} h^{(2)}\left(\boldsymbol{\xi}_{i,p}^{(2)}\right)\right)\right)^{-1}$$

$$\leq \sum_{i\in\mathcal{I}_{\neq}^{(2)}} \left(1 + \exp\left(h^{(2)}\left(\mathbf{u}^{(2)}\right) + h^{(2)}\left(\mathbf{v}^{(2)}\right) - h^{(2)}\left(\boldsymbol{\zeta}^{(2)}\right) - CP \max_{c\in[C]i\in\mathcal{I}^{(2)}, p\in\mathcal{P}_i^{\xi}} \phi\left(\left|\left\langle \mathbf{w}_c^{(2)}, \boldsymbol{\xi}_{i,p}^{(2)}\right\rangle\right|\right)\right)\right)^{-1}$$

$$- \sum_{i\in\mathcal{I}_{=}^{(2)}} \left(1 + \exp\left(h^{(2)}\left(\mathbf{u}^{(2)}\right) + h^{(2)}\left(\mathbf{v}^{(2)}\right) + h^{(2)}\left(\boldsymbol{\zeta}^{(2)}\right) + CP \max_{c\in[C]i\in\mathcal{I}^{(2)}, p\in\mathcal{P}_i^{\xi}} \phi\left(\left|\left\langle \mathbf{w}_c^{(2)}, \boldsymbol{\xi}_{i,p}^{(2)}\right\rangle\right|\right)\right)\right)^{-1}$$

$$= n_{\neq}^{(2)} \left(1 + \exp\left(h^{(2)}\left(\mathbf{u}^{(2)}\right) + h^{(2)}\left(\mathbf{v}^{(2)}\right) - h^{(2)}\left(\boldsymbol{\zeta}^{(2)}\right) - CP \max_{c\in[C]i\in\mathcal{I}^{(2)}, p\in\mathcal{P}_i^{\xi}} \phi\left(\left|\left\langle \mathbf{w}_c^{(2)}, \boldsymbol{\xi}_{i,p}^{(2)}\right\rangle\right|\right)\right)\right)^{-1}$$

$$- n_{=}^{(2)} \left(1 + \exp\left(h^{(2)}\left(\mathbf{u}^{(2)}\right) + h^{(2)}\left(\mathbf{v}^{(2)}\right) + h^{(2)}\left(\boldsymbol{\zeta}^{(2)}\right) + CP \max_{c\in[C]i\in\mathcal{I}^{(2)}, p\in\mathcal{P}_i^{\xi}} \phi\left(\left|\left\langle \mathbf{w}_c^{(2)}, \boldsymbol{\xi}_{i,p}^{(2)}\right\rangle\right|\right)\right)\right)^{-1}$$

Let $\tilde{\gamma} = -\sum_{c\in[C]} \phi\left(\left\langle \mathbf{w}_c^{(2)}, \boldsymbol{\zeta}^{(2)}\right\rangle\right) - CP \max_{c,i,p} \phi\left(\left|\left\langle \mathbf{w}_c^{(2)}, \boldsymbol{\xi}_{i,p}^{(2)}\right\rangle\right|\right)$, we yield

$$\sum_{i\in\mathcal{I}_{\neq}^{(2)}} -\ell'(F(\mathbf{x}_i^{(2)}), y_i^{(2)}) - \sum_{i\in\mathcal{I}_{=}^{(2)}} \left(-\ell'(F(\mathbf{x}_i^{(2)}), y_i^{(2)})\right)$$

$$\leq \frac{n_{\neq}^{(2)} - n_{=}^{(2)} + \exp\left(h^{(2)}\left(\mathbf{u}^{(2)}\right) + h^{(2)}\left(\mathbf{v}^{(2)}\right)\right)\left(n_{\neq}^{(2)} \exp\left(-\tilde{\gamma}\right) - n_{=}^{(2)} \exp\left(\tilde{\gamma}\right)\right)}{\left(1 + \exp\left(h^{(2)}\left(\mathbf{u}^{(2)}\right) + h^{(2)}\left(\mathbf{v}^{(2)}\right) + \tilde{\gamma}\right)\right)\left(1 + \exp\left(h^{(2)}\left(\mathbf{u}^{(2)}\right) + h^{(2)}\left(\mathbf{v}^{(2)}\right) - \tilde{\gamma}\right)\right)}$$

$$= \frac{\exp\left(-\tilde{\gamma}\right)\left(-n_{=}^{(2)} \exp\left(h^{(2)}\left(\mathbf{u}^{(2)}\right) + h^{(2)}\left(\mathbf{v}^{(2)}\right)\right)\exp\left(2\tilde{\gamma}\right) + n_{\neq}^{(2)} \exp\left(h^{(2)}\left(\mathbf{u}^{(2)}\right) + h^{(2)}\left(\mathbf{v}^{(2)}\right)\right) + \left(n_{\neq}^{(2)} - n_{=}^{(2)}\right)\exp\left(\tilde{\gamma}\right)\right)}{\left(1 + \exp\left(h^{(2)}\left(\mathbf{u}^{(2)}\right) + h^{(2)}\left(\mathbf{v}^{(2)}\right) + \tilde{\gamma}\right)\right)\left(1 + \exp\left(h^{(2)}\left(\mathbf{u}^{(2)}\right) + h^{(2)}\left(\mathbf{v}^{(2)}\right) - \tilde{\gamma}\right)\right)}.$$

Similarly, we fix the values of $h^{(2)}\left(\mathbf{u}^{(2)}\right)$ and $h^{(2)}\left(\mathbf{v}^{(2)}\right)$, and consider the equation

$$-n_{=}^{(2)} \exp\left(h^{(2)}\left(\mathbf{u}^{(2)}\right) + h^{(2)}\left(\mathbf{v}^{(2)}\right)\right)\exp\left(2\gamma\right) + n_{\neq}^{(2)} \exp\left(h^{(2)}\left(\mathbf{u}^{(2)}\right) + h^{(2)}\left(\mathbf{v}^{(2)}\right)\right) + \left(n_{\neq}^{(2)} - n_{=}^{(2)}\right)\exp\left(\gamma\right) = 0,$$

where the solution is given by

$$\tilde{\gamma}^{\star} = \log\left(\frac{n_{\neq}^{(2)} - n_{=}^{(2)} + \sqrt{(n_{\neq}^{(2)} - n_{=}^{(2)})^2 + 4n_{\neq}^{(2)} n_{=}^{(2)} \exp\left(2h^{(2)}\left(\mathbf{u}^{(2)}\right) + 2h^{(2)}\left(\mathbf{v}^{(2)}\right)\right)}}{2n_{=}^{(2)} \exp\left(h^{(2)}\left(\mathbf{u}^{(2)}\right) + h^{(2)}\left(\mathbf{v}^{(2)}\right)\right)}\right)$$

$$\tag{23}$$

We then use a similar technique in bounding $\gamma^{\star}$. Since $\left|h^{(2)}\left(\mathbf{u}^{(2)}\right) + h^{(2)}\left(\mathbf{v}^{(2)}\right)\right| \leq O(1)$, by using Equations (3) and (4) in Lemma A.8, with a probability of $1 - O\left(\frac{1}{\text{poly}(n)}\right)$, we yield

$$\tilde{\gamma}^{\star} \leq \frac{\left|n_{=}^{(2)} - n_{\neq}^{(2)}\right|}{n_{=}^{(2)} \exp\left(h^{(2)}\left(\mathbf{u}^{(2)}\right) + h^{(2)}\left(\mathbf{v}^{(2)}\right)\right)} + \sqrt{\frac{n_{\neq}^{(2)}}{n_{=}^{(2)}}} - 1 \leq \widetilde{O}\left(n^{-1/2}\right) + \sqrt{1 + \widetilde{O}\left(n^{-1/2}\right)} - 1 \leq \widetilde{O}\left(n^{-1/2}\right).$$

This implies that $\mathcal{G}^{(2)} < 0$ requires

$$h^{(2)}\left(\boldsymbol{\zeta}^{(2)}\right) \geq -\tilde{\gamma}^\star - CP\max_{c,i,p}\phi\left(\left|\left\langle \mathbf{w}_c^{(2)}, \boldsymbol{\xi}_{i,p}^{(2)}\right\rangle\right|\right) \geq -\widetilde{O}\left(n^{-1/2} + CP\sigma_0^3\sigma_\xi^3\right)$$

as a sufficient condition. Therefore, if $h^{(2)}\left(\boldsymbol{\zeta}^{(2)}\right) \leq -\widetilde{\Omega}\left(n^{-1/2} + CP\sigma_0^3\sigma_\xi^3\right)$, then it follows that $\mathcal{G}^{(2)} > 0$.

We conclude our proof. $\qquad\square$

**Lemma E.3.** *Under Condition 3.6, given $T \geq \widetilde{\Omega}\left(\frac{1}{\eta\sigma_0\alpha_u^3}\right)$, we suppose*

$$\max_{c\in[C]}\max_{\mathbf{e}\in\left\{\mathbf{u}^{(2)},\mathbf{v}^{(2)}\right\}}\left|\left\langle \mathbf{w}_c^{(2)}(t), \mathbf{e}\right\rangle\right| \leq O\left(C^{-1/3}\right)$$

*holds for all $t \leq T$. We let $\alpha_\zeta = \Theta\left(\alpha_u\right)$. Then there exists $T_\zeta^{(2)} \leq \widetilde{O}\left(\frac{1}{\eta\sigma_0\alpha_u^3}\right)$ such that*

$$\max_{c\in[C]}\left\langle \mathbf{w}_c^{(2)}(T_\zeta^{(2)}), \boldsymbol{\zeta}^{(2)}\right\rangle \leq o\left(1\right), \qquad \left|h^{(2)}(\boldsymbol{\zeta}^{(2)})\right| \leq o(1).$$

*Proof.* At $t = 0$, when $\alpha_\zeta = \Theta\left(\alpha_u\right)$, By using Corollary 5.3, we yield

$$\sum_{c\in[C]}\phi\left(\left\langle \mathbf{w}_c^{(2)}, \boldsymbol{\zeta}^{(2)}\right\rangle\right) \geq \Theta\left(\alpha_\zeta^3\alpha_u^{-3}C^{-1}\right) = \widetilde{\Theta}\left(1\right), CP\max_{c\in[C],i\in\mathcal{I}_=,p\in\mathcal{P}_i^\xi}\phi\left(\left|\left\langle \mathbf{w}_c^{(2)}, \boldsymbol{\xi}_{i,p}^{(2)}\right\rangle\right|\right) \leq O\left(CP\sigma_0^3\sigma_\xi^3\right) \leq o\left(1\right).$$

We then have

$$\sum_{c\in[C]}\phi\left(\left\langle \mathbf{w}_c^{(2)}, \boldsymbol{\zeta}^{(2)}\right\rangle\right) \geq \omega\left(n^{-1/2} + CP\sigma_0^3\sigma_\xi^3\right),$$

which means $\Delta_c\left(\boldsymbol{\zeta}^{(2)}\right) \leq 0$ due to Lemma E.2. By rewriting $\mathcal{G}^{(2)}$, we yield

$$\mathcal{G}^{(2)} = \sum_{i\in\mathcal{I}_=^{(2)}}-\ell'(F(\mathbf{x}_i), y_i) - \sum_{i\in\mathcal{I}_{\neq}^{(2)}}(-\ell'(F(\mathbf{x}_i), y_i))$$

$$= \sum_{i\in\mathcal{I}_=^{(2)}}\left(1 + \exp\left(h^{(2)}\left(\mathbf{u}^{(2)}\right) + h^{(2)}\left(\mathbf{v}^{(2)}\right) + h^{(2)}\left(\boldsymbol{\zeta}^{(2)}\right) + y_i\sum_{p\in\mathcal{P}_i^\xi}h^{(2)}\left(\boldsymbol{\xi}_{i,p}^{(2)}\right)\right)\right)^{-1}$$

$$- \sum_{i\in\mathcal{I}_{\neq}^{(2)}}\left(1 + \exp\left(h^{(2)}\left(\mathbf{u}^{(2)}\right) + h^{(2)}\left(\mathbf{v}^{(2)}\right) - h^{(2)}\left(\boldsymbol{\zeta}^{(2)}\right) + y_i\sum_{p\in\mathcal{P}_i^\xi}h^{(2)}\left(\boldsymbol{\xi}_{i,p}^{(2)}\right)\right)\right)^{-1}$$

By the assumption, for any $i$, we have

$$\left|h^{(2)}\left(\mathbf{u}^{(2)}\right) + h^{(2)}\left(\mathbf{v}^{(2)}\right) + y_i\sum_{p\in\mathcal{P}_i^\xi}h^{(2)}\left(\boldsymbol{\xi}_{i,p}^{(2)}\right)\right| \leq O(1).$$

At the beginning of the second stage, we have $h^{(2)}(\boldsymbol{\zeta}^{(2)}) \geq \widetilde{\Omega}\left(\left(\max_{c\in[C]}\left\langle \mathbf{w}_c^{(2)}(t), \boldsymbol{\zeta}^{(2)}\right\rangle\right)^3\right) \geq \widetilde{\Omega}\left(1\right)$. Then, for some $t$, if $h^{(2)}(\boldsymbol{\zeta}^{(2)}) \geq \widetilde{\Omega}\left(1\right)$, it is clear to show that there must exist $\alpha = \Theta\left(1\right)$ such that for all $\alpha_\zeta \geq \alpha$, we have

$$-O\left(n\right) \leq \mathcal{G}^{(2)} \leq -\Omega\left(n\right).$$

We consider the update rule for $\max_{c \in [C]} \left\langle \mathbf{w}_c^{(2)}(t), \boldsymbol{\zeta}^{(2)} \right\rangle$:

$$\left\langle \mathbf{w}_c^{(2)}(t+1), \boldsymbol{\zeta}^{(2)} \right\rangle - \left\langle \mathbf{w}_c^{(2)}(t), \boldsymbol{\zeta}^{(2)} \right\rangle = -\Theta \left( \eta \lambda_c \left\| \boldsymbol{\zeta}^{(2)} \right\|_2^2 \phi' \left( \left\langle \mathbf{w}_c^{(2)}(t), \boldsymbol{\zeta}^{(2)} \right\rangle \right) \right). \tag{24}$$

This shows that $\left\langle \mathbf{w}_c^{(2)}(t), \boldsymbol{\zeta}^{(2)} \right\rangle$ decreases monotonically for all $c \in [C]$. Consequently, $h^{(2)}(\boldsymbol{\zeta}^{(2)})$ also decreases. Let $T_\zeta^{(2)}$ be the first time $t$ such that $h^{(2)}(\boldsymbol{\zeta}^{(2)})$ drops below $\widetilde{\Omega}(1)$. Then, the monotonic decrease described in Equation (24) holds for any $0 \leq T^{(2)} < T_\zeta^{(2)}$.

Given $0 \leq t' < t \leq T_\zeta^{(2)}$, starting from some $\left\langle \mathbf{w}_c^{(2)}(t'), \boldsymbol{\zeta}^{(2)} \right\rangle$, the number of iterations it takes to reach $\max_{c \in [C]} \left\langle \mathbf{w}_c^{(2)}(t), \boldsymbol{\zeta}^{(2)} \right\rangle \leq \frac{1}{2} \max_{c \in [C]} \left\langle \mathbf{w}_c^{(2)}(t'), \boldsymbol{\zeta}^{(2)} \right\rangle$ is at most $O \left( \frac{\max_{c \in [C]} \left\langle \mathbf{w}_c^{(2)}(t'), \boldsymbol{\zeta}^{(2)} \right\rangle}{\eta \alpha_\zeta^2 \left( \max_{c \in [C]} \left\langle \mathbf{w}_c^{(2)}(t'), \boldsymbol{\zeta}^{(2)} \right\rangle \right)^2} \right)$. Then, starting from $\widetilde{\Theta}(1)$, it takes at most

$$\widetilde{O} \left( \sum_{i=0}^{r-1} \frac{2^{-i}}{\eta \alpha_\zeta^2 2^{-2i}} \right) \leq \widetilde{O} \left( \frac{2^r}{\eta \alpha_\zeta^2} \right). \tag{25}$$

time steps to reach $\max_{c \in [C]} \left\langle \mathbf{w}_c^{(2)}(t), \boldsymbol{\zeta}^{(2)} \right\rangle \leq 2^{-r} \max_{c \in [C]} \left\langle \mathbf{w}_c^{(2)}(t'), \boldsymbol{\zeta}^{(2)} \right\rangle$. Setting $r = \left\lceil \frac{\log(1/\sigma_0 \alpha_u)}{\log(2)} \right\rceil$, we obtain $T_\zeta^{(2)} \leq \widetilde{O} \left( \frac{1}{\eta \sigma_0 \alpha_u^3} \right)$ such that

$$\max_{c \in [C]} \left\langle \mathbf{w}_c^{(2)}(T_\zeta^{(2)}), \boldsymbol{\zeta}^{(2)} \right\rangle \leq o \left( C^{-1/3} \right) = o(1), \quad h^{(2)}(\boldsymbol{\zeta}^{(2)}) \leq o(1).$$

Then, the update magnitude $\left| \left\langle \mathbf{w}_c^{(2)}(t+1), \boldsymbol{\zeta}^{(2)} \right\rangle - \left\langle \mathbf{w}_c^{(2)}(t), \boldsymbol{\zeta}^{(2)} \right\rangle \right|$ is small, and according to Lemma E.2, $h^{(2)}(\boldsymbol{\zeta}^{(2)})$ cannot decrease to $-\Omega(1)$ because we have $\mathcal{G}^{(2)} > 0$ if $h^{(2)} \left( \boldsymbol{\zeta}^{(2)} \right) \leq -\widetilde{\Omega} \left( n^{-1/2} + CP\sigma_0^3 \sigma_\xi^3 \right)$. Hence, we conclude that

$$\left| h^{(2)}(\boldsymbol{\zeta}^{(2)}) \right| \leq o(1),$$

at some $T_\zeta^{(2)} \leq \widetilde{O} \left( \frac{1}{\eta \sigma_0 \alpha_u^3} \right)$. $\qquad \square$

We then analyze the change of $\left\langle \mathbf{w}_c^{(2)}, \mathbf{u}^{(2)} \right\rangle$ and $\left\langle \mathbf{w}_c^{(2)}, \mathbf{v}^{(2)} \right\rangle$ in the second task.

**Lemma E.4** (Learning the Task-Specific and General Features). *Under Condition 3.6, starting from $T^{(2)} = 0$, there exists $T_u^{(2)} \leq \widetilde{O} \left( \frac{1}{\eta \sigma_0 \alpha_u^3} \right)$ such that*

$$\max_{c \in [C]} \left\langle \mathbf{w}_c^{(2)}(T_u^{(2)}), \mathbf{u}^{(2)} \right\rangle \geq \Omega \left( C^{-1/3} \right),$$

$$\max_{c \in [C]} \left\langle \mathbf{w}_c^{(2)}(T_u^{(2)}), \mathbf{v}^{(2)} \right\rangle \leq \widetilde{O} \left( \sigma_0 \alpha_v \right).$$

*Proof.* At $T^{(2)} = 0$, for all $c \in [C]$, we have

$$\left\langle \mathbf{w}_c^{(2)}(0), \mathbf{u}^{(1)} \right\rangle \geq \left\langle \mathbf{w}_c^{(1)}(0), \mathbf{u}^{(1)} \right\rangle \geq -\widetilde{O} \left( \sigma_0 \alpha_u \right),$$

where the first inequality follows that $\left\{ \left\langle \mathbf{w}_c^{(1)}(t), \mathbf{u}^{(1)} \right\rangle \right\}_{t=0}^{T^{(1)}}$ is an increasing sequence. The second inequality holds because $\mathbf{w}_c^{(1)}(0)$ is drawn from an isotropic Gaussian distribution and Lemma 4.1. It follows that

$$\forall c \in [C], \left\langle \mathbf{w}_c^{(2)}(0), \boldsymbol{\zeta}^{(2)} \right\rangle = \alpha_z \alpha_u^{-1} \left\langle \mathbf{w}_c^{(2)}(0), \mathbf{u}^{(1)} \right\rangle \geq -\widetilde{O} \left( \sigma_0 \alpha_\zeta \right).$$

and similarly,

$$\forall c \in [C], \left\langle \mathbf{w}_c^{(2)}(0), \mathbf{v}^{(2)} \right\rangle = \left\langle \mathbf{w}_c^{(2)}(0), \mathbf{v}^{(1)} \right\rangle \geq -\widetilde{O}\left(\sigma_0 \alpha_v\right), \tag{26}$$

where the last inequality uses the monotonicity of $\left\{ \left\langle \mathbf{w}_c^{(1)}(t), \mathbf{v}^{(1)} \right\rangle \right\}_{t=0}^{T^{(1)}}$ and Lemma 4.1.

By Equation (18) and Corollary 5.3, we obtain

$$\max_{c \in [C]} \max_{\mathbf{e} \in \{\mathbf{u}^{(2)}, \mathbf{v}^{(2)}\}} \left| \left\langle \mathbf{w}_c^{(2)}(0), \mathbf{e} \right\rangle \right| \leq O\left(C^{-1/3}\right).$$

We proceed by induction. Suppose that for some $t > 0$, we have

$$\max_{c \in [C]} \max_{\mathbf{e} \in \{\mathbf{u}^{(2)}, \mathbf{v}^{(2)}\}} \left| \left\langle \mathbf{w}_c^{(2)}(t), \mathbf{e} \right\rangle \right| \leq O\left(C^{-1/3}\right), \quad \text{and} \quad \left| h^{(2)}\left(\boldsymbol{\zeta}^{(2)}\right) \right| \leq \widetilde{O}(1).$$

Then by Lemma E.3, $\left| h^{(2)}\left(\boldsymbol{\zeta}^{(2)}\right) \right|$ is non-increasing, and at $t + 1$, we still have

$$\left| h^{(2)}\left(\boldsymbol{\zeta}^{(2)}\right) \right| \leq \widetilde{O}(1).$$

This ensures that

$$\Omega(1) \leq -\ell_i' \leq 1, \quad \forall i \in [n],$$

which further implies

$$\max_{c \in C} \left\langle \mathbf{w}_c^{(2)}(t+1), \mathbf{v}^{(2)} \right\rangle - \max_{c \in C} \left\langle \mathbf{w}_c^{(2)}(t), \mathbf{v}^{(2)} \right\rangle = \Theta\left( n^{-1} \eta \alpha_v^2 \left( \sum_{i=1}^n \ell_i' \right) \phi'\left( \left| \left\langle \mathbf{w}_c^{(2)}(t), \mathbf{v}^{(2)} \right\rangle \right| \right) \right).$$

It shows that $\left\{ \max_{c \in [C]} \left\langle \mathbf{w}_c^{(2)}(t), \mathbf{v}^{(2)} \right\rangle \right\}$ is an increasing sequence, hence

$$\max_{c \in [C]} \left\langle \mathbf{w}_c^{(2)}(t+1), \mathbf{v}^{(2)} \right\rangle \geq \max_{c \in [C]} \left\langle \mathbf{w}_c^{(2)}(t), \mathbf{v}^{(2)} \right\rangle \geq \widetilde{\Omega}(\sigma_0 \alpha_v). \tag{27}$$

Since $\max_{c \in [C]} \left\langle \mathbf{w}_c^{(2)}(t), \mathbf{v}^{(2)} \right\rangle \leq \widetilde{O}(\sigma_0 \alpha_v)$, $\eta \leq O\left(1/\left(\sigma_0 \alpha_u^3\right)\right)$, and $\alpha_v \leq o(\alpha_u)$, it follows that

$$\max_{c \in [C]} \left\langle \mathbf{w}_c^{(2)}(t+1), \mathbf{v}^{(2)} \right\rangle \leq \max_{c \in [C]} \left\langle \mathbf{w}_c^{(2)}(t), \mathbf{v}^{(2)} \right\rangle + \Theta(\eta \alpha_v^4 \sigma_0^2) \leq \widetilde{O}(\sigma_0 \alpha_v) + o(\sigma_0 \alpha_v) \leq \widetilde{O}(\sigma_0 \alpha_v). \tag{28}$$

Combining Equations (27) and (28), we yield

$$\max_{c \in [C]} \left\langle \mathbf{w}_c^{(2)}(t+1), \mathbf{v}^{(2)} \right\rangle = \widetilde{\Theta}(\sigma_0 \alpha_v). \tag{29}$$

Thus, combining Equations (26) and (29), we maintain

$$\max_{c \in [C]} \left| \left\langle \mathbf{w}_c^{(2)}(t+1), \mathbf{v}^{(2)} \right\rangle \right| \leq \widetilde{O}\left(\sigma_0 \alpha_v\right).$$

Meanwhile, for $\mathbf{u}^{(2)}$ we have

$$\max_{c \in C} \left\langle \mathbf{w}_c^{(2)}(t+1), \mathbf{u}^{(2)} \right\rangle - \max_{c \in C} \left\langle \mathbf{w}_c^{(2)}(t), \mathbf{u}^{(2)} \right\rangle$$
$$= \Theta\left( n^{-1} \eta \alpha_u^2 \left( \sum_{i=1}^n -\ell_i' \right) \phi'\left( \left| \left\langle \mathbf{w}_c^{(2)}(t), \mathbf{u}^{(2)} \right\rangle \right| \right) \right) = \Theta\left( \eta \alpha_u^2 \phi'\left( \left| \left\langle \mathbf{w}_c^{(2)}(t), \mathbf{u}^{(2)} \right\rangle \right| \right) \right).$$

It shows that $\left\{\max_{c\in[C]}\left\langle\mathbf{w}_c^{(2)}(t),\mathbf{u}^{(2)}\right\rangle\right\}$ is an increasing sequence. Let $T_u^{(2)}$ be the first time $t$ such that $\max_{c\in C}\left\langle\mathbf{w}_c^{(2)}(t),\mathbf{u}^{(2)}\right\rangle\geq\Omega\left(C^{-1/3}\right)$. Then for all $0\leq T^{(2)}<T_u^{(2)}$, we have

$$\max_{c\in[C]}\left|\left\langle\mathbf{w}_c^{(2)}(T^{(2)}),\mathbf{v}^{(2)}\right\rangle\right|\leq\widetilde{O}\left(\sigma_0\alpha_v\right),\quad\text{and}\quad\left|h^{(2)}\left(\boldsymbol{\zeta}^{(2)}\right)\right|\leq\widetilde{O}\left(1\right).$$

and

$$\max_{c\in C}\left\langle\mathbf{w}_c^{(2)}(T^{(2)}+1),\mathbf{u}^{(2)}\right\rangle-\max_{c\in C}\left\langle\mathbf{w}_c^{(2)}(T^{(2)}),\mathbf{u}^{(2)}\right\rangle=\Theta\left(\eta\alpha_u^2\phi'\left(\left|\left\langle\mathbf{w}_c^{(2)}(T^{(2)}),\mathbf{u}^{(2)}\right\rangle\right|\right)\right).\tag{30}$$

Corollary 5.3 implies that $\max_{c\in[C]}\left\langle\mathbf{w}_c^{(2)}(0),\mathbf{u}^{(2)}\right\rangle=\widetilde{\Theta}\left(\sigma_0\alpha_u\right)$. Starting from some $\max_{c\in C}\left\langle\mathbf{w}_c^{(2)}(t'),\mathbf{u}^{(2)}\right\rangle$, the number of iterations it takes to reach $\max_{c\in C}\left\langle\mathbf{w}_c^{(2)}(t),\mathbf{u}^{(2)}\right\rangle\geq2\max_{c\in C}\left\langle\mathbf{w}_c^{(2)}(t'),\mathbf{u}^{(2)}\right\rangle$ is at most $O\left(\frac{\max_{c\in C}\left\langle\mathbf{w}_c^{(2)}(t'),\mathbf{u}^{(2)}\right\rangle}{\eta\alpha_u^2\left(\max_{c\in C}\left\langle\mathbf{w}_c^{(2)}(t'),\mathbf{u}^{(2)}\right\rangle\right)^2}\right)$. Then, starting from $\widetilde{\Theta}\left(\sigma_0\alpha_u\right)$, it takes at most

$$T_u^{(2)}\leq\widetilde{O}\left(\sum_{i=0}^{\infty}\frac{2^i\sigma_0\alpha_u}{\eta\alpha_u^2\left(2^i\sigma_0\alpha_u\right)^2}\right)\leq\widetilde{O}\left(\frac{1}{\eta\sigma_0\alpha_u^3}\right)\tag{31}$$

times steps to reach $\max_{c\in C}\left\langle\mathbf{w}_c^{(2)}(t),\mathbf{u}^{(2)}\right\rangle\geq\Omega\left(C^{-1/3}\right)$.

This completes the proof. □

After training the second task, the following theorem and corollary show that at the end of the second stage, CF occurs if $\alpha_\zeta\geq\Omega\left(\alpha_u\right)$, i.e., at $\widetilde{T}^{(2)}=\widetilde{\Theta}\left(\frac{1}{\eta\sigma_0\alpha_u^3}\right)$, only $\mathbf{u}^{(2)}$ can be used for classification.

**Theorem 5.4.** *In the second stage, given a training set $\mathcal{S}_{tr}^{(2)}$ with size $n$, there exists $\widetilde{T}^{(2)}=\widetilde{\Theta}\left(\frac{1}{\eta\sigma_0\alpha_u^3}\right)$ such that for $T^{(2)}\geq\widetilde{T}^{(2)}$, the network $F_{T^{(2)}}$ fits all training data points with a high probability:*

$$\mathbb{P}\left[\forall i\in\mathcal{S}_{tr}^{(2)},y_iF_{T^{(2)}}^{(2)}(\mathbf{x}_i)\geq\widetilde{\Omega}(1)\right]\geq1-O\left(\frac{n^2P^2C}{poly(d)}+\frac{1}{poly(n)}\right).$$

*Moreover, $F_{T^{(2)}}$ achieves a high accuracy on test data sampled from the second task:*

$$\mathbb{P}_{(\mathbf{x},y)\sim\mathcal{D}_{\mathbf{z}}^{(2)}}\left[yF_{T^{(2)}}^{(2)}(\mathbf{x})>0\right]\geq1-O\left(\frac{nP^2C}{poly(d)}+\frac{1}{poly(n)}\right).$$

*If $\alpha_v\leq o\left(\alpha_u\right)$, and $\alpha_\zeta\geq\Omega\left(\alpha_u\right)$, $F_{T^{(2)}}$ achieves a low accuracy on test data sampled from the first task*

$$\mathbb{P}_{(\mathbf{x},y)\sim\mathcal{D}_{\mathbf{z}}^{(1)}}\left[yF_{T^{(2)}}^{(2)}(\mathbf{x})>0\right]\leq\frac{1}{2}+O\left(\frac{nP^2C}{poly(d)}+\frac{1}{poly(n)}\right).$$

*Proof.* For any training sample $(\mathbf{x}_i,y_i)$ from the second task, the output can be rewritten as:

$$\begin{aligned}
y_iF_{T^{(2)}}(\mathbf{x}_i)=&y_i\sum_{c=1}^{C}\lambda_c\phi\left(\left\langle\mathbf{w}_c^{(2)},y_i\mathbf{u}^{(2)}\right\rangle\right)+y_i\sum_{c=1}^{C}\lambda_c\phi\left(\left\langle\mathbf{w}_c^{(2)},y_i\mathbf{v}^{(2)}\right\rangle\right)\\
&+y_i\sum_{c=1}^{C}\lambda_c\phi\left(\left\langle\mathbf{w}_c^{(2)},\epsilon_i\boldsymbol{\zeta}^{(2)}\right\rangle\right)+y_i\sum_{c=1}^{C}\sum_{p\in\mathcal{P}_i^\xi}\phi(\left\langle\mathbf{w}_c^{(2)},\boldsymbol{\xi}_{i,p}^{(2)}\right\rangle)\\
=&\sum_{c=1}^{C}\lambda_c\phi\left(\left\langle\mathbf{w}_c^{(2)},\mathbf{u}^{(2)}\right\rangle\right)+\sum_{c=1}^{C}\lambda_c\phi\left(\left\langle\mathbf{w}_c^{(2)},\mathbf{v}\right\rangle\right)\\
&+y_i\epsilon_i\sum_{c=1}^{C}\lambda_c\phi\left(\left\langle\mathbf{w}_c^{(2)},\boldsymbol{\zeta}^{(2)}\right\rangle\right)+y_i\sum_{c=1}^{C}\sum_{p\in\mathcal{P}_i^\xi}\phi\left(\left\langle\mathbf{w}_c^{(2)},\boldsymbol{\xi}_{i,p}^{(2)}\right\rangle\right)
\end{aligned}$$

By using Lemmas E.3 and E.4, there exists some time $\widetilde{T}^{(2)} = \widetilde{\Theta}\left(\frac{1}{\eta\sigma_0\alpha_u^3}\right)$ such that for $T^{(2)} \geq \widetilde{T}^{(2)}$, we have

$$
y_i F_{T^{(2)}}(\mathbf{x}_i) \geq \max_{c\in[C]} \lambda_c \phi\left(\left\langle \mathbf{w}_c^{(2)}, \mathbf{u}^{(2)} \right\rangle\right) + (C-1)\min_{c\in[C]} \lambda_c \phi\left(\left\langle \mathbf{w}_c^{(2)}, \mathbf{u}^{(2)} \right\rangle\right)
$$

$$
- C\max_{c\in[C]} \lambda_c \phi\left(\left|\left\langle \mathbf{w}_c^{(2)}, \mathbf{v}^{(2)} \right\rangle\right|\right) - \sum_{c=1}^{C} \lambda_c \phi\left(\left|\left\langle \mathbf{w}_c^{(2)}, \boldsymbol{\zeta}^{(2)} \right\rangle\right|\right) - CP\max_{c\in[C]}\max_{p\in\mathcal{P}_i^\xi} \phi\left(\left|\left\langle \mathbf{w}_c^{(2)}, \boldsymbol{\xi}_{i,p}^{(2)} \right\rangle\right|\right)
$$

$$
\geq \Omega(1/C) - CO\left(\sigma_0^3\alpha_u^3\right) - CO\left(\sigma_0^3\alpha_v^3\right) - Co(1) - CPO\left(\sigma_0^3\sigma_\xi^3\right) \geq \widetilde{\Omega}(1). \tag{32}
$$

The probability of Equation (32) hold is $1 - O\left(\frac{n^2P^2C}{\text{poly}(d)} + \frac{1}{\text{poly}(n)}\right)$. For a test sample $(\mathbf{x}, y)$ drawn from the second task, with a probability of $1 - O\left(\frac{nP^2C}{\text{poly}(d)} + \frac{1}{\text{poly}(n)}\right)$, we have

$$
yF_{T^{(2)}}(\mathbf{x}) = y\sum_{c=1}^{C} \lambda_c \phi\left(\left\langle \mathbf{w}_c^{(2)}, y\mathbf{u}^{(2)} \right\rangle\right) + y\sum_{c=1}^{C} \lambda_c \phi\left(\left\langle \mathbf{w}_c^{(2)}, y\mathbf{v}^{(2)} \right\rangle\right)
$$

$$
+ y\sum_{c=1}^{C} \lambda_c \phi\left(\left\langle \mathbf{w}_c^{(2)}, \epsilon\boldsymbol{\zeta}^{(2)} \right\rangle\right) + y\sum_{c=1}^{C}\sum_{p\in\mathcal{P}_i^\xi} \phi\left(\left\langle \mathbf{w}_c^{(2)}, \boldsymbol{\xi}_p^{(2)} \right\rangle\right)
$$

$$
\geq \widetilde{\Omega}(1).
$$

Additionally, for a test data $(\mathbf{x}, y)$ drawn form the first task, let $\epsilon$ as the definition in Definition 3.1, we have

$$
\epsilon F_{T^{(2)}}(\mathbf{x}) = \epsilon\sum_{c=1}^{C} \lambda_c \phi\left(\left\langle \mathbf{w}_c^{(2)}, y\mathbf{u}^{(1)} \right\rangle\right) + \epsilon\sum_{c=1}^{C} \lambda_c \phi\left(\left\langle \mathbf{w}_c^{(2)}, y\mathbf{v}^{(1)} \right\rangle\right)
$$

$$
+ \epsilon\sum_{c=1}^{C} \lambda_c \phi\left(\left\langle \mathbf{w}_c^{(2)}, \epsilon\boldsymbol{\zeta}^{(1)} \right\rangle\right) + \epsilon\sum_{c=1}^{C}\sum_{p\in\mathcal{P}_i^\xi} \phi\left(\left\langle \mathbf{w}_c^{(2)}, \boldsymbol{\xi}_{i,p}^{(1)} \right\rangle\right)
$$

$$
= y\epsilon\sum_{c=1}^{C} \lambda_c \alpha_u^3\alpha_\zeta^{-3} \phi\left(\left\langle \mathbf{w}_c^{(2)}, \boldsymbol{\zeta}^{(2)} \right\rangle\right) + y\epsilon\sum_{c=1}^{C} \lambda_c \phi\left(\left\langle \mathbf{w}_c^{(2)}, \mathbf{v}^{(2)} \right\rangle\right)
$$

$$
+ \sum_{c=1}^{C} \lambda_c \alpha_u^{-3}\alpha_\zeta^3 \phi\left(\left\langle \mathbf{w}_c^{(2)}, \mathbf{u}^{(2)} \right\rangle\right) + \epsilon\sum_{c=1}^{C}\sum_{p\in\mathcal{P}_i^\xi} \phi\left(\left\langle \mathbf{w}_c^{(2)}, \boldsymbol{\xi}_{i,p}^{(2)} \right\rangle\right)
$$

$$
\geq \max_{c\in[C]} \lambda_c \alpha_u^{-3}\alpha_\zeta^3 \phi\left(\left\langle \mathbf{w}_c^{(2)}, \mathbf{u}^{(2)} \right\rangle\right) + (C-1)\min_{c\in[C]} \lambda_c \alpha_u^{-3}\alpha_\zeta^3 \phi\left(\left\langle \mathbf{w}_c^{(2)}, \mathbf{u}^{(2)} \right\rangle\right) - C\max_{c\in[C]} \lambda_c \phi\left(\left|\left\langle \mathbf{w}_c^{(2)}, \mathbf{v}^{(2)} \right\rangle\right|\right)
$$

$$
- \alpha_u^3\alpha_\zeta^{-3} \sum_{c=1}^{C} \lambda_c \phi\left(\left|\left\langle \mathbf{w}_c^{(2)}, \boldsymbol{\zeta}^{(2)} \right\rangle\right|\right) - CP\max_{c\in[C],p\in\mathcal{P}_i^\xi} \phi\left(\left|\left\langle \mathbf{w}_c^{(2)}, \boldsymbol{\xi}_{i,p}^{(2)} \right\rangle\right|\right)
$$

$$
\geq \Omega\left(\frac{\alpha_u^{-3}\alpha_\zeta^3}{C}\right) - \alpha_u^{-3}\alpha_\zeta^3 CO\left(\sigma_0^3\alpha_u^3\right) - CO\left(\sigma_0^3\alpha_v^3\right) - C\alpha_u^3\alpha_\zeta^{-3}o(1) - CPO\left(\sigma_0^3\sigma_\xi^3\right)
$$

$$
\geq \widetilde{\Omega}(1).
$$

Note that $\mathbb{P}(\epsilon = y) = 1/2$, using the union bound, we conclude our proof. $\square$

**Corollary 5.5.** *At the end of the second task, for $\widetilde{T}^{(2)} = \widetilde{\Theta}\left(\frac{1}{\eta\sigma_0\alpha_u^3}\right)$, if $\alpha_u \geq \omega(\alpha_v)$ and $\alpha_\zeta \geq \Omega(\alpha_u)$, we have*

$$
\max_{c\in[C]} \left\langle \mathbf{w}_c^{(2)}(\widetilde{T}^{(2)}), \mathbf{u}^{(1)} \right\rangle \leq o(1),
$$

$$
\max_{c\in[C]} \left\langle \mathbf{w}_c^{(2)}(\widetilde{T}^{(2)}), \mathbf{v}^{(2)} \right\rangle = \widetilde{\Theta}\left(\sigma_0\alpha_v\right),
$$

$$
\max_{c\in[C]} \left\langle \mathbf{w}_c^{(2)}(\widetilde{T}^{(2)}), \mathbf{u}^{(2)} \right\rangle \geq \widetilde{\Omega}(1).
$$

*Proof.* Lemma E.4 implies that at $\widetilde{T}^{(2)} = \widetilde{\Theta}\left(\frac{1}{\eta\sigma_0\alpha_u^3}\right)$, we have

$$\max_{c\in[C]}\left\langle \mathbf{w}_c^{(2)}(t), \mathbf{u}^{(2)}\right\rangle \geq \Omega\left(C^{-1/3}\right),$$

$$\max_{c\in[C]}\left\langle \mathbf{w}_c^{(2)}(t), \mathbf{v}^{(2)}\right\rangle = \widetilde{\Theta}\left(\sigma_0\alpha_v\right).$$

Moreover, Lemma E.3 implies that

$$\max_{c\in[C]}\left\langle \mathbf{w}_c^{(2)}(\widetilde{T}^{(2)}), \boldsymbol{\zeta}^{(2)}\right\rangle \leq o\left(1\right).$$

Since $\mathbf{u}^{(1)} = \alpha_u\alpha_\zeta^{-1}\boldsymbol{\zeta}^{(2)}$ and $\alpha_\zeta \geq \Omega\left(\alpha_u\right)$, we conclude the proof. $\qquad\square$

# F. Additional Experiments

## F.1. Simulated Dataset

For simulated dataset, we use PyTorch as the deep learning framework for training. The CNN model used is defined as Equation (1) with $C = 10$, and the model is optimized using GD. The learning rate is set to 0.1. In each stage, the model is trained for 100 epochs. The hyperparameters are set as follows: $P = 1$, $\sigma_\xi = 0.5\sqrt{d}$, and $d = 50$. The value of $\alpha_u$ is fixed at 1, while $\alpha_v$ and $\alpha_\zeta$ are varied across discrete values. Specifically, $\alpha_v$ takes values in the range $\{0, 0.1, 0.2, \ldots, 1.0\}$, and $\alpha_\zeta$ takes values in the range $\{0, 0.1, 0.2, 0.3, \ldots, 1.0\}$. For each combination of $\alpha_v$ and $\alpha_\zeta$, In Figure 2, we perform 20 independent experiments to calculate the average accuracy of both tasks at the completion of the first and second stages.

We then include additional experimental results for simulated datasets. Figure 6 shows the case that both $\mathbf{u}^{(1)}$ and $\mathbf{u}^{(2)}$ are captured by the model. Since the signal of $\boldsymbol{\zeta}^{(2)}$ is slight, CF does not occur in this case. Moreover, Figure 7 shows the case that $\mathbf{v}^{(1)} = \mathbf{v}^{(2)}$ is learned by the model while none of the other components are fitted by the model since $\mathbf{v}$ has a large norm in this experiment.

## F.2. Real-World Datasets

In real-world datasets, we use ResNet-18 as the CNN model, which is trained for 100 epochs in both the first and second stage. We show the full results for Figure 3. Figures 8 to 10 show the performance of the model on both first and second tasks in different stages in CIFAR-10, CIFAR-100, and Tiny-ImageNet, respectively.

We then use T-SNE (van der Maaten & Hinton, 2008) to visualize the feature in both the first and second stages of CL in CIFAR-100 and Tiny-ImageNet. The results in CIFAR-100 and Tiny-ImageNet are shown in Figures 11 and 12, respectively. Additionally, we calculate the maximal singular vector in the feature space in CIFAR-100 and Tiny-ImageNet, and the results are shown in Figures 13 and 14.

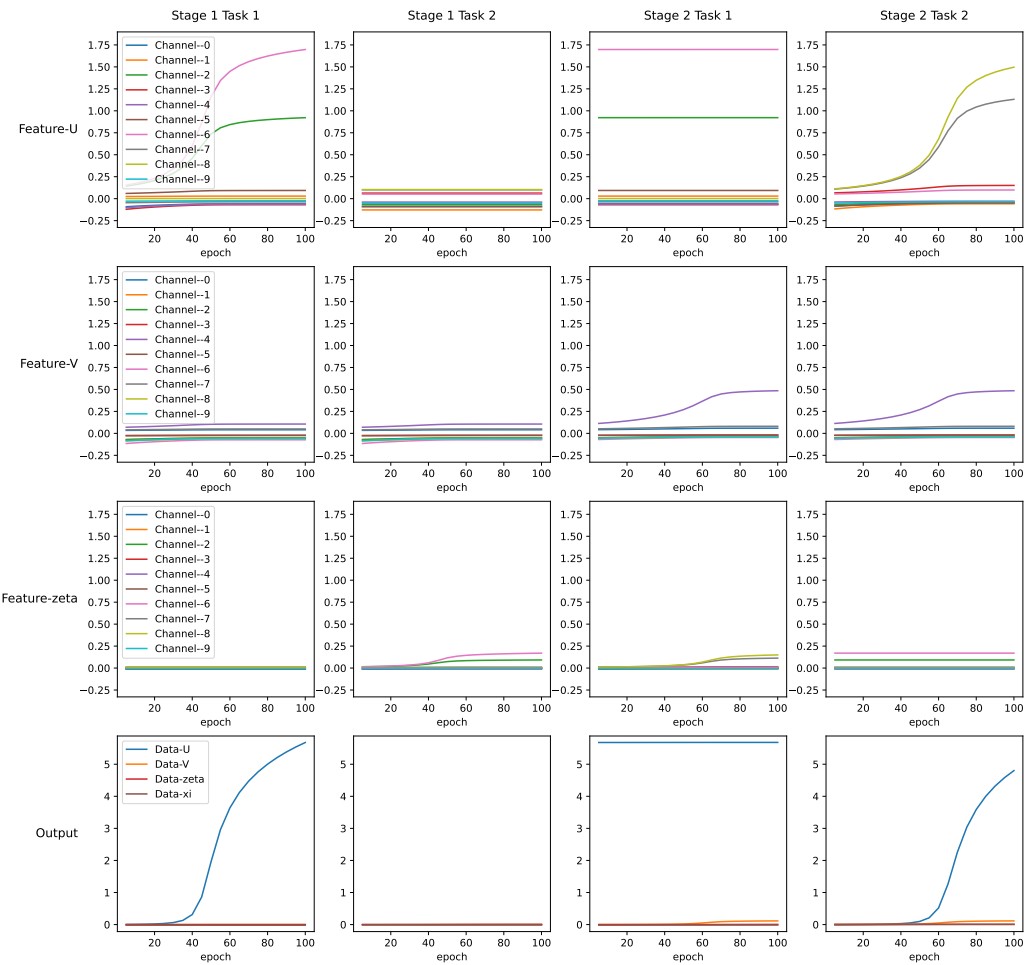

*Figure 6.* The output of each layer in the CNN model using different inputs. **Top three rows**: The output of each channel in the first layer. **Bottom row**: The output of the last layer. ($\alpha_u = 1$, $\alpha_v = 0.9$, and $\alpha_\zeta = 0.1$)

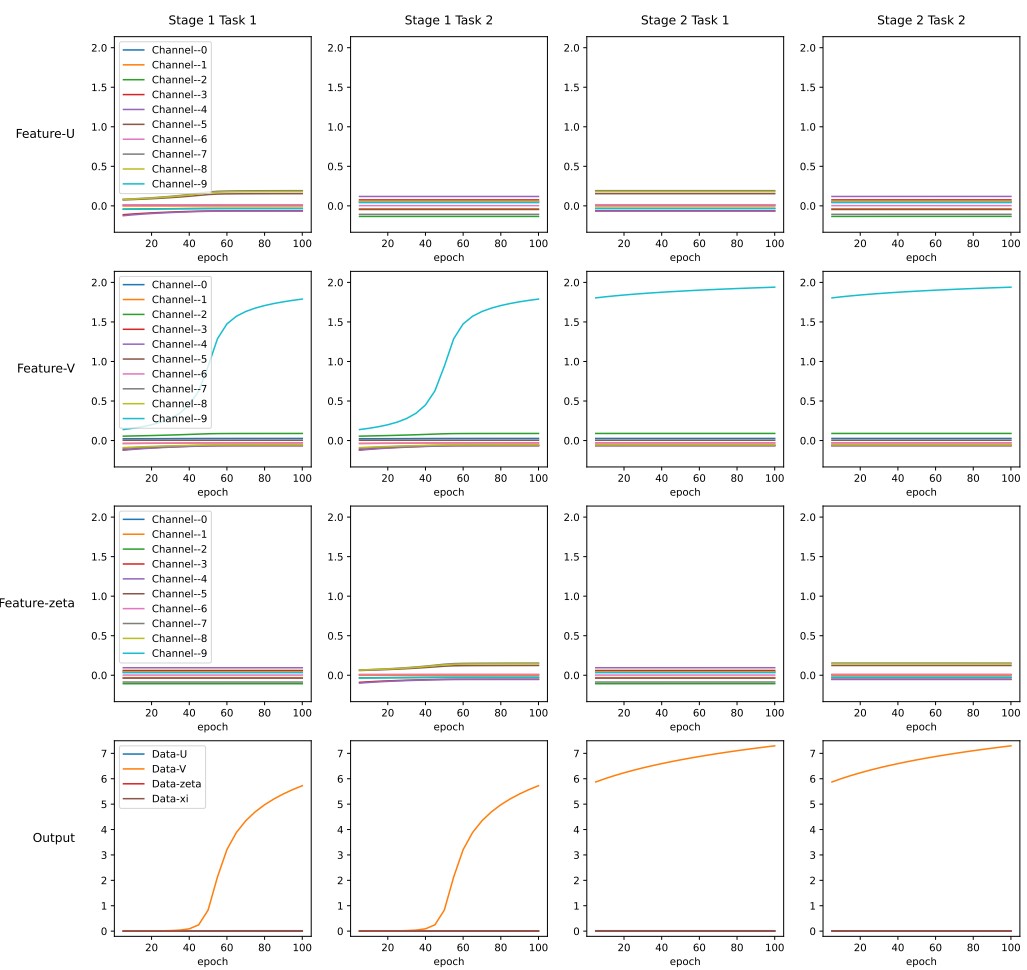

*Figure 7.* The output of each layer in the CNN model using different inputs. **Top three rows**: The output of each channel in the first layer. **Bottom row**: The output of the last layer. ($\alpha_u = 1$, $\alpha_v = 1$, and $\alpha_\zeta = 0.8$)

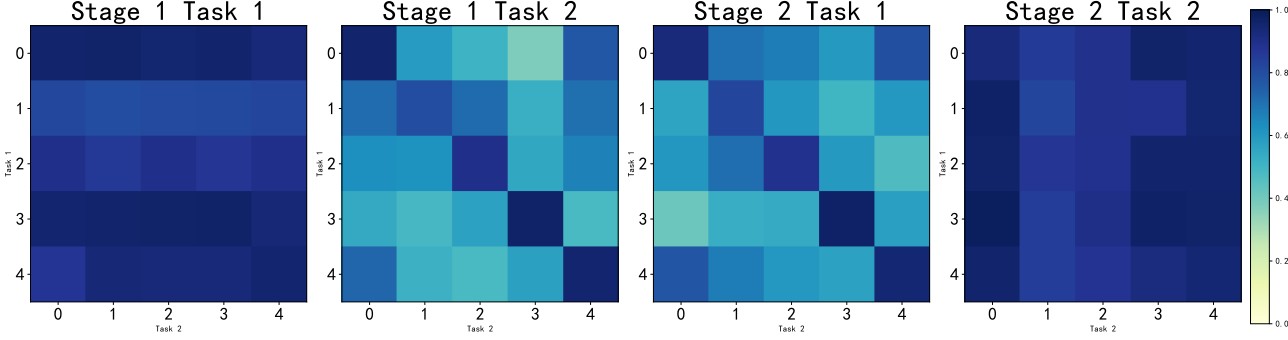

*Figure 8.* Overview of CF in CIFAR-10.

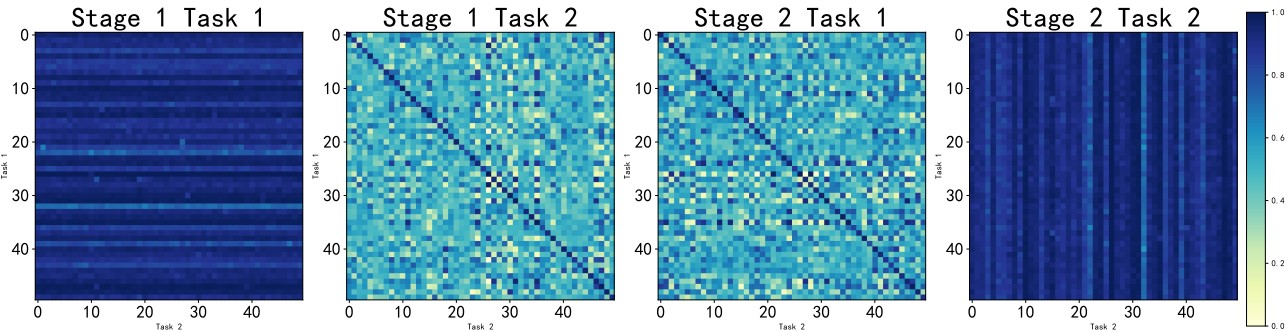

*Figure 9.* Overview of CF in CIFAR-100.

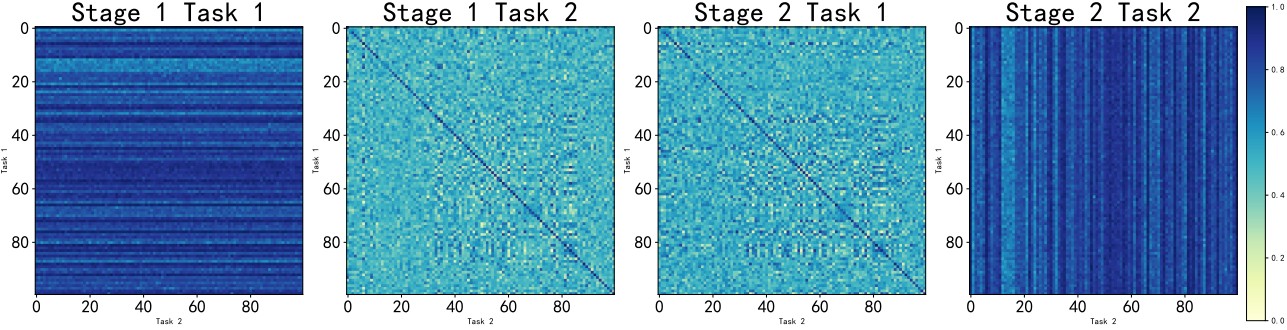

*Figure 10.* Overview of CF in Tiny-ImageNet.

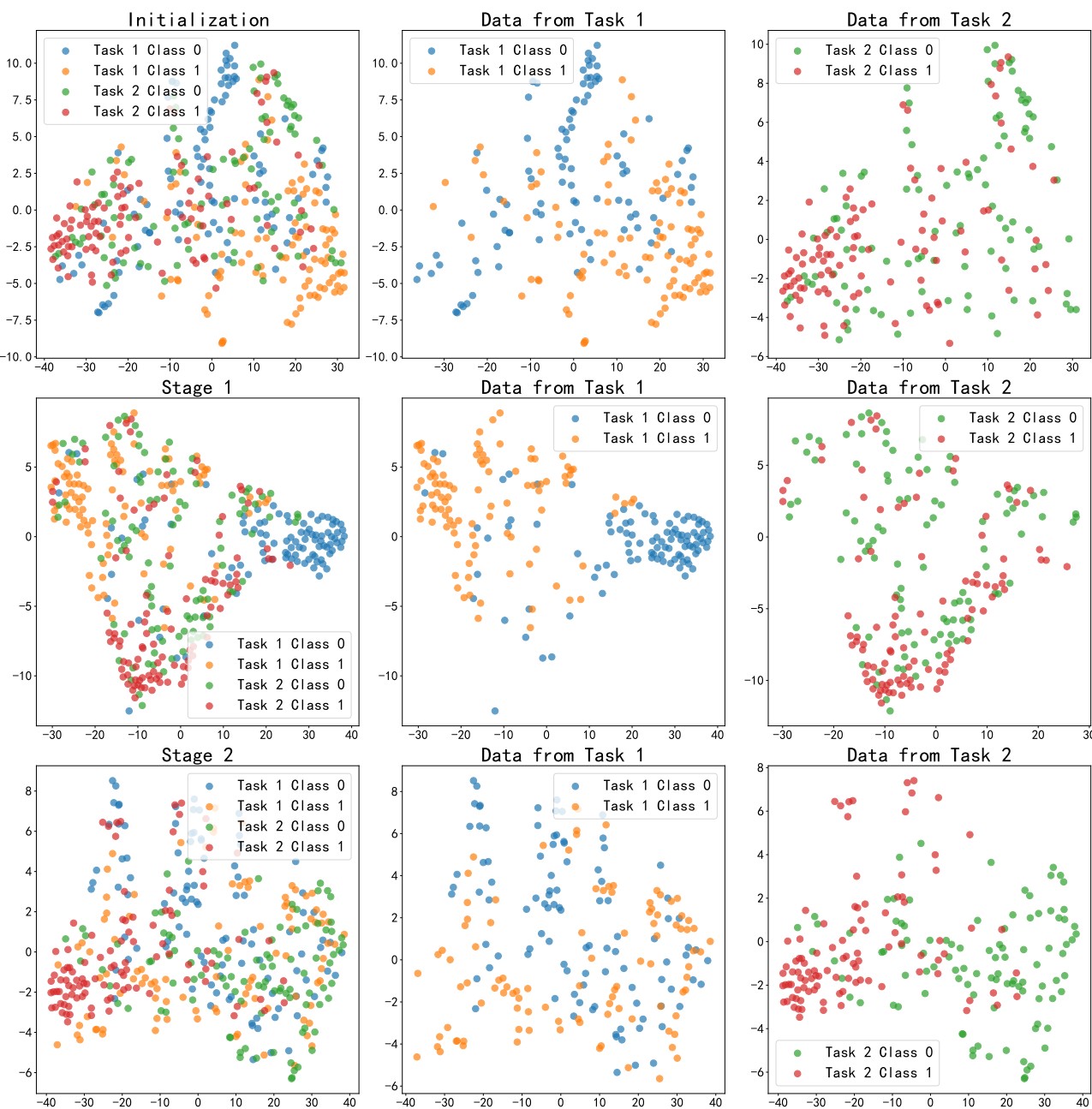

*Figure 11.* Using T-SNE to visualize the feature space at different stages of CL when sequentially train the model on the Task-0 and Task-2 in CIFAR-100. **First row**. The model is randomly initialized. **Second row**. At the end of the first stage. **Third row**. At the end of the second stage.

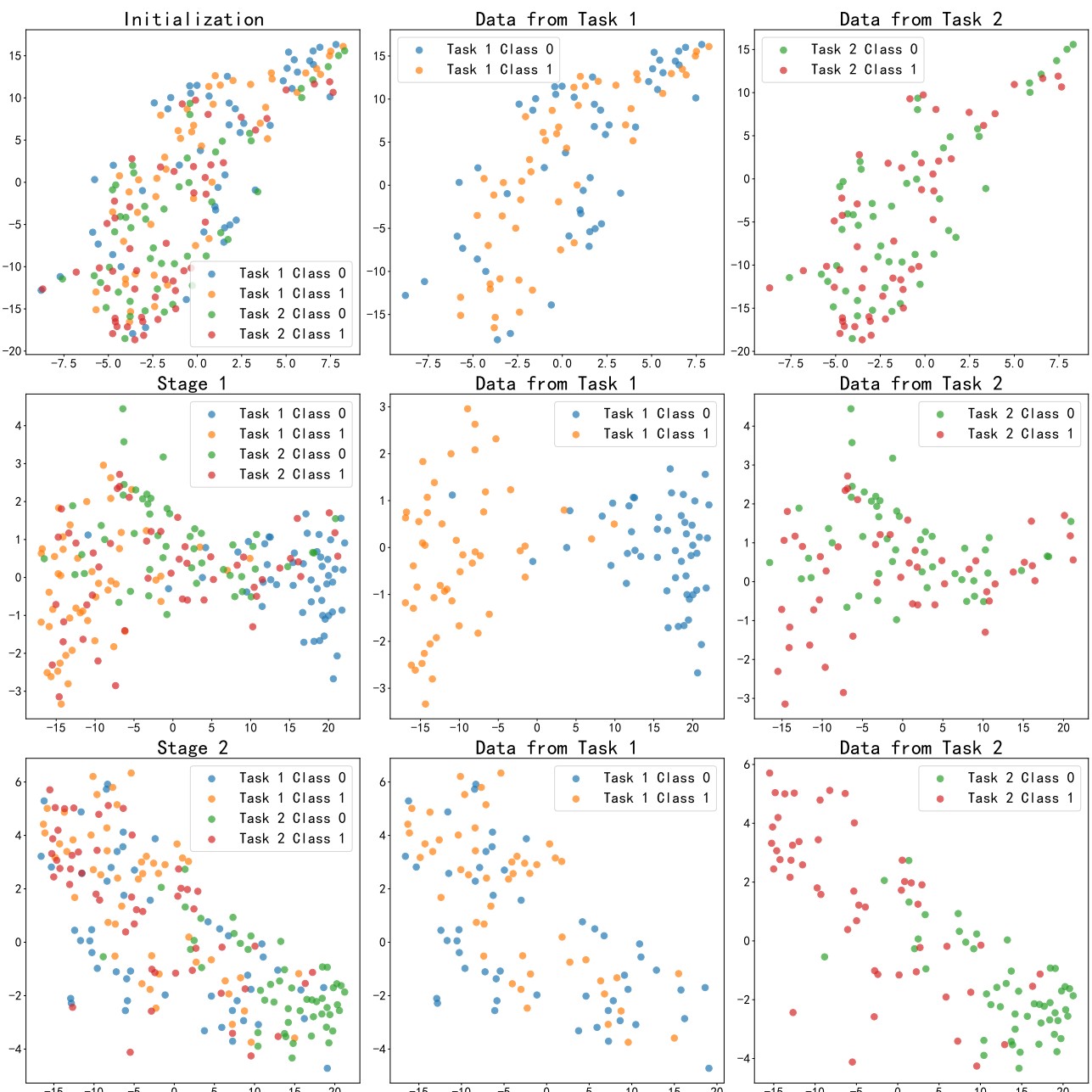

*Figure 12.* Using T-SNE to visualize the feature space at different stages of CL when sequentially train the model on the Task-0 and Task-3 in Tiny-ImageNet. **First row**. The model is randomly initialized. **Second row**. At the end of the first stage. **Third row**. At the end of the second stage.

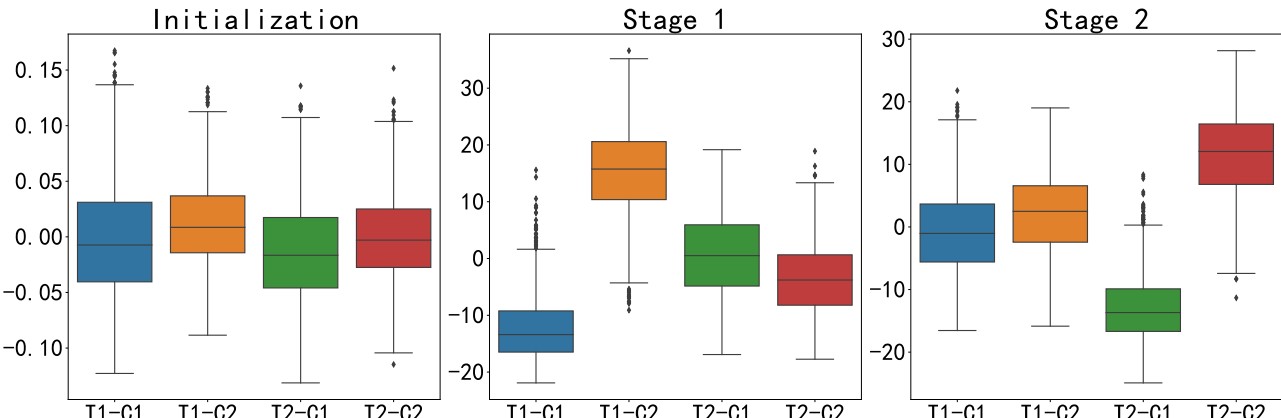

*Figure 13.* Inner product of the features with the maximal singular vector at different stages of CL. The label T$a$-C$b$ indicates that the data drawn from class-$b$ in $a^{\mathbf{th}}$ task. We choose Task-0 and Task-2 in CIFAR-100 as the first and second task.

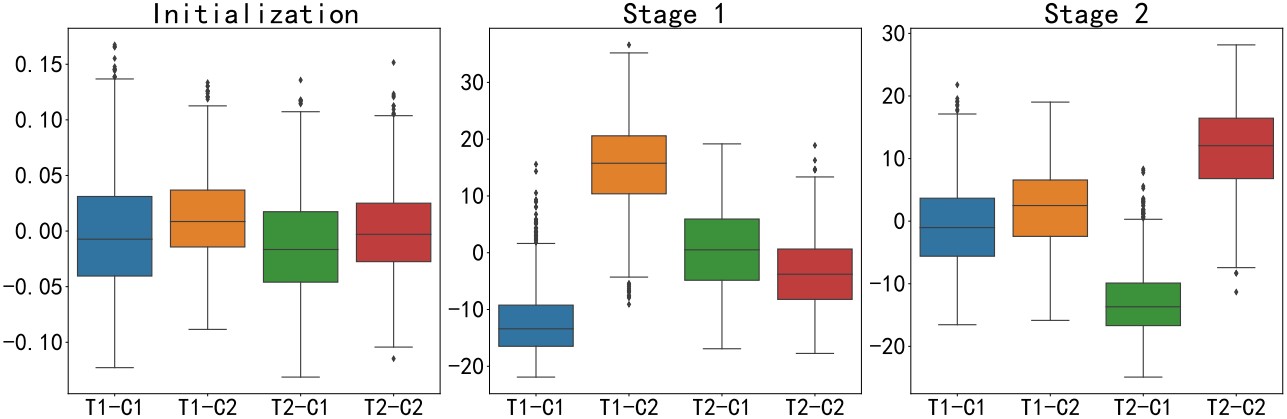

*Figure 14.* Inner product of the features with the maximal singular vector at different stages of CL. The label T$a$-C$b$ indicates that the data drawn from class-$b$ in $a^{\mathbf{th}}$ task. We choose Task-0 and Task-3 in Tiny-ImageNet as the first and second task.

