# OpenReview forum: "Towards Understanding Catastrophic Forgetting in Two-layer Convolutional Neural Networks"
_ICML.cc/2025/Conference — ICML 2025 poster_

### Official Review · Reviewer_ZF1H · 2025-03-11

**Overall Recommendation:** 3

**Summary:**

This paper presents a theoretical analysis of catastrophic forgetting in continual learning using a two-layer neural network model. It constructs a multi-view data structure consisting of task-specific, general, and random features and examines the learning dynamics of these features. The study identifies two potential causes of catastrophic forgetting:
- Task-specific features have a stronger signal than general features.
- Task-specific features from previous tasks are treated as random features in future tasks.

**Claims And Evidence:**

Ambiguity in Feature Representation:
- The data model defines over Features $=\left[e_1, e_2, e_{\text {rob }}\right]$, where the random feature for each task is expressed as $\alpha_\zeta e_{3-\tau}$.
- This implies that the random feature for Task 1 is $\alpha_\zeta e_2$, which coincides with the task-specific feature of Task 2.
- Consequently, the model appears to learn Task 2's feature during Task 1's training phase, and also learn Task 1's feature during Task 2's training phase, potentially confusing the conclusions.

Dependence on Coefficient Choices:
- The proof relies heavily on specific choices of $\alpha_u, \alpha_v, \alpha_\zeta$.
- However, Theorems 5.2 and 5.4 use identical values for $T_u^{(1)}$ and $T_u^{(2)}$, suggesting that for some $T \geq T_u^{(1)}, T_u^{(2)}$, both conclusions hold simultaneously--leading to a contradiction.

Line 191: Setting $\sigma_{\xi}=0$ is invalid under the last condition 3.5.

**Essential References Not Discussed:**

The paper lacks discussions on several theoretical works related to continual learning, including:

- Zheng et al. (2024) – Understanding memory buffer-based continual learning.
- Li et al. (2023) – Fixed design analysis of regularization-based continual learning.
- Ding et al. (2024) – Understanding forgetting in continual learning with linear regression.
- Li et al. (2024) – Theory on mixture-of-experts in continual learning.
- Benjamin et al. (2024) – Continual learning with the neural tangent ensemble.

**Experimental Designs Or Analyses:**

Yes.

**Methods And Evaluation Criteria:**

N/A

**Other Comments Or Suggestions:**

Above.

**Other Strengths And Weaknesses:**

Two Additional Limitations:

Confusion in the Simplified Model (Section 4):
- The analysis in Section 4 is somewhat unclear.
- For instance, if $v_1=v_2$, the paper claims that the model performs well on both tasks during training and testing.
- However, this seems counterintuitive, as it suggests that task-specific vectors do not contribute meaningfully to classification.
- Are there missing conditions on task-specific vectors that would clarify their role in the model's performance?

Limited Generalization: The analysis is restricted to a two-task setting, making it unclear how well the results generalize to m-task scenarios.

**Questions For Authors:**

Above.

**Relation To Broader Scientific Literature:**

- The paper introduces a novel framework to analyze catastrophic forgetting in continual learning, offering valuable theoretical insights into its underlying causes.
- The data model distinguishing task-specific and robust (general) features provides a reasonable perspective for understanding catastrophic forgetting.

**Theoretical Claims:**

I reviewed the main framework of the proof but did not go through the step-by-step details.

---

> ### Author Rebuttal · Authors · 2025-03-31
>
> We thank reviewer ZF1H for the insightful comments. We find the reviewer misunderstanding our framework, and we show a illustration for our framework in **Q3** in our response to reviewer iCej. We the answer the questions as follows.
>
> **Q1**. Ambiguity in Feature Representation
>
> **A1**. Yes, both $\mathbf{e} _1$ and $\mathbf{e} _2$ appear in both tasks, please see our example in **Q3** in our response to reviewer iCej. **The existence of random noise ensures that the task-specific feature can only be used for one task**. For this reason, we let the random feature be $\epsilon \alpha_{\zeta} \mathbf{e}_{3-\tau}$ rather than $y \alpha_{\zeta} \mathbf{e}_{3-\tau}$. Our definition of task-specific feature is reasonable. Furthermore, the task 2's feature $\mathbf{e} _2$ **is not** learned in Task 1's training phase, as shown in Corollary 5.3.
>
>
> **Q2**. The proof relies heavily on specific choices of $\alpha _{u}, \alpha _{v}, \alpha _{\zeta}$.
>
> **A2**. Please see **Q3** in our response to reviewer iCej for clarification of our main ideas. While our theoretical results shows the conditions leading to CF depends on $\alpha _{u}, \alpha _{v}, \alpha _{\zeta}$, they help reveal the underlying causes of CF, as shown on Page 7, left column, lines 341-453. Empirical studies of $\alpha _{u}, \alpha _{v}, \alpha _{\zeta}$ in Figure 2 support our theoretical findings.
>
> **Q3**. About contradiction from $T$.
>
> **A3**. We want to clarify that at the beginning of the second stage we set $\mathbf{w} _{c}^{(2)}(0) = \mathbf{w} _{c}^{(1)}(T^{(1)} _{u})$. Here, $T^{(1)} _{u}$ refers to the time in the first stage while $T^{(2)} _{u}$ refers to the time in the second stage, **but not the total time in the first and seond stages**. The entire process can be described as follows:
> 1. Initially, it takes $T^{(1)} _{u}$ epochs to reach the end of the first stage, after which the process transitions to the second stage. Theorem 5.2 holds in the end of the **first** stage.
> 2. After an additional $T^{(2)} _{u}$ epochs, the second stage concludes. Theorem 5.4 holds in the end of the **second** stage.
>
> **There's no contradiction** for Theorems 5.2 and 5.4.
>
> **Q4**. Setting  $\sigma _\xi = 0$ is invalid under the last condition 3.5.
>
> **A4**. Thanks for pointing this out. First, $\sigma _\xi = 0$ is only used for simplicity in section 4 and **does not impact the analysis of our main results**. Additionally, when setting $\sigma _\xi = 0$ in Section 4, all conditions involving $\sigma _\xi$ in Condition 3.5 should be removed. We will clarify this in our revised version.
>
> **Q5**. The paper lacks discussions on several theoretical works.
>
> **A5**.  Thank you for your suggestions. Benjamin et al. (2024) introduce the concept of NTE in CL, reformulating a single neural network as an ensemble of fixed classifiers. Li et al. (2024) provide a theoretical study of mixture-of-experts models in CL. Ding et al. (2024) analyze factors contributing to forgetting under linear models in CL. Li et al. (2023) investigate $\ell _2$-regularized CL with two linear regression tasks. Zheng et al. (2024) examine memory-based CL under overparameterized linear models. **Our study focuses on CF in a two-layer CNN using multi-view data, which differs from these works.** We will include discussions on these related works in our revised version.
>
> **Q6**. Confusion in the Simplified Model.
>
> **A6**. We clarify that **the reviewer miss the condition** $\alpha _v \gg \alpha _u$ for learning $\mathbf{v}$ in the first stage. If $\alpha _v \gg \alpha _u$, it implies that $T^{(1)} _{v} \gg T^{(1)} _{u}$, and the model will **only capture the general feature $\mathbf{v}^{(1)}$ and not the task-specific feature** $\mathbf{u}^{(1)}$, and   $\mathbf{u}^{(1)}$ will not contribute to classification in task 1. In the right column of Line 200, we similarly shows that if $\alpha _u \gg \alpha _v$, the model will only capture the task-specific feature $\mathbf{v}^{(1)}$ and not the general feature in the first stage, and $\mathbf{v}^{(1)}$ will not contribute to classification in task 1.
>
>
> **Q7**. Limited Generalization.
>
> **A7**. Please see our response to **Q1** to reviewer QA6F.

---

> > ### Comment · Reviewer_ZF1H · 2025-04-01
> >
> > Thank you for the authors' rebuttal. While some concerns have been addressed, I am still confused about something as follows.
> >
> > 1. **Feature Setting:** As the authors responded to Reviewer iCej, the random feature is uncorrelated with the true label. Given this, why do the authors use $\epsilon \cdot e$ instead of just $e$? What does the $\epsilon$ represent in this context? In most theoretical works on feature learning, I’ve only seen one label $y$ involved, rather than both $y$ and $\epsilon$.
> >
> > 2. **Generalization:** I read the authors' response to Reviewer QA6F and examined the proof framework. I think the most challenging aspect of extending the analysis to an $m$-task setting is handling the relationship between $y_i$ and $\epsilon_j$, where $i$ and $j$ index different tasks. This is a central point of confusion for me in this work: how do the authors model or address the dependencies between $y_i$ and $\epsilon_j$, and between $y_i$ and $y_j$, in the learning dynamics? For example, when predicting new data from task 1 after training on task 2, shouldn’t some terms involving $y_1 y_2$ and $y_1 \epsilon_2$ appear? If I’m misunderstanding this, could the authors kindly clarify?
> >
> > 3. **Data Noise:** Regarding the data noise $\xi$, I don’t believe its presence can be ignored in the analysis of the main results. If data noise is involved, the analysis should also account for its impact on learning dynamics. I also question the SNR conditions: in Cao’s work, only one feature is considered, whereas this work involves two features, $u$ and $v$. What is the SNR condition for $v$? And where in the analysis is this condition used?

---

> > > ### Author Response · Authors · 2025-04-03
> > >
> > > Thanks for your response. We address your further concerns as follows:
> > >
> > > **Q1. Feature Setting.**
> > >
> > > **A1.** The definition of $e$ and $ε$ can refer to **A3** to iCej, where $ε \sim U(\{+1,-1\})$ is a ($y$-independent) random noise term generating random feature $\zeta$. We could also use $e _{3-\tau}$ instead of $ε\cdot e _{3-\tau}$. The only technical difference is **how we partition the sample into two sets** when studying the learning dynamics of $\zeta$:
> > >
> > > $$
> > > \langle w^{(\tau)} _{c}(t+1), \zeta^{(\tau)}\rangle-\langle w^{(\tau)} _{c}(t),\zeta^{(\tau)} \rangle
> > > =-\frac{\eta}{n}\sum _{i=1}^{n}y^{(\tau)} _i ε^{(\tau)} _i\ell^{\prime}(F({x}^{(\tau)} _i),y^{(\tau)} _i)(\langle{w}^{(\tau)} _{c}(t), \zeta^{(\tau)}\rangle)^2α _\zeta^2
> > > =-\frac{\eta}{n}(\sum _{i\in\mathcal{I} _{=}}\ell^{\prime}(F({x}^{(\tau)} _i),y^{(\tau)} _i)-\sum _{i\in\mathcal{I} _{\ne}}\ell^{\prime}(F({x}^{(\tau)} _i),y^{(\tau)} _i))(\langle{w}^{(\tau)} _{c}(t),\zeta^{(\tau)}\rangle)^2α _\zeta^2
> > > $$
> > > - With  $ε\cdot e _{3-\tau}$, we define $\mathcal{I} _{=}=\{i:y _i=ε _i\}$ and $\mathcal{I} _{\ne}=\{i:y _i\ne ε _i \}$.
> > > - If we insteadly use $e$ along, the partition would instead be $\mathcal{I} _{=}=\{i:y _i=+1\}$ and $\mathcal{I} _{\ne}=\{i:y _i\ne-1\}$.
> > >
> > > **Our analysis remains valid in either case**.
> > >
> > > **Q2. Generalization.**
> > >
> > > **A2. All elements in $\{y _i\} _{i\in[M]}\cup\{ε _i\} _{i\in[M]}$ are mutually independent**. In stage $\tau$, the model is **only** trained on task $\tau$. The update rule of features depend solely on the current task's $y^{(\tau)}$ and $ε^{(\tau)}$ not on other tasks’ $y,ε$:
> > > - **Task specific feature**. $ \langle w^{(\tau)} _{c}(t+1),u^{(\tau)} \rangle-\langle w^{(\tau)} _{c}(t),u^{(\tau)}\rangle=-\frac{\eta}{n}\sum^{n} _{i=1}y^{(\tau)} _iy^{(\tau)} _i\ell^{\prime}(F({x}^{(\tau)} _i),y^{(\tau)} _i)(\langle  {w}^{(\tau)} _{c}(t),u^{(\tau)}\rangle)^2α _u^2$.
> > > - **General feature** $\langle w^{(\tau)} _{c}(t+1),v^{(\tau)}\rangle-\langle w^{(\tau)} _{c}(t),v^{(\tau)}\rangle=-\frac{\eta}{n}\sum^{n} _{i=1}y^{(\tau)} _iy^{(\tau)} _i\ell^{\prime}(F({x}^{(\tau)} _i),y^{(\tau)} _i)(\langle {w}^{(\tau)} _{c}(t),v^{(\tau)}\rangle)^2α _v^2$.
> > > - **Other task's specific feature (Random Feature)**. Refer to **A1** in this response.
> > >
> > > We show how $y,ε$ affects the analysis:
> > > - Stage 1 (learning on task 1): we study the order of each component (Lemma 4.1) at the initialization, then study the learning speed of each component. Both $\max _{c\in[C]} \langle w^{(1)} _{c}(t),u^{(1)}\rangle$ and $\max _{c\in[C]}\langle w^{(1)} _{c}(t),v^{(1)} \rangle$ **monotonically increasing** with speed depending on $α _u$ and $α _v$, respectively. **The direction of the update of random feature depends on the sign of** $\sum _{i\in\mathcal{I} _{=}}\ell^{\prime}(F({x}^{(\tau)} _i),y^{(\tau)} _i)-\sum _{i\in\mathcal{I} _{\ne}}\ell^{\prime}(F({x}^{(\tau)} _i), y _i^{(\tau)} )$, and the speed depends on $α _\zeta$.
> > >
> > > - Stage 2 (learning on task 2): we only care about the **order of components at the beginning (Corollary 5.3)**. While $y^{(1)}$ and $ε^{(1)}$ affects the direction of the random noise updates in Stage 1, their effects **do not explicitly appear in the order of each component in Corollary 5.3**. We then study learning speeds and directions of components in the second stage based on Corollary 5.3 and update rules. **$y,ε$ in past tasks disappear in both update rules and Corollary 5.3** in stage 2.
> > >
> > > By iteratively analyzing the component orders at the start/end of each stage and learning speeds, the framework generalizes to $m$-task IL, **and $y^{(1)}y^{(2)},y^{(1)}ε^{(2)}$ will not appear in the analysis.**
> > >
> > > **Q3. Data Noise.**
> > >
> > > **A3.** We claim again that **we do not ignore $\xi$ in our main results in section 5, while SNR condition requires $\sigma^3_\xi$ to be relatively small compared with $nα^3_u$**. $\xi$ is only omitted in section 4 for simplicity in illustrating intuitions. Following Cao et al. (2022b), we let $\xi$ be orthogonal to the common feature space spanned by the vectors $\{e _1,e _{rob},e _2\}$, so that **update rules** of $\langle w^{(\tau)} _{c}(t),u^{(\tau)}\rangle$ and $\langle w^{(\tau)} _{c}(t),v^{(\tau)}\rangle$ (shown in **A2** of this response) **remain unchanged** regardless of whether $\xi=0$, which motivates us to set $\xi=0$ in section 4.
> > >
> > > Unlike prior work, we involve two features. However, **no additional SNR condition is needed for** $v$ because:
> > >
> > > 1. When $α _v\leq o(α _u)$, then the learning speed of  $v$ is lower than $u$, and SNR condition of $u$ is sufficient to guarantee that the model fits $u$ rather than noise $\xi$.
> > > 2. When $α _v\geq\Omega(α _u)$, The SNR condition for $v$ (i.e., $nα^3 _v/\sigma^3 _\xi\geq\omega(1)$) immediately holds. This ensures the model fits $v$ over noise $\xi$.
> > >
> > > The SNR condition is used in Lemma 5.1 and Corollary E.1. The analyses show that **noise updates remain slow during the whole training process** under the SNR condition. Therefore, we ignore the effect of noise in section 4.

---

### Official Review · Reviewer_QA6F · 2025-03-12

**Overall Recommendation:** 5

**Summary:**

Continual learning is an important area in machine learning, and catastrophic forgetting is the most important problem in continual learning. Despite the empirical efforts on suppressing forgetting in continual learning, the theoretical understanding of catastrophic forgetting remains less studied, especially in CNN. In this work, the authors propose a novel framework  to theoretically study catastrophic forgetting in a simple CNN model. Their analysis shows that the different distribution of features in different tasks causes catastrophic forgetting. They conduct experiments to verify their findings.

--------------------after rebuttal----------------------
Thanks the author for the reply. They have sufficiently addressed my questions and concerns. After checking other reviewer's comments and the response, I think this paper takes a nice step on understanding the CL. Futher works may be inspired by this paper. Thus, I increase my score.

**Claims And Evidence:**

The main claim of this paper is presented in Section 5, where the authors identify the condition leading to catastrophic forgetting as follows:

1. The task-specific feature has a relatively larger signal than the general feature.
2. The task-specific feature from one task appears as a random feature in the second task with a strong signal.

**Essential References Not Discussed:**

Most of the important related works are included in this work. I have not found any other important references that should be discussed in this work.

**Experimental Designs Or Analyses:**

I have reviewed the experimental designs and analyses presented in Section 6, and most of the analyses are reasonable and convincing to me.

**Methods And Evaluation Criteria:**

1. Theoretically, the authors propose a framework to show the evidence for the main claim. They formally define catastrophic forgetting in Definition 3.3.

2. Experimentally, they present a numerical analysis with simulated data. By evaluating the performance of model on the old and new tasks after training, they find that the signals of the general feature and the task-specific feature are relevant to catastrophic forgetting. Furthermore, they visualize the feature space in a real-world data set to show that the learned features after learning the old and new tasks are different.

**Other Comments Or Suggestions:**

1. Line 79, right column, $\mathbb{O} \in \mathbb{R}^{d \times (P+3)} \to \mathbb{O} \subseteq \mathbb{R}^{d \times (P+3)}$.
2. Line 83, right column, $\mathbf{\xi}(\tau) \to \mathbf{\xi}^{(\tau)}$.
3. The notation can be improved. For example, the authors use both $n_{=}$ and $\vert \mathcal{I}_ {=} \vert$ to denote the size of $\mathcal{I}_{=}$.

**Other Strengths And Weaknesses:**

**Strengths**

1. Catastrophic forgetting is a valuable and important problem to study in continual learning. The authors propose a novel framework and study this problem in the CNN model, which is less explored.

2. The authors analyze the training dynamics of continual learning in two stages, which is different from previous studies on studying the learning dynamics in the two-layer CNN model.

3. The authors conduct experiments on both simulated and real-world datasets to verify the theoretical results.


**Weaknesses**

1. The authors only study the case of two-task continual learning, which limits the contribution of the work.

2. In the section 6.2, the authors should add more description to illustrate how to distinguish the task-specific feature and general feature in real-world dataset.

3. The remarks about Condition 3.5 could benefit from further clarification. The authors argue that the condition on $\sigma_\xi$ is to ensure that ***"at the beginning of the training process, the network cannot easily classify the data with the task-specific feature"***, this claim requires a more detailed explanation.

4. The authors use Figure 5 to show that after training on the second task, the model forgets the learned feature from the first task. It is not convincing to me, because the signal of the feature in the second task may be larger than in the first task, and the authors only show the inner product of the features with the largest singular vector.

**Questions For Authors:**

1. Why do some results, such as Corollaries 5.3 and 5.5, show only the order of the largest inner product among $C$ channels?

2. What is the motivation and intuition behind the random noise in the multi-view data model?

**Relation To Broader Scientific Literature:**

The main contribution of this paper is that they analyze the catastrophic forgetting problem in a specific CNN model, which has not been studied in recent works. As for the results in this paper, the authors analyze the reason behind the catastrophic forgetting of the features stored in the data. In the previous works, they show the reason behind the catastrophic forgetting from task dissimilarity, task order, and so on.

**Theoretical Claims:**

The theoretical claim in this paper consists of two sections: In Section 4, the authors study catastrophic forgetting in a simplified setting, and in Section 5 they show the main results. I have read the proofs in both sections, and their theoretical claim seems correct to me.

---

> ### Author Rebuttal · Authors · 2025-03-31
>
> Thanks for your positive comments. We answer the questions as follows.
>
> **Q1**. The authors only study the case of two-task continual learning.
>
> **A1**. Our framework can be naturally extended to $M$-class CL. Our framework can be extended to $M$-task scenarios. A key modification is to assume $m+1$ basis vectors $\\{\mathbf{e} _{(\tau)}\\}^{M} _{\tau=1} \cup \\{\mathbf{e} _{rob}\\}$ exists in common feature space. In task $\tau$, the data $\mathbf{x}$ with label $y$ contains features $\\{ y\alpha _u \mathbf{e} _{(\tau)}, y\alpha _v \mathbf{e} _{rob} \\} \cup \\{\epsilon\alpha^{(m,\tau)} _\zeta\mathbf{e} _{(m)}\\} _{m \in [M], m \ne \tau}$ along with background noise, where $\epsilon$ is defined in Definition 3.1.
>
> **Proof sketch**:
> - Step 1. Show that at the initialization, $\forall \tau \in [M],$
> $$
> \max _{c\in[C]} \langle \mathbf{w} _{c}, \mathbf{u}^{(\tau)} \rangle = \widetilde{\Theta}(\sigma _0\alpha _u), \max _{c\in[C]} \langle \mathbf{w} _{c}, \mathbf{v} \rangle = \widetilde{\Theta}(\sigma _0\alpha _v), \max _{c\in[C]} \langle \mathbf{w} _{c}, \mathbf{\zeta}^{(\tau)} \rangle = \widetilde{\Theta}(\sigma _0\sigma _\zeta), \max _{c\in[C]} \langle \mathbf{w} _{c}, \mathbf{\xi} \rangle = \widetilde{\Theta}(\sigma _0\sigma _\xi).
> $$
> - Step 2. For any $m \in [M]$, iteratively analyze the learning speed of each component and derive the order of $\max _{c\in[C]} \langle \mathbf{w} _{c}, \mathbf{u}^{(\tau)} \rangle$ for any $\tau$, as well as $\max _{c\in[C]} \langle \mathbf{w} _{c}, \mathbf{v} \rangle$, $\max _{c\in[C]} \langle \mathbf{w} _{c}, \mathbf{\zeta}^{(\tau)} \rangle$ and $\max _{c\in[C]} \langle \mathbf{w} _{c}, \mathbf{\xi} \rangle$ at the end of the stage $m$. Those orders depend on $\alpha^{(m,\tau)} _\zeta, \alpha _v, \alpha _u$. The technique is similar within two-task scenario.
> - Step 3.  Identify the conditions under which catastrophic forgetting (CF) occurs.
>
> As a result, we believe **the conclusions and implications are similar to the two-task scenario**.
>
> Similarly, our framework can be extended to **Class-IL**: We can **modify our loss from logistic loss to cross entropy loss** for multi-class classification. The main idea remains the same—analyzing the learning dynamics of different features and determining the order of each component at the end of each stage, just as in task-IL.
>
> **Q2**. Distinguish the task-specific and general feature in real-world dataset.
>
> **A2**. It is a good question. In real-world datasets, our empirical study on Figure 3 shows that the existence of general feature is hard to satisfy. In most case, the model learns the task-specific feature. We believe that distinguishing features is a challenging problem to be solved in future works.
>
> **Q3**. More detailed explanation about $\sigma _\xi$
>
> **A3**. $\sigma _\xi$ reflects the strength of noises in an image. Our condition about  $\sigma _\xi$ is that $\omega(1)\leq \sigma _\xi \leq o(P^{-1}\sigma _{0}^{-1})$ and $n\alpha^{3} _{u}/\sigma^{3} _{\xi}\geq \omega(1)$. For the first condition, $\sigma _\xi \leq o(P^{-1}\sigma _{0}^{-1})$ is to ensure that at the initilization, $\max _{c\in[C]} \langle \mathbf{w} _{c}, \sigma _{\xi} \rangle$ is small enough. $\sigma _\xi \geq \omega(1)$ is to ensure the learning of features is hard under the noises. For the second condition, it provides  a lower bound for $\text{SNR}^3$, which is also shown in Cao et al.(2022b).
>
> **Q4**. About Figure 5.
>
> **A4**. First, Figure 5 demonstrates that in the second stage, the model relies on task 2's task-specific feature for classification. Additionally, in the bottom row, middle column of Figure 4, we observe that the data from the first task is no longer separable, unlike in the middle row, middle column of Figure 4. These empirical findings provide evidence that the model forgets the learned feature from the first task.
>
> **Q5**. Why do some results, such as Corollaries 5.3 and 5.5, show only the order of the largest inner product among $C$ channels?
>
> **A5**. We have shown that both $\max _{c\in[C]} \langle \mathbf{w} _{c}, \mathbf{u}^{(\tau)} \rangle$ and $\max _{c\in[C]} \langle \mathbf{w} _{c}, \mathbf{v} \rangle$ monotonically increases. The analysis of the largest inner product among $C$ channels demonstrates that **at least one channel learns the component**.
>
> **Q6**. Motivation and Intuition behind the random noise in the multi-view data model?
>
> **A6**. **The existence of random noise ensures that the task-specific feature can only be used for one task**. For this reason, we let the random feature be $\epsilon \alpha _{\zeta} \mathbf{e} _{3-\tau}$ rather than $y \alpha _{\zeta} \mathbf{e} _{3-\tau}$. Moreover, we use $\alpha _\zeta$ to control the strength of the random feature. When $\alpha _\zeta$ is zero, the random feature disappears; and as it increases, forgetting occurs.

---

### Official Review · Reviewer_g9HH · 2025-03-12

**Overall Recommendation:** 4

**Summary:**

This paper investigates the catastrophic forgetting (CF) phenomenon in continual learning (CL) for convolutional neural networks (CNNs) from the perspective of training dynamics. The paper considers a multi-view data model with four components: task-specific features, general features, random features, and background noise. The theoretical results in this paper show some reasons for the occurrence of CF: (1) the task-specific feature has a larger signal than the robust feature, which can genearize well in both tasks, causing the model to learn the task-specific feature rather than the robust feature; (2) the task-specific feature from one task acts as a random feature for another task, causing the model to forget the learned task-specific feature for task 1 while learning task 2. Experiments conducted on both simulated and real-world data sets validate the theoretical claims.

**Claims And Evidence:**

Yes, the central claims are supported by evidence. The theoretical claims in this paper are all supported by strict proofs. The results are also validated by experiments on both simulated and real data sets.

**Essential References Not Discussed:**

I do not find any essential references that are not discussed.

**Experimental Designs Or Analyses:**

Yes, the experimental design is reasonable. The main purpose of the experiments in this paper is to validate the findings derived from the theoretical results. In addition to the experiments on simulated data, the experiments on the real-world datasets are also very important, which makes the theoretical findings meaningful in the real-world.

**Methods And Evaluation Criteria:**

Yes. For the theoretical part, the paper uses the classification accuracy to measure the performance of a model, it is very reasonable. For the experimental part on the real-world dataset, it is hard to decide which feature has a stronger signal, the paper uses the singular vector with maximal singular value, I think it makes sense.

**Other Comments Or Suggestions:**

Typos:
  - On line 99, the left column, "a lot of works study" but not "a lot of works studies".
  - In the second part of Definition 3.3, it should be $(x,y) \sim \mathcal{D}_ {z}^{(1)}$ but not $(x,y) \sim \mathcal{D}_{z}^{(2)}$.
  - On line 79, the right column, it might be "for any $T^{(2)}$" but not "for $T^{(2)}$".
  - In the second and third equations in Theorem 5.4, $(x,y)$ is better than $(x_i,y_i)$.

Some Minor Issues:
  - **The dimension of $e$'s**: On line 80, the right column, since $\mathbb{O}$ is a subspace in $\mathbb{R}^{d \times (P+3)}$ and $\\{ e_1, e_2, e_{rob} \\}$ span $\mathbb{O}$, they should be of dimension $d \times (P+3)$. While on line 116, the left column, the expression $I_d - e_1 e_1^T$ shows that $e_1$ is of dimension $d$.
  - **The definition of $\ell$**: On line 151, the left column, the logistic loss is defined as a function of two variables $\ell(F(x),y)$. While on line 159, the left column, the derivative of $\ell$ is used as a scalar, which is confusing. I think a good way is to define the logistic loss as $\ell(F(x),y) = L(yF(x)) = \log \left( 1 + \exp{(-y F(x))} \right)$ so on line 159, we can use $L^\prime(y_i F(x_i))$.
  - **The expression in Definition 3.3**: the expression "with a high probability $\delta_1$" is not proper, since correctly classifying the test data  with a high probability requires $1-\delta_1$ is close to $1$. So is can be corrected as "with a high probability $1-\delta_1$". The expression in the second part faces the same problem, it might be better to change "with a high probability $\delta_2$" into "for a small $\delta_2$ that is close to $0$".
  - **The definition of $\mathcal{I}$**: On line 117, the left column, $\mathcal{I}$ is the index set of $\mathcal{S}$, do you mean that $\mathcal{I} = [n]$?
  - **The assumptions in Condition 3.5**: The usual concept of over-parameterization refers to the scenario where the number of parameters of the model exceed the size of the training dataset, in this paper, it means that $n \le d\times(P+3)$, which is a little far from $nP \le o(\sqrt{d})$. Maybe the order of the expression should be changed, for example, you can say "$nP \le o(\sqrt{d})$, i.e., the network is over-parameterized".

**Other Strengths And Weaknesses:**

Strengths

- The paper investigates CF in CL from a training dynamics perspective, it is a novel view from feature learning.
- The paper explains the reason of CF in CL through theoretical analysis. The results of the paper are meaningful and insightful, and are consistent with the validation experiments in this paper.
- The experimental results support the theoretical claims, making the results more convincing.
- The writing is kind to readers, the authors show the insights of the learning process, which helps to understand the main ideas of this paper.

Weaknesses

- Although theoretical results identify the condition that leads to the CF, the main implication of the theoretical results remains unclear.

- This work studies CF based on learned features, but it is unclear how their analysis relates to over-parameterization, task similarity, and other factors identified in earlier research.

- In the right row of line 231. The authors claim that $\ell^{\prime}(\mathbf{z}_1)/\ell^{\prime}(\mathbf{z}_2)$ has a same order with $\exp(\mathbf{z}_2 - \mathbf{z}_1)$, but lacks a theorem to illustrate it.

**Questions For Authors:**

See the "Weaknesses" and "Other Comments Or Suggestions".

**Relation To Broader Scientific Literature:**

The contribution of this paper lies in two parts. Firstly, for the CL community, this paper shows reasons for CF in CL, which provides a theoretical explanation for CF. Secondly, for the feature learning community, most works study cases where the parameters of the models are initialized according to a Gaussian distribution, while this paper also studies another scenario where the parameters are not randomly initialized (in the second stage). So this paper contributes to both CL and feature learning communities.

**Theoretical Claims:**

Yes, I have checked the proofs of the main theoretical results in this paper. I did not find any explicit errors in the proofs, and they appear to be correct.

---

> ### Author Rebuttal · Authors · 2025-03-31
>
> We thank reviewer g9HH for the careful reading and insightful suggestions. We will fix typos and improve our paper based on these suggestions in our revised version. We answer the major concerns as follows:
>
> **Q1**. The main implication of the theoretical results remains unclear.
>
> **A1**. Intuitively, our results suggest that CF occurs under two key conditions::
> - **The general feature is absent or has a weak signal while the task-specific feature has a relatively large signal**. If $\alpha _v \leq o(\alpha _u)$, after training in the first stage, only $\mathbf{\mathbf{u}}^{(1)}$ is learned by the model.
> - **The task-specific feature from the first task manifests as a random feature with a strong signal in the second task**. The  model forgets $\mathbf{\mathbf{u}}^{(1)}$ with a fast speed while learns $\mathbf{\mathbf{u}}^{(2)}$ in the second stage.
>
> Futhermore, it implies that
>
> 1. **We can overcome CF by breaking these two conditions**. To empirically validate this, we conduct experiments in Figure 2. As shown in the first row, third column of Figure 2, when $\alpha _\zeta$ decreases or $\alpha _v$ increases, CF can be mitigated.
> 2. **CNN tends to learn the strongest feature rather than the most robust feature** with (S)GD, which suggests the need for designing robust training algorithms to prevent CF.
> 3. **CF is less likely to occur when learning on similar tasks**. We can quantify the similarity through the parameters $\alpha _u,\alpha _v,\alpha _\zeta$. For example we fix $\alpha _u,\alpha _\zeta$,  as $\alpha _v$ increases, the tasks become more similar. Our results suggest that CF is less likely to occur when learning on such similar tasks.
> 4. **We provide a novel perspecitve for understanding CF**. We believe our works can inspire future works to think CF from the perspective of learning features with interesting ideas.
>
> We will include the discussion in our revised version.
>
> **Q2**. It is unclear how their analysis relates to over-parameterization, task similarity, and other factors identified in earlier research.
>
> **A2**. **Our study is conducted in the over-parameterized regime** as we assume $n \ll d$. Regarding task similarity, **we can quantify the similarity through the parameters $\alpha _u,\alpha _v,\alpha _\zeta$**, as we mentioned in **A1** in this response.
>
> **Q3**. In the right row of line 231, it lacks a theorem for $\ell^{\prime}(\mathbf{z} _1)/\ell^{\prime}(\mathbf{z} _2)$.
>
> **A3**. The formal theorem is provided in Lemmas A.4 and A.5 in Appendix A. We will clarify this in the main body of the paper in the revision.
>
> **Q4**. About the dimension of $e$'s.
>
> **A4**. We will rewrite it as "$\mathbb{O}$ is a subspace in $\mathbb{R}^{d}$".

---

### Official Review · Reviewer_iCej · 2025-03-14

**Overall Recommendation:** 3

**Summary:**

The paper aims to provide a theoretical understanding of catastrophic forgetting in a two layered CNN using a multiview data model to understand the learning dynamics of different features. The key theoretical insights are that task specific features have a larger signal than the general features and task specific features of prior tasks appear as a random feature with a strong signal in other tasks. Authors provide theoretical proof and test their findings on both simulated and real world datasets.

## Update after rebuttal

I would like to thank the authors for addressing the questions and raised concerns. The clarifications in the rebuttal increases my confidence in the work and I would increase my score accordingly. I would still strongly request the authors to improve the readibilty of their work and provide more intuition for their main theorems so that the paper reaches a wider audience and has more impact on the commmunity.

**Claims And Evidence:**

The theoretical model is simplified (two-layer CNN, task-incremental setting with only two binary classification tasks). While insightful, it’s unclear if the findings generalize to deeper CNNs or more complex continual learning settings (e.g., class-incremental CL).

**Essential References Not Discussed:**

Not to my knowledge.

**Experimental Designs Or Analyses:**

The numerical analysis in section 6.1 states that the last layer of the CNN s fixed but authors don’t provide a reason for why it is necessary to analyze the feature extraction capabilities of the model.

The experiments on real world datasets are hard to follow and the authors do not provide sufficiently convincing evidence for their claims. The expectation that models should be able to learn general features from earlier tasks that allows them to generalize on unseen tasks without even seeing these classes or training the model is unfounded. Similarly, using the performance on the current task and comparing it to unseen tasks to make the claim that the model tends to extract task-specific features isn’t convincing. The authors seem to discount the difference between representation learning and learning classifiers on top of it to learn a decision boundary. Even if the model is able to learn general features in Task 0, we cannot expect the model to be able to classify objects in task 4 without ever training the model on them. The same expectation is carried over to the T-SNE plots. Overall the reviewer did not find section 6.2 technically sound and convincing.

**Methods And Evaluation Criteria:**

Task-incremental CL is less challenging than class-incremental CL. Results may not generalize. Only two tasks are studied—how does CF evolve across many tasks?

**Other Comments Or Suggestions:**

The analysis would be more useful for the research community if the authors can add a discussion on how these insights help us design better architectures/methods to improve CL ability of the model.

**Other Strengths And Weaknesses:**

Strengths:

- The paper aims to understand catastrophic forgetting in CL and provides interesting insights which might be useful for the CL research community.
- Useful framework for analyzing feature learning in CL.

Weaknesses:
- In addition to the concerns raised earlier, the paper is difficult to follow and would benefit a lot from major restructuring and better contextualization of theoretical insights and findings. The authors can provide more intuition for their theorems and assumptions before delving into theoretical proofs. This would help a broader audience to understand the key takeaways of the paper.

- Missing discussions on other CL approaches (e.g., regularization-based, architectural methods).

**Questions For Authors:**

Q1) Can the authors comment on how they believe the insights from their analysis can benefit the CL research community and if these can be used to design better CL methods or benchmarks. Reviewer believe this is an important section missing in the paper.

Q2) Do the authors believe their findings would generalize to deeper architectures and more complex CL settings (e.g. Class-IL)?

**Relation To Broader Scientific Literature:**

The paper provides theoretical insights to understand catastrophic forgetting in two layered CNN. However it lacks suggestions on how these insights can be utilized to design a CL method and there is limited evidence to suggest the findings will generalize to commonly used architectures on complex datasets.

**Theoretical Claims:**

Reviewer was not able to validate the correctness of the proofs and theoretical claims.

---

> ### Author Rebuttal · Authors · 2025-03-31
>
> We thank the thoughtful and insightful review, and we answer the concerns as follows:
>
> **Q1**. Generalizing the simple theoretical model to complex settings:
>
> **A1**. Studying the learning dynamics in a two-layer CNN and multi-view model is a widely used approach for understanding deep learning. Despite its simplicity, our results are supported by our numerical and real-world results.
>
> We agree with the reviewer that task-IL is less challenging. However, **even within task-IL, a theoretical understanding about CF remains under-explored, particularly in CNNs**. Moreover, we believe our analysis provides a novel perspective on CF by analyzing feature learning. Our framework is flexible and can be extended to more complex settings, please refers to **Q1** in our response to Reviewer QA6F.
>
> **Q2.** Why analyze feature extraction.
>
> **A2.** By fixing the last layer, we can focus on the learning dynamics of features to find the hidden reason behind CF. **It is crucial for identifying whether and how certain features contribute to CF**. Our setup of numerical empiriments follows the theoretical framework.
>
> **Q3**. Intuitions for theorems and assumptions.
>
> **A3**. Yes, the intuition behind our assumptions is shown in Remark 3.6. We additionally provide intuitions:
>
> **An example on two-task-IL to illustrate our data model** ($\epsilon \sim U(\\{+1,-1\\})$):
>    - Task 1: Data $\mathbf{x}$ with label $y$ contains features $\\{ y \alpha _u \mathbf{e} _1,  y \alpha _v \mathbf{e} _{rob}, \epsilon \alpha _{\zeta}\mathbf{e} _2 \\}$ and noises.
>    - Task 2, Data $\mathbf{x}$ with label $y$ contains features $\\{ y \alpha _u \mathbf{e} _2,  y \alpha _v \mathbf{e} _{rob}, \epsilon \alpha _{\zeta}\mathbf{e} _1 \\}$ and noises.
>
> **Note**: All of $\\{ \mathbf{e} _1, \mathbf{e} _{rob}, \mathbf{e} _2 \\}$ appear in each task, but only $y\alpha _u\mathbf{e} _{\tau}$ and $y\alpha _v\mathbf{e} _{rob}$ are correlated with the true label in task $\tau$, which means the two features can be both used for classification in task $\tau$. $\epsilon\alpha _{\zeta}\mathbf{e} _{3-\tau}$ is called random feature, which  is uncorrelated with the true label. In such a model, **the task-specific feature is only used in its respective task**. $\alpha _u, \alpha _v, \alpha _{\zeta}$ control the strength of the features.
>
> To analyze the learning dynamics, we quantify the learning speed of feature components in the first and second stages.
> 1. **Initialization**, Lemma 4.1 shows the order of $\max _{c\in[C]} \langle \mathbf{w} _{c}, \mathbf{u}^{(\tau)} \rangle, \max _{c\in[C]} \langle \mathbf{w} _{c}, \mathbf{v} \rangle, \max _{c\in[C]} \langle \mathbf{w} _{c}, \mathbf{\zeta} \rangle, \max _{c\in[C]} \langle \mathbf{w} _{c}, \mathbf{\xi} \rangle.$
> 2. **First stage**, we analyze the learning speed of each component and find that when $\alpha _v \leq o(\alpha _u)$, $\mathbf{u}^{(1)}$ is captured by the model.
> 3. **Second stage**, Corollary 5.3 shows the order of each component at the beginning of the second stage. We then analyze the learning speed of each component and observe that $\max _{c\in[C]} \langle \mathbf{w} _{c}, \mathbf{\zeta} \rangle$ decreases with the rate depending on $\alpha _\zeta$. Specifically, when $\alpha _\zeta \geq \Omega(\alpha _u)$, the decline becomes significant, leading to forgetting.
> 4. **Conclusion** We identify two key conditions leading to forgetting:
>    1. $\alpha _v \leq o(\alpha _u)$. The common feature is absent or has weak signal.
>    2. $\alpha _\zeta \geq \Omega(\alpha _u)$. The random feature has a strong signal.
>
> **Q4**. Inconvincing evidence on real-world datasets.
>
> **A4**.  The reviewer misunderstands our feature definitions; we clarify this in **Q3** in this response. Based on our definition, we **evaluate the performance of the second task at the end of the first stage** (Figure 3) to study the learned feature at the end of the first stage is task-specific or general:
> - If the model learns a general feature on first task, it can achieve a great performance on the unseen second task (Task-0 and Task-4 in CIFAR-10).
> - If the model learns the task-specific feature in first task, it can not achieve a great performance on the unseen second task (Most Case in Figure 3). **This aligns with the condition $\alpha _v \leq o(\alpha _u)$**.
>
> **Note:  We do not compare the performance of model on current and unseen task**
>
> Additionally, Figures 4 and 5 analyze learning on Task-0 and Task-3 in CIFAR-10, where forgetting occurs:
> - Figure 4 provides evidence for the existence of random features in real-world datasets, as features from task 1 and task 2 exhibit significant overlap, **which aligns the condition $\alpha _\zeta \geq \Omega(\alpha _u)$**.
> - Figure 5 demonstrates that in the second stage, the model relies on task 2's task-specific feature for classification, and task 2's task-specific feature is helpless for task 1.
>
> **Q5**. Impact on CL research.
>
> **A5**. Please refers to **Q1** in our response to Reviewer g9HH.

---

### Decision · Program_Chairs · 2025-05-01

**Decision:**

Accept (poster)

**Comment:**

This submission studies the catastrophic forgetting of continual learning in two-layer convolutional neural networks. The authors provide principled analysis under this regime by means of the multi-view model context. Extensive experiments support the theoretical findings and claim.

The submission received the review of four reviewers, which respectively recommended 3, 4, 5, 3. Basically, their initial concerns place on the unclear implication of the theoretical results, requiring more clarity in the technical deduction and missing related works with the proper discussion. After the authors's substantial rebuttal, most concerns have been well addressed. AC have checked the reviewer's questions and the authors' reply in the interaction. Overall, AC agrees with the reviewers' suggestion and tends to recommend "Acceptance". Hope the authors carefully incorporate the reviewers' suggestion into the main body of the manuscript.